# CoFact: Conformal Factuality Guarantees for Language Models under Covariate Shift

**Zirui Hu[1], Zheng Zhang[1], Yingjie Wang[1]\*, Leszek Rutkowski[2], Dacheng Tao[1]\***
[1]Generative AI Lab, College of Computing and Data Science, Nanyang Technological University
[2]The AGH University of Krakow
`zirui.hu@ntu.edu.sg`,
`{zhengz1999.ustc, yingjiewang1201, dacheng.tao}@gmail.com`,
`rutkowski@agh.edu.pl`

## Abstract

Large Language Models (LLMs) excel in natural language processing (NLP) tasks but often generate false or misleading information, known as hallucinations, raising reliability concerns in high-stakes applications. To provide statistical guarantees on the factuality of LLM outputs, conformal prediction based techniques have been proposed. Despite their strong theoretical guarantees, they rely heavily on the exchangeability assumption between calibration and test data, which is frequently violated in real-world scenarios with dynamic covariate shifts. To overcome this limitation, we introduce **CoFact**, a conformal prediction framework that uses online density ratio estimation to adaptively reweigh calibration data, ensuring alignment with evolving test distributions. With this approach, CoFact bypasses the exchangeability requirement and provides robust factuality guarantees under non-stationary conditions. To theoretically justify CoFact, we establish an upper bound on the gap between the actual hallucination rate and the target level $\alpha$, demonstrating that the bound asymptotically approaches zero as the number of rounds and calibration samples increase. Empirically, CoFact is evaluated on MedLFQA, WikiData, and the newly introduced **WildChat+** dataset, which captures real-world covariate shifts through user-generated prompts. Results demonstrate that CoFact consistently outperforms existing methods, maintaining reliability even under dynamic conditions.

## 1 Introduction

Large Language Models (LLMs) have demonstrated exceptional performance across a wide range of natural language processing (NLP) tasks (Touvron et al., 2023; Devlin et al., 2019; OpenAI, 2023). Despite their impressive capabilities, their reliability and trustworthiness remain significant concerns. A critical issue is hallucination, where LLMs generate false or misleading information (Nadeau et al., 2024) that can lead to severe consequences in sensitive areas like healthcare (Jung, 2025; Ren et al., 2026), finance (Kang & Liu, 2023), and legal advice (Dahl et al., 2024). These limitations pose significant barriers to the broader adoption of LLMs in critical applications.

To address this issue, recent research have sought to improve the reliability of their outputs (Lewis et al., 2020; Chuang et al., 2023; Nakano et al., 2022; Ho et al., 2024). While these methods enhance factuality, they fall short of providing precise statistical guarantees, which are essential for high-stakes applications. To bridge this gap, a line of work has explored the use of conformal prediction to establish statistical guarantees on the factuality of LLM outputs. Specifically, Mohri & Hashimoto (2024) proposed splitting LLM-generated outputs into atomic sub-claims and filtering out those with factuality scores below a threshold determined via conformal inference, thereby offering marginal guarantees on factuality. Building on this, Cherian et al. (2024) extended the framework to provide subgroup-specific guarantees using conditional conformal prediction (Gibbs et al., 2025).

Although conformal inference-based methods provide factuality guarantees, they rely heavily on the assumption of exchangeability between calibration and test data (Vovk et al., 2005). In practice,

---
\*Corresponding authors

however, this assumption is frequently violated due to factors such as topic drift (Reimer et al., 2023) and changes in user composition (Li et al., 2023). For example, Reimer et al. (2023) observed that the frequency of Covid-19-related terms in user queries fluctuated dramatically during the pandemic, exhibiting distinct peaks and troughs over time. Similar dynamics are also found in our analysis of real-world user prompt data, where topics evolve rapidly, as discussed in Section 6.2. Under such conditions, the distribution of test prompts can deviate significantly from that of the calibration set, thereby violating the exchangeability assumption.

Since the exchangeability assumption underpins the theoretical reliability of conformal prediction, its violation fundamentally compromises these guarantees, rendering such methods ineffective for ensuring reliable factuality in dynamic, real-world settings. To address this critical limitation, in this paper, we aim to answer the following question:

> *Given a stream of prompts from an unknown and dynamically evolving distribution,*
> *how can we provide factuality guarantees for the outputs of LLMs?*

Answering this question presents two key challenges: (1) test samples arrive sequentially under continuous covariate shifts, making conformal prediction methods handling covariate shift that require static density ratio estimation across the entire test dataset infeasible (Tibshirani et al., 2019); and (2) in our scenario, unlike existing online conformal prediction methods (Gibbs & Candes, 2021; Gibbs & Candès, 2024; Areces et al., 2025), we do not have access to ground-truth labels for test samples after predictions, which prevents direct application of these methods.

To address these challenges, we introduce **CoFact**, a novel conformal prediction framework that integrates techniques from online learning into conformal prediction framework seamlessly. CoFact handles the continuous shifting by employing an online density ratio estimation mechanism that dynamically activates and updates multiple expert models across different time intervals. This adaptive approach enables CoFact to effectively track and learn the evolving density ratios between calibration and test distributions in real time. Leveraging these density ratio estimates, CoFact strategically reweighs calibration examples to align with the shifting test distribution. Through such integration, CoFact bypasses the traditional exchangeability assumption, providing robust reliability guarantees for LLMs even under a continual shifting prompt stream.

We theoretically establish an upper bound on the gap between the actual hallucination rate and the user-specified hallucination level $\alpha$ under shifting distributions, demonstrating that this bound asymptotically approaches zero as the number of rounds and calibration samples increase. This analysis offers a novel perspective that addresses limitations in existing methodologies.

To empirically demonstrate the effectiveness of CoFact, we evaluate CoFact on two well-established benchmarks, MedLFQA and WikiData, as well as a newly introduced benchmark, **WildChat+**. Built upon WildChat (Zhao et al., 2023), WildChat+ includes prompts generated by real users, effectively capturing real-world covariate shifts and enabling a more comprehensive evaluation. Experimental results demonstrate that CoFact significantly outperforms existing conformal prediction methods that rely on the exchangeability assumption, consistently maintaining factuality guarantees even under dynamically shifting distributions. These findings underscore CoFact's effectiveness in providing reliability guarantees in complex and dynamic real-world scenarios.

In summary, our contributions are as follows:

- **Novel Framework:** We propose **CoFact**, a conformal prediction framework designed to provide reliability guarantees for LLMs in the presence of continually shifting distributions.

- **Theoretical Guarantees:** We present rigorous theoretical analysis, establishing an upper bound on the gap between the actual hallucination rate and the target level $\alpha$ under shifting distributions. This analysis provides a solid foundation for CoFact's reliability in dynamic environments.

- **New Dataset:** To enable robust evaluation in real-world scenarios, we introduce **WildChat+**, which contains real user prompts along with LLM-generated responses and factuality annotations.

- **Extensive Experiments:** We conduct a comprehensive set of experiments across multiple benchmarks, including MedLFQA, WikiData, and WildChat+. The results demonstrate CoFact's effectiveness in maintaining reliability guarantees under diverse and evolving distributions.

## 2 RELATED WORK

**Hallucination and Its Mitigation**  Hallucination in LLMs represents a critical research challenge, driving extensive efforts to improve their factuality. Based on the stage of application, these efforts can generally be divided into two categories: inference-level and training-level methods. At the inference stage, a common approach is retrieval-augmented generation (RAG), which grounds model responses in external knowledge sources (Lewis et al., 2020; Fan et al., 2024; Wang et al., 2025). Training-level strategies include reinforcement learning from human feedback (RLHF) (Bai et al., 2022; Ouyang et al., 2022), supervised fine-tuning with factual supervision (Lin et al., 2022; Nakano et al., 2022), and factuality-oriented decoding techniques (Chuang et al., 2023; Lee et al., 2022).

In addition to these heuristic approaches, recent research has introduced methods with statistical guarantees. For instance, Mohri & Hashimoto (2024) applied conformal prediction to LLMs to provide marginal guarantees on factuality, while Cherian et al. (2024) extended this framework to group-wise guarantees using conditional conformal prediction (Gibbs et al., 2025). However, these methods are built on the assumption of exchangeability between calibration and test data—an assumption that is often violated in real-world covariate shifts. This highlights the need for more robust methods that are better suited to practical scenarios.

**Conformal Prediction**  Conformal prediction (CP) offers a principled framework for constructing prediction sets with finite-sample coverage guarantees (Shafer & Vovk, 2008; Vovk et al., 2005). Motivated by the need for more reliable language models, recent work has begun using CP to calibrate LLM outputs (Campos et al., 2024). At the response level, Kumar et al. (2023) and Quach et al. (2023) use CP to flag low-confidence generations and improve reliability. At the token level, Ulmer et al. (2024) and Ravfogel et al. (2023) integrate CP into decoding to guide token selection and enhance text quality. Although these approaches show promising empirical gains, they typically produce sets of candidate outputs rather than a single response with a guarantee of factuality—yet in many applications, returning one reliable answer is more practical than returning a set.

A second, closely related line of work extends CP to online and nonstationary settings. Gibbs & Candes (2021) introduced adaptive conformal inference (ACI) to maintain coverage under distribution shift, and Gibbs & Candès (2024) proposed adaptive step-size schemes to improve robustness. More recent methods provide stronger guarantees at a finer temporal resolution—controlling coverage beyond simple averages over the full horizon—via more advanced online learning techniques (Areces et al., 2025; Bhatnagar et al., 2023). However, most online CP frameworks assume that ground-truth labels are revealed immediately after each prediction. This assumption is unrealistic for hallucination mitigation in LLMs, where correctness feedback is often missing or delayed. As a result, existing online CP methods do not directly apply, motivating approaches that can operate under absent supervision.

## 3 PRELIMINARIES

**Conformal Prediction**  Conformal prediction is a statistical framework that converts the outputs of a black-box predictor (e.g., a single label) into prediction sets (e.g., a list of labels) that are guaranteed to include the true label with a user-specified confidence level of $1 - \alpha$. Formally, given an i.i.d. calibration set $\{(X_i, Y_i)\}_{i=1}^n$, where $X_i$ and $Y_i$ represent the features and labels, respectively, and a test sample $X_{n+1}$ that is exchangeable with the calibration data, conformal prediction constructs a prediction set $\hat{C}(X_{n+1})$ such that the true label $Y_{n+1}$ is included with probability at least $1 - \alpha$:

$$\mathbb{P}(Y_{n+1} \in \hat{C}(X_{n+1})) \geq 1 - \alpha. \tag{1}$$

To introduce the basic ideas behind conformal inference, we first define some notation, following Tibshirani et al. (2019). We denote by $\text{Quantile}(\alpha; \Psi)$ the level-$\alpha$ quantile of a distribution $\Psi$. Formally, for $X \sim \Psi$, the quantile is defined as:

$$\text{Quantile}(\alpha; \Psi) = \inf\{x : \mathbb{P}[X \leq x] \geq \alpha\}.$$

For empirical distributions, we denote the quantile of a multiset of values $v_1, \ldots, v_n$ as:

$$\text{Quantile}(\alpha; \{v_i\}_{i=1}^n) = \text{Quantile}\left(\alpha; \frac{1}{n}\sum_{i=1}^n \delta_{v_i}\right),$$

where $\delta_a$ represents a point mass at $a$ (i.e., the distribution that places all its probability mass at $a$).

The central idea of conformal prediction involves the use of conformity score $V(X_i, Y_i)$, which quantifies how well the label $Y_i$ corresponds to the features $X_i$. Using this score, the prediction set $\hat{C}(X_{n+1})$ is constructed by including all candidate labels $y$ for which the conformity score $V(X_{n+1}, y)$ exceeds or meets a threshold $\tau$. Given the conformity scores from the calibration set $\{V(X_i, Y_i)\}_{i=1}^n$, the threshold $\tau$ is determined as the $(1-\alpha)$-quantile of these scores combined with $\{\infty\}$ to ensure proper coverage. Formally,

$$\tau = \text{Quantile}\left(1 - \alpha; \{V(X_i, Y_i)\}_{i=1}^n \cup \{\infty\}\right). \tag{2}$$

**Conformal Factuality Control** To generate a single response with guaranteed factuality, rather than a prediction set containing multiple potential factual responses, Mohri & Hashimoto (2024) proposes treating each response as a set of atomic claims and using conformal prediction to filter out hallucinated claims. Specifically, their method assumes access to an annotated calibration set consisting of $n$ i.i.d. prompt-response-claim-label tuples, denoted as $D_0 = \{(P_i, R_i, \mathbf{C}_i, \mathbf{W}_i)\}_{i=1}^n$. Here, $P_i$ represents the prompt for sample $i$, $R_i$ is the corresponding response generated by the LLM, $\mathbf{C}_i = \{C_{i,j}\}_{j=1}^{k_i}$ denotes the set of claims extracted from $R_i$, and $\mathbf{W}_i = \{W_{i,j}\}_{j=1}^{k_i}$ represents the binary factuality labels for each claim, where $W_{i,j} = 1$ indicates that $C_{i,j}$ is factual, and $W_{i,j} = 0$ indicates that it is hallucinated.

The objective is to output a filtered response $F(\mathbf{C}_{n+1})$ for a test sample $(P_{n+1}, R_{n+1})$, which is exchangeable with the calibration data, such that the probability of including any hallucinated claims is bounded by a pre-defined level $\alpha$:

$$\mathbb{P}\left(\exists C_{n+1,j} \in F(\mathbf{C}_{n+1}) \text{ such that } W_{n+1,j} = 0\right) \leq \alpha. \tag{3}$$

The filtered response $F(\mathbf{C}_{n+1})$ is constructed by excluding claims with low factuality scores:

$$F(\mathbf{C}_{n+1}) = \{C_{n+1,j} \in \mathbf{C}_{n+1} \mid p(C_{n+1,j}, P_{n+1}) \geq \tau\}, \tag{4}$$

where $\tau$ is the $(1-\alpha)$-quantile of the conformity scores $\{V(\mathbf{C}_i, \mathbf{W}_i)\}_{i=1}^n \cup \{\infty\}$. The conformity score $V(\mathbf{C}_i, \mathbf{W}_i)$ is defined as:

$$V(\mathbf{C}_i, \mathbf{W}_i) = \inf \{\tau \mid \forall C_{i,j} \in F(\mathbf{C}_i), W_{i,j} = 1\}, \tag{5}$$

and $p(C_{n+1,j}, P_{n+1})$ represents the factuality score, which measures how likely the claim $C_{n+1,j}$ is to be factual given the prompt $P_{n+1}$.

Building on the above framework, Cherian et al. (2024) argue that the guarantee provided by conformal factuality control is only marginal, meaning it applies globally across all test samples but does not account for specific subgroups of data. To address this limitation, they propose a group-wise guarantee inspired by conditional conformal prediction (Gibbs et al., 2025). This approach ensures that the factuality guarantee holds for all subgroups $G \in \mathcal{G}$, where the groups are defined by a family of functions. Specifically, the group-wise guarantee ensures:

$$\mathbb{P}\left(\exists C_{n+1,j} \in F(\mathbf{C}_{n+1}) \text{ such that } W_{n+1,j} = 0 \mid Z_{n+1} \in G\right) \leq \alpha \quad \text{for all } G \in \mathcal{G}. \tag{6}$$

## 4 METHODOLOGY

In this section, we provide a detailed introduction to our proposed framework, CoFact. We begin by outlining the problem setup, including the continual covariate shift settings and our goal. Next, we present an oracle method that assumes access to the true density ratio to address the covariate shift. Lastly, we introduce our practical algorithm, which leverages online density ratio estimation to operate effectively in real-world scenarios.

### 4.1 PROBLEM SETUP

For clarity and notational simplicity, we define $Z_i$ as the prompt-response pair, i.e., $Z_i = (P_i, R_i, \mathbf{C}_i)$. Thus, each sample in the calibration set can be represented as $(Z_i, \mathbf{W}_i)$. We consider an online setting with covariate shift, where we initially have access to a calibration set $D_0$ of size $n$, independently drawn from an initial distribution $\mathcal{D}_0$. At each subsequent round $t \in [T] \triangleq \{1, \ldots, T\}$, an unlabeled dataset $D_t$ of size $n_t$ is independently sampled from the current distribution $\mathcal{D}_t$, which may evolve continuously over time. For simplicity, and without loss of generality, we assume $n_t = 1$, representing the test sample arriving at time $t$ as $Z_{n+t}$. To address the challenges posed by the covariate shift, we introduce the following assumption:

Table 1: Glossary of commonly used symbols.

| Symbol | Meaning | Symbol | Meaning |
|---|---|---|---|
| $P_i$ | $i$-th prompt | $Z_i$ | Tuple $(P_i, R_i, \mathbf{C}_i)$ |
| $R_i$ | Response to $P_i$ generated by the LLM | $D_0, \mathcal{D}_0$ | Calibration dataset/distribution |
| $\mathbf{C}_i$ | Claims parsed from $R_i$ | $D_t, \mathcal{D}_t$ | Test dataset/distribution at time $t$ |
| $\mathbf{W}_i$ | Factuality labels of $\mathbf{C}_i$ | $r_t^*, \hat{r}_t$ | True/estimated density ratio |
| $C_{i,j}$ | $j$-th claim of $\mathbf{C}_i$ | $w_t^*, \hat{w}_t$ | True/estimated importance weights |
| $W_{i,j}$ | Factuality label of $C_{i,j}$ | $F_t, \hat{F}_t$ | Filtering function at round $t$ using $w_t^*/\hat{w}_t$ |

**Assumption 1** (Continuous Covariate Shift). *For any $Z \in \mathcal{Z}$ in the prompt-response space and any round $t \in [T]$, the conditional distribution of $\mathbf{W}$ given $Z$ remains unchanged, i.e.,*

$$\mathcal{D}_t(\mathbf{W} \mid Z) = \mathcal{D}_0(\mathbf{W} \mid Z),$$

*and the density ratio between $\mathcal{D}_t$ and $\mathcal{D}_0$ satisfies:*

$$r_t^*(Z) = \frac{\mathcal{D}_t(Z)}{\mathcal{D}_0(Z)} \leq B < \infty.$$

**Objective**  Our objective is to generate a filtered response $\hat{F}_t(\mathbf{C}_{n+t})$ for each test sample $Z_{n+t}$ at round $t$, ensuring that the probability of including any hallucinated claims remains below a pre-defined threshold $\alpha$. Given the challenges of providing exact guarantees at each time step under non-stationary distributions, we adopt the metric of prior works (Gibbs & Candes, 2021; Gibbs & Candès, 2024) and focus on bounding the gap between the average hallucination rate over $T$ rounds and the target level $\alpha$: $\left| \frac{1}{T} \sum_{t=1}^{T} \widehat{\mathrm{err}}_t - \alpha \right|$, where the error indicator for round $t$ is defined as:

$$\widehat{\mathrm{err}}_t = \mathbb{1}\left[ \exists C_{n+t,i} \in \hat{F}_t(\mathbf{C}_{n+t}) \text{ such that } W_{n+t,i} = 0 \right]. \tag{7}$$

Here, $\hat{F}_t$ denotes the filtering function constructed by our method at round $t$, distinguishing it from $F_t$, which is constructed using the true density ratio. The latter will be introduced in the next subsection. To aid understanding, Table 1 provides a glossary of commonly used symbols.

## 4.2 Conformal Factuality Control Under Covariate Shift with Oracle

We first consider the ideal scenario where the true density ratio $r_t^*(Z)$ is available for all $t \in [T]$. In this case, a standard approach to address covariate shift is to reweigh the calibration samples and the test sample using the density ratio when calculating the threshold $\tau_t$ (Tibshirani et al., 2019). Formally, given the conformity scores computed on the calibration set $\{V_i\}_{i=1}^{n} = \{V(Z_i, \mathbf{W}_i)\}_{i=1}^{n}$ and the test sample $Z_{n+t}$, the threshold $\tau_t$ at any round $t \in [T]$ is defined as:

$$\tau_t = \mathrm{Quantile}(1 - \alpha; \sum_{i=1}^{n} w_t^*(Z_i)\delta_{V_i} + w_t^*(Z_{n+t})\delta_{\infty}), \tag{8}$$

where $w_t^*$ is the weight function derived from the normalized density ratio:

$$w_t^*(Z) = \frac{r_t^*(Z)}{\sum_{i=1}^{n} r_t^*(Z_i) + r_t^*(Z_{n+t})}. \tag{9}$$

Using this threshold, the filtered response is constructed as:

$$F_t(\mathbf{C}_{n+t}) = \{C_{n+t,j} \in \mathbf{C}_{n+t} \mid p(C_{n+t,j}, P_{n+t}) \geq \tau_t\}, \tag{10}$$

where $p(C_{n+t,j}, P_{n+t})$ represents the factuality score of the $j$-th claim given the prompt $P_{n+t}$.

**Corollary 1.** *Given a calibration set $D_0$ and a test sample $Z_{n+t}$ independent with $D_0$, if the true density ratio $r_t^*(Z)$ is available for all $t \in [T]$, then the filtered response constructed using Equation 10 with the threshold defined by Equation 8 satisfies the following guarantee:*

$$\mathbb{P}\left( \exists C_{n+t,i} \in F_t(\mathbf{C}_{n+t}) \text{ such that } W_{n+t,i} = 0 \right) \leq \alpha. \tag{11}$$

This result directly follows from Theorem 1 in Mohri & Hashimoto (2024) and Corollary 1 in Tibshirani et al. (2019).

### 4.3 Conformal Factuality Control with Online DRE

While guarantees on the hallucination rate can be established under the assumption of access to the true density ratios, the true density ratios are typically inaccessible in practice—particularly in scenarios where the underlying distribution is continuously evolving. To address this challenge, we adapt the method proposed in Zhang et al. (2023) to estimate a sequence of density ratios, $\{\hat{r}_t\}_{t=1}^T$, that approximate the true density ratios, $\{r_t^*\}_{t=1}^T$, under a dynamically changing distribution. In this section, we first reformulate the problem of online density ratio estimation (DRE) as a dynamic regret minimization problem. Next, we provide a brief overview of the online ensemble method employed to minimize dynamic regret. Finally, we describe how the estimated density ratios are integrated into the CoFact framework.

#### 4.3.1 Online DRE via Dynamic Regret Minimization

As shown by Sugiyama et al. (2012), the problem of density ratio estimation can be reformulated as a Bregman divergence minimization problem. Consequently, to accurately estimate the density ratio at each time step $t \in [T]$, we solve the following optimization problem to obtain $\hat{r}_t$:

$$\min_{r \in \mathcal{H}_r} \; L_t^\psi(r) - L_t^\psi(r_t^*), \tag{12}$$

where $L_t^\psi$ is the loss function for the density ratio, defined as:

$$L_t^\psi(r) = \mathbb{E}_{Z \sim \mathcal{D}_0} \left[ \partial\psi(r(Z))r(Z) - \psi(r(Z)) \right] - \mathbb{E}_{Z \sim \mathcal{D}_t} \left[ \partial\psi(r(Z)) \right]. \tag{13}$$

Here, $\psi$ is the associated divergence function. By choosing different forms of $\psi$, various existing density ratio estimation methods can be recovered, including LSIF (Kanamori et al., 2009), the logistic regression method (Bickel et al., 2009), and UKL (Nguyen et al., 2007).

Building on this single-round density ratio estimation, it is natural to construct a sequence of estimators $\{\hat{r}_t\}_{t \in [T]}$ that perform well over time by minimizing the cumulative loss gap:

$$\sum_{t=1}^T \left( L_t^\psi(\hat{r}_t) - L_t^\psi(r_t^*) \right).$$

**Implementation**    To implement this optimization, we make the following design choices:

- **Function Class and Divergence Function Specification**: We instantiate the density ratio function class $\mathcal{H}_r$ as a logistic regression model:

$$\mathcal{H}_r \triangleq \mathcal{H}_\theta = \left\{ \mathbf{z} \mapsto \exp(-\phi(\mathbf{z})^\top \theta) \mid \|\phi(\mathbf{z})\|_2 \leq R, \|\theta\|_2 \leq S \right\},$$

  i.e., we model the density ratio estimator $\hat{r}_t$ as $\hat{r}_t(\cdot) = \exp(-\phi(\cdot)^\top \hat{\theta}_t)$, where $\phi(\mathbf{z})$ is a feature mapping function (e.g., the representation extracted by a neural network), and $\hat{\theta}_t$ is the parameter corresponding to $\hat{r}_t$. The bounded norms of $\phi(\mathbf{z})$ and $\theta$ ensure that the generalization gap can be analyzed. Moreover, we choose the divergence function $\psi$ as:

$$\psi = \psi_{\text{LR}} \triangleq t \log t - (t+1) \log(t+1).$$

- **Empirical Risk Minimization**: Since the true distributions $\mathcal{D}_0$ and $\mathcal{D}_t$ are inaccessible in practice, we use samples from a calibration set $D_0 = \{Z_i\}_{i=1}^n$ and a test set $D_t$. At each time step $t \in [T]$, $\hat{r}_t$ is obtained by solving the following empirical risk minimization problem:

$$\min_{\theta \in \Theta} \sum_{t=1}^T \hat{L}_t(\theta) - \hat{L}_t(\theta_t^*), \tag{14}$$

  where $\Theta$ denotes the parameter space, $\theta_t^*$ is the optimal parameter corresponding to the true density ratio $r_t^*$, defined as $r_t^*(\cdot) = \exp(-\phi(\cdot)^\top \theta_t^*)$, and

$$\hat{L}_t(\theta) = \mathbb{E}_{Z \sim D_0} \left[ \partial\psi(r(Z;\theta))r(Z;\theta) - \psi(r(Z;\theta)) \right] - \mathbb{E}_{Z \sim D_t} \left[ \partial\psi(r(Z;\theta)) \right]. \tag{15}$$

Based on the above design choices, we are actually finding a sequence of parameters $\{\hat{\theta}_t\}_{t=1}^T$ to minimize the empirical dynamic regret in Equation 14.

### 4.3.2 Online Ensemble Framework for Dynamic Regret Minimization

To find the parameter sequence that minimizes dynamic regret, we adopt the online ensemble framework proposed by Zhang et al. (2023), which maintains a pool of experts. Each expert estimates the density ratio over its designated lifetime, and predictions from all active experts are aggregated to construct a global model at each time step, providing the final density ratio estimation. The framework operates through three key steps at each time step:

1. **Active-set update**: Experts are initialized with lifetimes chosen geometrically ($2^0, 2^1, 2^2, \ldots$), and are re-initialized upon the expiration of their lifetimes.

2. **Model aggregation**: The parameters of active experts are weighted based on their historical performance and aggregated to form the global model $\hat{\theta}_t$. This aggregation step enables the global model to adaptively emphasize different segments of historical data, thereby enhancing its ability to capture covariate shifts.

3. **Expert update**: Active experts update their parameters $\theta_{t,i}$ using an online Newton step (ONS) method, which minimizes the regret $\hat{L}_t^\psi(\theta_{t,i}) - \hat{L}_t^\psi(\theta_t^*)$ at the current time step.

For a comprehensive description of the algorithm, please refer to Appendix B.

### 4.3.3 The Overall Framework

After obtaining the density ratio estimator $\hat{r}_t$ parameterized by $\hat{\theta}_t$ at time step $t$, we substitute it for the true density ratio $r_t^*$ in Equation 8 to compute the threshold $\hat{\tau}_t$. This threshold is then used to filter hallucinated claims in the response:

$$\hat{\tau}_t = \text{Quantile}\left(1 - \alpha; \sum_{i=1}^n \hat{w}_t(Z_i)\delta_{V_i} + \hat{w}_t(Z_{n+t})\delta_\infty\right), \tag{16}$$

where $\hat{w}_t(Z)$ is the normalized estimated density ratio:

$$\hat{w}_t(Z) = \frac{\hat{r}_t(Z)}{\sum_{i=1}^n \hat{r}_t(Z_i) + \hat{r}_t(Z_{n+t})}. \tag{17}$$

Filtered responses are then given by:

$$\hat{F}_t(\mathbf{C}_{n+t}) = \{C_{(n+t)j} \in \mathbf{C}_{n+t} \mid p(C_{(n+t)j}, P_{n+t}) \geq \hat{\tau}_t\}. \tag{18}$$

## 5 Theoretical Guarantee

To obtain the theoretical guarantee on the hallucination rate, we need to make the following assumptions on the function class of the density ratio estimator $r_t^*$ and the property of the divergence function $\psi$.

**Assumption 2.** *The true density ratio $r_t^*$ is contained in the hypothesis space as $r_t^* \in \mathcal{H}_r = \mathcal{H}_\theta^{LR} \triangleq \{\mathbf{z} \mapsto \exp(-\phi(\mathbf{z})^\top \theta) \mid \theta \in \Theta\}$ for any $t \in [T]$ and the norm of $\theta$ and $\phi(\mathbf{z})$ are bounded by $S$ and $R$ respectively, i.e., $\|\theta\|_2 \leq S$ and $\|\phi(\mathbf{z})\|_2 \leq R$.*

**Assumption 3.** *The divergence function $\psi$ is $\mu$-strongly convex function satisfying $t\partial^3\psi(t) \leq 0$ and $\partial^3\psi(t) \leq 0$ for all $t \in dom\,\psi$.*

This assumption can be satisfied by many commonly used divergence functions such as $\psi_{\text{LS}}(t) = (t-1)^2/2$ and $\psi_{\text{LR}}(t) = t\log t - \log(t+1)$ when the input is bounded, which is guaranteed by Assumption 2.

**Assumption 4.** *There exist a burn-in time $t_0 \in \mathbb{N}$ and a constant $l > 0$ such that, for all $t > t_0$,*
$$\mathbb{E}_{Z \sim D_0}\left[\hat{r}_t(Z)\right] \geq l.$$

This assumption requires that, after a burn-in period, the estimated density ratio $\hat{r}_t$ has a strictly positive mean under the reference (calibration) distribution $D_0$. Equivalently, $\hat{r}_t$ does not collapse to values that are uniformly (or nearly uniformly) close to zero on $D_0$. The condition is mild: it rules out only the degenerate case where the estimator becomes essentially uninformative on average by assigning extremely small weights to most calibration samples.

**Theorem 1.** *Under the assumptions 1, 2, 3 and 4, with probability at least $1 - 2\delta$, the gap between the averaged hallucination rate over $T$ time steps and the target level $\alpha$ is bounded as*

$$\left| \frac{1}{T} \sum_{t=1}^{T} \widehat{\mathrm{err}}_t - \alpha \right| \leq \tilde{\mathcal{O}} \left( \max \left\{ T^{-\frac{2}{3}} V_T^{\frac{2}{3}}, T^{-\frac{1}{2}} \right\} + n^{-\frac{1}{2}} \right) \tag{19}$$

*when the parameter of the online ensemble is properly set. Here, $V_T = \sum_{t=2}^{T} \| \mathcal{D}_t(\mathbf{z}) - \mathcal{D}_{t-1}(\mathbf{z}) \|_1$ measures the variation of input densities and the notation $\tilde{\mathcal{O}}$ hides logarithmic factors of $T$ and $1/\delta$.*

In this theorem, we can observe that the gap converges to $0$ as the time horizon $T$ and calibration set size $n$ increase, and the convergence rate depends on the variation of input densities. This observation is consistent with our intuition that the more drastic the covariate shift is, the harder it is to adapt to the changing distribution. The proof of Theorem 1 is provided in Appendix F.3.

# 6 EXPERIMENTS

In this section, we demonstrate the effectiveness of CoFact through experiments on both simulated and real-world covariate shifts. For all experiments, we set the target factuality level to $1 - \alpha = 0.9$. We compare CoFact against the following baseline methods: (1) **SCP** (Mohri & Hashimoto, 2024), which employs standard conformal prediction to provide marginal factuality guarantees; and (2) **CondCP** (Cherian et al., 2024), which uses conditional conformal prediction to achieve group-wise factuality guarantees. To assess the performance of each method, we use two key metrics: **Factuality** and **Claims Retained**, defined as follows:

$$\textbf{Factuality} = 1 - \frac{1}{T} \sum_{t=1}^{T} \widehat{\mathrm{err}}_t, \quad \text{and} \quad \textbf{Claims Retained} = \frac{1}{T} \sum_{t=1}^{T} \frac{|\hat{F}_t(\mathbf{C}_{n+t})|}{|\mathbf{C}_{n+t}|}. \tag{20}$$

## 6.1 RESULTS ON SIMULATED COVARIATE SHIFTS

**Datasets** We evaluate our method under simulated continual covariate shifts using two established datasets: **MedLFQA** (Jeong et al., 2024) and **WikiData** (Cherian et al., 2024). The MedLFQA dataset is a long-form medical question-answering benchmark with answers given by experts or LLMs, which are used to evaluate the factuality for sub-claims. WikiData is constructed by generating short biographies for sampled Wikipedia names. The factuality of sub-claims is evaluated through an adapted FAcTscore procedure, leveraging evidence from Wikipedia passages.

Since neither MedLFQA nor WikiData naturally exhibits covariate shifts, we simulate such shifts as follows. The dataset is first randomly divided into a calibration set ($D_0$) and a test set ($D_{\text{test}}$) of the same size. Then, at each time step $t$, the test samples $Z_{n+t}$ are drawn from $D_{\text{test}}$ according to a time-varying distribution $\mathcal{D}_t$, which is defined as a mixture of two base distributions, $\mathcal{D}'$ and $\mathcal{D}''$. To emulate continual covariate shifts, we define four patterns for $\mathcal{D}_t$: periodic shifts following sine (**Sin**) or square wave (**Squ**) patterns, gradual linear transitions from $\mathcal{D}'$ to $\mathcal{D}''$ over $T$ time steps (**Lin**), and rapid stochastic alternations between $\mathcal{D}'$ and $\mathcal{D}''$ based on a fixed probability (**Ber**). Additional details on the dataset construction and shift simulation procedures are provided in Appendix C.1 and Appendix C.2.

**Results** We conduct experiments on MedLFQA and WikiData under the four types of simulated covariate shifts for $T = 2000$ time steps. The results for MedLFQA and WikiData are summarized in Table 2 and Table 3. Several key observations can be drawn from these tables. First, SCP experiences a significant drop in factuality under all types of covariate shifts across both datasets and fails to achieve the target factuality level of $0.9$. This highlights the vulnerability of SCP when the exchangeability assumption is violated, which can severely degrade its performance. Second, while CondCP achieves high factuality on the MedLFQA dataset, it suffers from an extremely low claims retention rate. Additionally, CondCP exhibits very low factuality on the WikiData dataset, indicating that group-wise factuality guarantees alone are insufficient for maintaining robust performance in shifting environments. Finally, among the three methods evaluated, our proposed method consistently achieves factuality closest to the target level of $0.9$ under all types of covariate shifts across both datasets, demonstrating its effectiveness in adapting to dynamic distribution changes.

Table 2: Averaged factuality and claims retained on the MedLFQA dataset under four types of shifts. Values in the range [0.89, 0.91] are highlighted in bold. Each experiment is repeated five times with different random seeds, and the results are reported as the mean ± standard deviation.

| | Lin | | Squ | | Sin | | Ber | |
|---|---|---|---|---|---|---|---|---|
| | **Factuality** | **Claims Retained** | **Factuality** | **Claims Retained** | **Factuality** | **Claims Retained** | **Factuality** | **Claims Retained** |
| **SCP** | 0.811 ± 0.014 | 0.912 ± 0.011 | 0.808 ± 0.017 | 0.911 ± 0.012 | 0.810 ± 0.014 | 0.910 ± 0.010 | 0.828 ± 0.022 | 0.910 ± 0.006 |
| **CondCP** | 0.940 ± 0.004 | 0.389 ± 0.028 | 0.937 ± 0.007 | 0.400 ± 0.030 | 0.939 ± 0.005 | 0.394 ± 0.030 | 0.949 ± 0.010 | 0.364 ± 0.057 |
| **CoFact** | **0.895 ± 0.026** | 0.715 ± 0.031 | **0.897 ± 0.022** | 0.718 ± 0.030 | **0.894 ± 0.018** | 0.715 ± 0.031 | **0.900 ± 0.019** | 0.714 ± 0.036 |

Table 3: Averaged factuality and claims retained on WikiData. Settings are the same as Table 2.

| | Lin | | Squ | | Sin | | Ber | |
|---|---|---|---|---|---|---|---|---|
| | **Factuality** | **Claims Retained** | **Factuality** | **Claims Retained** | **Factuality** | **Claims Retained** | **Factuality** | **Claims Retained** |
| **SCP** | 0.884 ± 0.006 | 0.780 ± 0.005 | 0.883 ± 0.006 | 0.781 ± 0.005 | 0.883 ± 0.006 | 0.781 ± 0.004 | 0.875 ± 0.010 | 0.782 ± 0.005 |
| **CondCP** | 0.724 ± 0.010 | 0.910 ± 0.002 | 0.726 ± 0.009 | 0.910 ± 0.003 | 0.725 ± 0.010 | 0.910 ± 0.002 | 0.716 ± 0.007 | 0.909 ± 0.004 |
| **CoFact** | **0.896 ± 0.010** | 0.748 ± 0.006 | **0.895 ± 0.009** | 0.748 ± 0.006 | **0.895 ± 0.008** | 0.748 ± 0.006 | **0.897 ± 0.008** | 0.749 ± 0.006 |

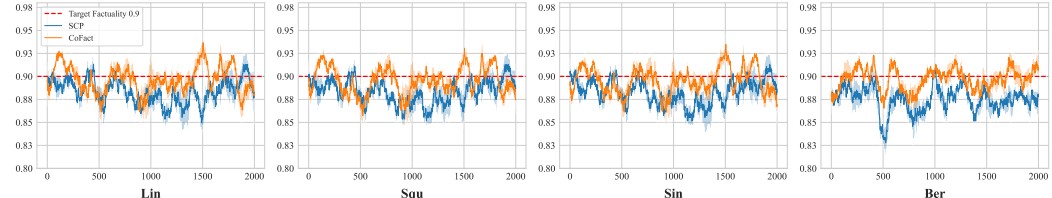

Figure 1: Factuality over time on the WikiData dataset. Each subplot corresponds to a different type of covariate shift, with the X-axis denoting the time steps and the Y-axis representing factuality. Factuality is computed using a sliding window that includes 50 steps before and after each time step. The curve shows the mean across 5 runs, while the shaded area indicates the standard deviation.

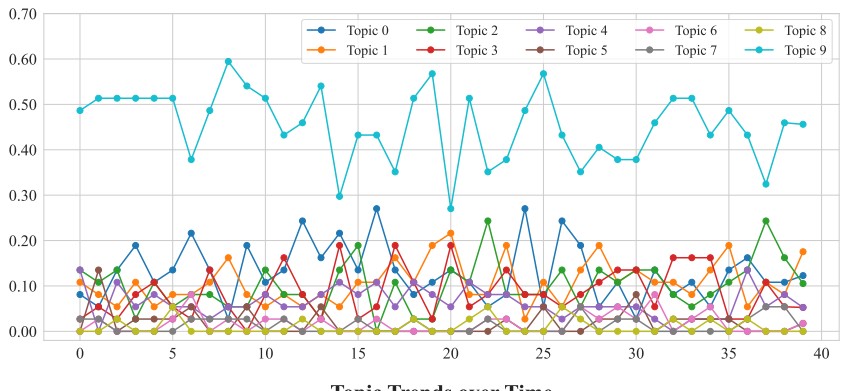

Figure 2: Topic proportions over time on WildChat+. X-axis represent the index of time intervals and Y-axis represent the proportion of each topic. Each line represent a topic identified by BERTopic.

Furthermore, we visualize how factuality evolves over time on the WikiData dataset in Figure 1. Due to CondCP's significantly lower factuality compared to SCP and our method, we exclude its results from the figure for clarity. To produce smooth curves, factuality is calculated using a sliding window that spans 50 steps before and after the current time step. The figure reveals that our method consistently maintains factuality near the target level of 0.9 over time. Notably, the curve representing our method remains above that of SCP, particularly beyond time step 1000, further underscoring the advantage of our approach in adapting to shifting distributions.

## 6.2 RESULTS ON REAL-WORLD COVARIATE SHIFTS

**Dataset and Analysis**    To evaluate our method in a real-world shifting setting, we construct a new benchmark **WildChat+** from WildChat Zhao et al. (2023), which contains user-generated prompts in the wild. For further construction details of the dataset, please refer to Appendix C.1. We conduct data analysis on the WildChat+ dataset to show that the topics of prompts in the dataset change over time. Specifically, we first use the Latent Dirichlet Allocation (LDA) algorithm to identify 10 topics

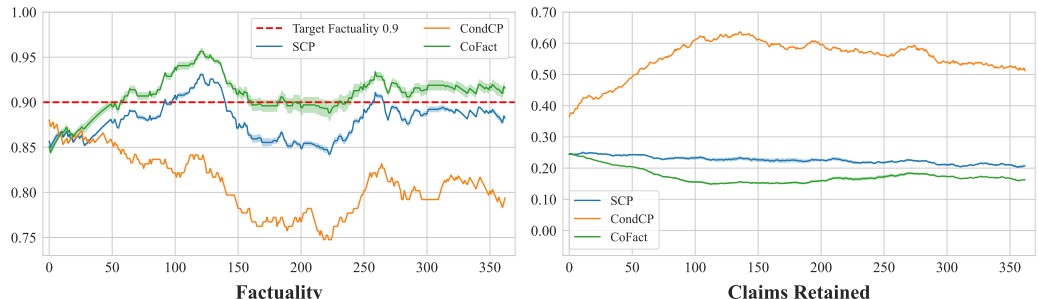

Figure 3: Factuality and retained claims ratio over time on the WildChat+ dataset. The X-axis represents the time steps, while the Y-axis displays the factuality and retained claims ratio. Factuality is computed in the same manner as described in Figure 1. The curves represent the mean across 5 runs, and the shaded areas indicate the standard deviation.

in the dataset and then split the dataset into 40 time intervals according to the timestamps of the conversations. After that, we plot the proportion of each topic in each time interval in Figure 2. From the figure, we can observe that the topics in the dataset change over time, which demonstrates the existence of a continual covariate shift in the dataset.

**Results**  To assess performance under realistic covariate shifts, we conduct experiments using the WildChat+ dataset. The data is temporally split, with the first 50% allocated for calibration and the remaining 50% reserved for testing. The results are summarized in Figure 3, leading to several key observations. First, both SCP and CondCP fail to maintain the target factuality level of $0.9$ for most time steps, revealing the limitations of existing approaches in non-stationary settings. Second, while CoFact, like SCP, falls short during the first 50 time steps due to limited adaptation data, it progressively adapts to the evolving distribution and achieves near-target factuality after time step 150, while SCP remains consistently below the target. Finally, despite higher claims retention, CondCP suffers from substantially lower factuality and thus fails to provide reliable factuality guarantees.

**Case Study**  To demonstrate the effectiveness of our method, we present a concrete example based on the filtered response to the prompt: "What is Visual Studio Code?" The filtered claims is expressed by red strikethrough text.

*Visual Studio Code is a free, ~~open-source~~ code editor developed by Microsoft. It is a ~~lightweight yet~~ powerful tool that supports various programming languages and offers features such as syntax highlighting, code completion, ~~debugging,~~ and Git integration. Visual Studio Code is highly customizable through extensions ~~and themes~~, making it popular among developers for writing ~~and debugging code~~.*

From this filtered response, we can see that our method successfully removes the hallucinated claim "open-source" while preserving the majority of the accurate information. This example highlights the capability of our approach to mitigate hallucinations in LLM-generated responses. Due to space constraints, we provide another case study in Appendix E.

## 7    CONCLUSION

In this paper, we tackle the critical challenge of providing factuality guarantees for LLMs in the presence of dynamic, real-world covariate shifts. To address the limitations of existing methods that rely on the exchangeability assumption, we introduce **CoFact**, a novel conformal prediction framework that utilizes online density ratio estimation to adaptively reweigh calibration data, ensuring alignment with evolving test distributions. Through both theoretical analysis and empirical evaluation, we demonstrate that CoFact consistently outperforms existing approaches in maintaining reliable factuality guarantees under dynamic and non-stationary conditions. The discussion of the limitations and future work can be found in Appendix G.

## 8 ETHICS STATEMENT

This paper introduces WildChat+, a derived dataset based on WildChat, which consists of real-world user-generated prompts. Due to the nature of real-world data, the dataset may contain personal information or potentially harmful content. While WildChat employs measures such as anonymization and the removal of sensitive information to address these concerns, it is still possible that some such content remains. Consequently, WildChat+ may also include similar issues. We strongly encourage users to handle the dataset responsibly and exercise caution. Beyond the concerns outlined above, we do not foresee any additional ethical issues associated with this study.

## 9 REPRODUCIBILITY STATEMENT

We have made significant efforts to ensure the reproducibility of our results. The code required to reproduce the experiments presented in this paper can be found in `https://github.com/huzr1999/CoFact`, and the implementation details are thoroughly described in Appendix C.3. Additionally, the detailed processing and construction procedures for our dataset are thoroughly described in the Appendix C.1. All assumptions underlying our theoretical results are clearly stated in Section 4.1 and 5 of the main text, and complete proofs of these results are provided in the Appendix F.3.

## 10 ACKNOWLEDGEMENTS

This project is supported by the National Research Foundation, Singapore, under its NRF Professorship Award No. NRF-P2024-001.

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

---

**Algorithm 1** CoFact's Online DRE Framework, adapted from Zhang et al. (2023)

---

**Require:** Calibration data $D_0 = \{Z_i\}_{i=1}^n$, number of time steps $T$
 1: Initialize the set of lifetime length list $\mathcal{C} = [1, 2, 4, ... \lceil \log_2 T \rceil]$
 2: Initialize the active set of experts $\mathcal{A}$ with $|\mathcal{C}|$ initialized experts
 3: **for** $t = 1, \ldots, T$ **do**
 4:     **for** $L \in \mathcal{C}$ **do**
 5:         **if** $t \equiv 0 \mod L$ **then**
 6:             Reinitialize the expert (its $\theta$, $\varepsilon$ and $v$) corresponding to the lifetime length $L$, i.e., $\mathcal{A}[\log_2 L]$
 7:     **for** $\mathcal{E}_i \in \mathcal{A}$ **do**
 8:         Update $p_{t,i}$ using $\varepsilon_{t-1,i}$ and $v_{t-1,i}$
 9:     Aggregate the global model $\hat{\theta}_t$
10:     **for** $\mathcal{E}_i \in \mathcal{A}$ **do**
11:         Update the parameters of $\mathcal{E}_i$, i.e., $\hat{\theta}_{t+1,i}$
12:         Update the potential $v_{t,i}$ and step size $\varepsilon_{t,i}$

---

## A  THE USE OF LLMS

In this work, LLMs were solely used for text polishing to enhance the clarity and readability of the manuscript. LLMs played no role in generating research ideas, problem formulations, proofs, theorems, algorithms, experiments, results, figures, or evaluations. All content produced or refined by LLMs was meticulously reviewed and validated by the authors to ensure its accuracy and consistency with the intended meaning. The technical contributions and intellectual work presented in this study are entirely the authors' own.

## B  OMITTED ALGORITHM DETAILS

In this section, we outline the approach to minimizing dynamic regret, as defined in Equation 14, using the online ensemble framework proposed by Zhang et al. (2023). At a high level, the framework maintains a pool of experts, where each expert models a density ratio estimator over its designated lifetime. At each time step, the predictions from all active experts are aggregated to form a global model, which provides the final density ratio estimation.

The overall algorithm is detailed in Algorithm 1, which consists of three main steps: active-set update (lines 3–6), model aggregation (lines 7–9), and expert update (lines 10–12). Below, we provide an overview of each step.

**Active-set update.**    The algorithm maintains a set of experts, each assigned a lifetime length chosen geometrically as $2^0, 2^1, 2^2, \ldots$. At each time step $t$, the algorithm checks if any expert's lifetime has expired. If so, the expired expert is re-initialized with updated parameters, including the model parameter $\hat{\theta}_{t,i}$, the potential $v_{t,i}$, and the step size $\epsilon_{t,i}$. Specifically, the initialization procedure is as follows:

- **Model parameter initialization.** The model parameter $\hat{\theta}_{t,i}$ is initialized to the current global model: $\hat{\theta}_{t,i} = \hat{\theta}_t$.
- **Potential initialization.** The potential $v_{t,i}$ is initialized as $v_{t,i} = 1/T$.
- **Step size initialization.** The step size $\epsilon_{t,i}$ is initialized as $\epsilon_{t,i} = \min\{1/2, \sqrt{\ln T}\}$.

**Model aggregation.**    After updating the active experts, the algorithm aggregates their predictions to form a global model. This global model is then used to make predictions for the current test sample $Z_{n+t}$. For each expert, a "potential" $v_{t,i}$ is maintained to reflect its historical performance, while a step size $\epsilon_{t,i}$ controls the update of this potential. The weights and the global model are computed as follows:

$$p_{t,i} = \frac{\epsilon_{t-1,i} v_{t-1,i}}{\sum_{i \in \mathcal{A}} \epsilon_{t-1,i} v_{t-1,i}}, \quad \text{and} \quad \hat{\theta}_t = \sum_{i \in \mathcal{A}} p_{t,i} \hat{\theta}_{t,i}.$$

**Expert update.** Once the global model $\hat{\theta}_t$ is obtained, each active expert $\mathcal{E}_i$ is updated using the newly arrived test sample $Z_{n+t}$. The expert update consists of two components: model parameter update and updates to the potential and step size.

1. **Model parameter update.** The model parameter $\hat{\theta}_{t,i}$ is updated using the online Newton step (ONS) method (Hazan et al., 2007; Agarwal et al., 2006), which incorporates second-order information to achieve efficient and adaptive updates. The update rule is given by:

$$\hat{\theta}_{t+1,i} = \Pi_\Theta^{A_{t,i}} \left[ \hat{\theta}_{t,i} - \gamma A_{t,i}^{-1} \nabla \hat{L}_t(\hat{\theta}_{t,i}) \right],$$

   where $A_{t,i}$ is the accumulated second-order matrix defined as:

$$A_{t,i} = \lambda I + \sum_{\tau=s_i}^{t} \nabla \hat{L}_\tau(\hat{\theta}_{\tau,i}) \nabla \hat{L}_\tau(\hat{\theta}_{\tau,i})^\top,$$

   and $s_i$ denotes the last initialization time step of expert $\mathcal{E}_i$. The term $\Pi_\Theta^{A_{t,i}}$ represents the projection of the updated parameter onto the feasible set $\Theta$, with the projection performed under the norm induced by the matrix $A_{t,i}$. This ensures that the updated parameter remains within the allowable parameter space.

2. **Potential and step size update.** The potential $v_{t,i}$ and step size $\epsilon_{t,i}$ are updated to reflect the expert's performance. First, we define the term $m_{t,i}$, which captures the performance gap between the expert $\hat{\theta}_{t,i}$ and the global model $\hat{\theta}_t$ over the linearized loss:

$$m_{t,i} = \frac{\langle \nabla \hat{L}_t(\hat{\theta}_t), \hat{\theta}_t - \hat{\theta}_{t,i} \rangle}{SR}.$$

   Using $m_{t,i}$, the updates are performed as follows:

   - **Potential update.** The potential $v_{t,i}$ is updated as:

$$v_{t,i} = v_{t-1,i} \cdot \left(1 + \epsilon_{t-1,i} m_{t,i}\right)^{\frac{\epsilon_{t,i}}{\epsilon_{t-1,i}}}.$$

   - **Step size update.** The step size $\epsilon_{t,i}$ is updated as:

$$\epsilon_{t,i} = \min \left\{ \frac{1}{2}, \sqrt{\frac{\ln T}{1 + \sum_{\tau=s_i}^{t} m_{\tau,i}^2}} \right\}.$$

## C  OMITTED EXPERIMENTAL DETAILS

### C.1  OMITTED DATASET DETAILS

**MedLFQA** (Jeong et al. (2024)) MedLFQA is a long-form medical question-answering dataset that integrates several previously established benchmarks. Each prompt in the dataset is paired with responses generated by either an LLM or a human. To obtain sub-claims annotated with factuality labels, Cherian et al. (2024) first use GPT-3.5-Turbo to generate responses for the prompts and GPT-4o to parse these responses into self-contained sub-claims. Then, the factuality of each sub-claim is assessed by querying GPT-3.5-Turbo, which evaluates the claims based on the LLM or human-generated responses provided for the prompts.

**WikiData** (Cherian et al. (2024)) WikiData is constructed by first sampling names from Wikipedia and then querying GPT-3.5-Turbo with the prompt: *"Write me a short biography of [NAME]."*. After that, the generated biographies are parsed into self-contained sub-claims using GPT-4o. Factuality labels for these sub-claims are assigned using a variant of the FAcTScore procedure developed by Min et al. (2023). This process involves identifying relevant Wikipedia passages using the BM25 ranking function and incorporating them into the LLM prompt to determine whether the claims are supported.

**WildChat+** To evaluate our method in a real-world dynamic setting, we construct a new dataset from WildChat Zhao et al. (2023), which features user-generated prompts in natural, uncontrolled environments. Since not all responses elicited by prompts in the dataset are suitable for the hallucination mitigation task, we first filter them by using GPT-4o-mini to identify whether the prompts can be answered using knowledge available on Wikipedia. For the filtered prompts, GPT-4o assigns relevant Wikipedia titles, and we retrieve the corresponding passages using the Wikipedia API. Finally, we apply the FAcTScore procedure, following Cherian et al. (2024), to annotate the factuality labels of the claims in the responses. Due to the high cost of annotation, we randomly sample 1450 prompts from the filtered prompts for annotation.

### C.2 Omitted Details of Shift Simulation

Here, we describe the procedure for sampling $Z_{n+t}$ from $D_{\text{test}}$ to simulate covariate shifts. Each prompt is associated with metadata, which is used to define a feature vector $\mathbf{x}$. Prompts $Z_{n+t}$ are sampled with probabilities proportional to $w(\mathbf{x}) = \exp(\mathbf{x}^T((1-\xi_t)\nu' + \xi_t\nu''))$, where $\nu'$ and $\nu''$ are predefined weight vectors, and $\xi_t \in [0, 1]$ is a time-varying factor. Since both $D_0$ and $D_{\text{test}}$ originate from the same underlying distribution, this resampling strategy effectively simulates a shift between the initial distribution $\mathcal{D}_0$ and the time-dependent test distribution $\mathcal{D}_t$.

For the MedLFQA dataset, the feature vector $\mathbf{x}$ is defined using five attributes, as detailed in Cherian et al. (2024): the length of the prompt, the length of the response, the mean log-probability of the response given the prompt, the standard error of the log-probability of the response given the prompt, and the dataset from which the prompt originates. $\nu'$ and $\nu''$ are configured such that $\nu''$ assigns higher weights to prompts with longer responses so that the sampling favors prompts with longer responses over time.

For the WikiData dataset, the feature vector $\mathbf{x}$ is constructed using the first, second, and third powers of the number of views received by the Wikipedia pages corresponding to each prompt, following Cherian et al. (2024). In this case, $\nu'$ and $\nu''$ are chosen such that $\nu''$ assigns higher weights to prompts associated with less-viewed Wikipedia pages.

The time-varying factor $\xi_t$ is introduced to model four types of covariate shifts: Linear Shift (**Lin**), Square Shift (**Squ**), Sine Shift (**Sin**), and Bernoulli Shift (**Ber**). Each type captures distinct real-world scenarios in which data distributions evolve over time:

- **Linear Shift (Lin)**: $\xi_t$ is defined as $\xi_t = t/T$, representing a gradual and continuous change in the environment over $T$ time steps.

- **Square Shift (Squ)**: $\xi_t$ alternates between 1 and 0 every $M$ steps, where $2M$ defines the period length. To model a rapidly changing environment with periodic behavior, we set $M = \Theta(\sqrt{T})$.

- **Sine Shift (Sin)**: $\xi_t$ follows a sinusoidal pattern, defined as $\xi_t = \sin(\pi t/M)$, where $M$ represents the period length. Similar to the Square Shift, we set $M = \Theta(\sqrt{T})$.

- **Bernoulli Shift (Ber)**: $\xi_t$ retains the value of $\xi_{t-1}$ with probability $p \in [0, 1]$, and switches to $1-\xi_{t-1}$ with probability $1-p$. To simulate a rapidly changing environment, we set $p = \Theta(1/\sqrt{T})$.

### C.3 Omitted Implementation Details

We preprocess the MedLFQA and WikiData datasets in accordance with the methodology described by Cherian et al. (2024). For the baseline methods, SCP and CondConf, we utilize the original code provided by Cherian et al. (2024). To implement our approach, we adapt the code from Zhang et al. (2023) to perform the online density ratio estimation procedure, using the same hyperparameter configuration as specified in their original implementation. All experiments are conducted on a machine equipped with a 16-core Ultra 9 285H CPU.

## D Additional Experimental Results

In this section, we present additional experimental results including comparisons with several oracle baselines and ablation studies of various components of our method.

Table 4: Comparison with the method handling covariate shift on MedLFQA dataset.

| | Lin | | Squ | | Sin | | Ber | | Fix | |
|---|---|---|---|---|---|---|---|---|---|---|
| | Factuality | Claims Retained | Factuality | Claims Retained | Factuality | Claims Retained | Factuality | Claims Retained | Factuality | Claims Retained |
| Total | 0.849±0.025 | 0.856±0.015 | 0.849±0.021 | 0.861±0.012 | 0.848±0.022 | 0.858±0.013 | 0.855±0.015 | 0.853±0.011 | 0.859±0.011 | 0.862±0.012 |
| Partial | 0.819±0.023 | 0.900±0.010 | 0.825±0.026 | 0.898±0.012 | 0.822±0.025 | 0.897±0.012 | 0.836±0.016 | 0.895±0.008 | 0.842±0.012 | 0.883±0.011 |
| CoFact | 0.895±0.026 | 0.715±0.031 | 0.897±0.022 | 0.718±0.030 | 0.894±0.018 | 0.715±0.031 | 0.900±0.019 | 0.713±0.027 | 0.904±0.014 | 0.713±0.027 |

Table 5: Comparison with the method handling covariate shift on WikiData dataset.

| | Lin | | Squ | | Sin | | Ber | | Fix | |
|---|---|---|---|---|---|---|---|---|---|---|
| | Factuality | Claims Retained | Factuality | Claims Retained | Factuality | Claims Retained | Factuality | Claims Retained | Factuality | Claims Retained |
| Total | 0.895±0.010 | 0.768±0.004 | 0.895±0.009 | 0.768±0.005 | 0.895±0.008 | 0.768±0.004 | 0.887±0.012 | 0.773±0.004 | 0.895±0.009 | 0.771±0.004 |
| Partial | 0.884±0.013 | 0.780±0.007 | 0.885±0.012 | 0.780±0.008 | 0.884±0.011 | 0.780±0.007 | 0.875±0.014 | 0.783±0.007 | 0.886±0.012 | 0.783±0.007 |
| CoFact | 0.896±0.010 | 0.748±0.006 | 0.895±0.009 | 0.748±0.006 | 0.895±0.008 | 0.748±0.006 | 0.897±0.008 | 0.749±0.006 | 0.896±0.010 | 0.753±0.007 |

### D.1 COMPARISON WITH METHOD IN CONFORMAL PREDICTION UNDER COVARIATE SHIFT

To provide a more comprehensive evaluation of CoFact, we compared it against existing approaches for handling covariate shifts by implementing the method proposed by Tibshirani et al. (2019), which specifically addresses covariate shift in conformal prediction. This method requires access to the entire set of test samples to train a fixed density ratio estimator between the training and test distributions. To thoroughly assess its performance, we implemented two variants of this method: (1) **Total**: This variant uses all test samples to estimate the density ratio and filter potentially hallucinated sub-claims; (2) **Partial**: This variant uses only the test samples observed during the first 500 time steps to estimate the density ratio and filter potentially hallucinated sub-claims for all time steps. It is important to emphasize that both variants are impractical in our online setting, as they require access to either the entire set of test samples or a substantial portion of them in advance, which violates the constraints of online learning. But we include them as oracle baselines to provide insights into the effectiveness of density ratio estimation in improving factuality control under covariate shifts.

In addition to the four shifting scenarios introduced in the manuscript, we also evaluated these variants under an additional scenario, **Fixed Shift (Fix)**, where the test distribution remains constant but differs from the training distribution—a setting similar to that adopted by Tibshirani et al. (2019).

The results of these experiments on the MedLFQA and WikiData datasets are presented in Table 4 and Table 5, respectively. From these results, we can make the following observations. First, The **Total** variant performs better than the **Partial** variant in terms of factuality control, indicating that access to all test samples improves density ratio estimation and enables the construction of better factuality control. Second, CoFact consistently outperforms both **Total** and **Partial** variants across nearly all shift types and datasets in terms of factuality control. Even under fixed shifts, CoFact achieves superior or comparable performance to the baselines, demonstrating its robustness in handling both stable and dynamic covariate shifts.

### D.2 COMPARISON WITH METHOD IN ONLINE CONFORMAL PREDICTION

We also compare our method with the approach proposed by Gibbs & Candes (2021), which is developed for online conformal prediction under potential distribution shifts over time. However, this method is not directly applicable to our setting, as it assumes immediate access to true labels after each prediction. Despite this mismatch, we include it as a baseline to provide a more comprehensive evaluation of CoFact.

We evaluated two update strategies from Gibbs & Candes (2021): **ACI-Simple**, defined by Equation (2) in the original paper, and **ACI-Momentum**, defined by Equation (3). Additionally, we varied the step size parameter $\gamma$ over $\{0.0005, 0.001, 0.005, 0.01\}$ to analyze its impact on performance. The results on the MedQA and Wiki datasets are presented in Table 6 and Table 7, respectively.

From the results, we can draw the following observations. First, the ability of ACI to precisely control the factuality level is highly sensitive to the choice of the step size $\gamma$. A step size that is too small results in poor control of factuality, while a step size that is too large produces overly conservative prediction sets with a low number of retained claims. Although the original paper recommends setting $\gamma = 0.005$, our results indicate that this choice leads to overly conservative filtering in our scenario. This suggests that ACI requires extensive tuning of $\gamma$ to achieve reasonable performance across different contexts. Secondly, although CoFact does not rely on access to true labels during the online learning process, it still achieves comparable factuality control performance

Table 6: Comparison with the method handling online conformal prediction on MedLFQA dataset.

| | Lin | | Squ | | Sin | | Ber | |
|---|---|---|---|---|---|---|---|---|
| | Factuality | Claims Retained | Factuality | Claims Retained | Factuality | Claims Retained | Factuality | Claims Retained |
| **ACI-Simple-0.0005** | 0.882±0.006 | 0.747±0.035 | 0.884±0.003 | 0.723±0.055 | 0.885±0.002 | 0.736±0.056 | 0.884±0.001 | 0.751±0.038 |
| **ACI-Momentum-0.0005** | 0.882±0.007 | 0.747±0.035 | 0.885±0.003 | 0.723±0.055 | 0.885±0.003 | 0.733±0.056 | 0.885±0.001 | 0.747±0.037 |
| **ACI-Simple-0.001** | 0.909±0.003 | 0.635±0.051 | 0.912±0.003 | 0.617±0.073 | 0.912±0.002 | 0.608±0.077 | 0.911±0.003 | 0.653±0.070 |
| **ACI-Momentum-0.001** | 0.910±0.004 | 0.633±0.051 | 0.913±0.004 | 0.612±0.073 | 0.912±0.002 | 0.606±0.078 | 0.913±0.003 | 0.639±0.069 |
| **ACI-Simple-0.005** | 0.975±0.003 | 0.207±0.050 | 0.977±0.003 | 0.194±0.035 | 0.978±0.003 | 0.185±0.041 | 0.972±0.006 | 0.269±0.088 |
| **ACI-Momentum-0.005** | 0.976±0.005 | 0.206±0.063 | 0.980±0.004 | 0.153±0.043 | 0.980±0.002 | 0.167±0.052 | 0.977±0.006 | 0.227±0.099 |
| **ACI-Simple-0.01** | 0.986±0.003 | 0.131±0.029 | 0.987±0.003 | 0.114±0.041 | 0.988±0.002 | 0.115±0.034 | 0.985±0.005 | 0.161±0.080 |
| **ACI-Momentum-0.01** | 0.987±0.004 | 0.121±0.065 | 0.990±0.002 | 0.088±0.030 | 0.989±0.003 | 0.102±0.044 | 0.993±0.003 | 0.069±0.026 |
| **CoFact** | 0.895±0.026 | 0.715±0.031 | 0.897±0.022 | 0.718±0.030 | 0.894±0.018 | 0.715±0.031 | 0.900±0.019 | 0.714±0.036 |

Table 7: Comparison with the method handling online conformal prediction on WikiData dataset.

| | Lin | | Squ | | Sin | | Ber | |
|---|---|---|---|---|---|---|---|---|
| | Factuality | Claims Retained | Factuality | Claims Retained | Factuality | Claims Retained | Factuality | Claims Retained |
| **ACI-Simple-0.0005** | 0.894±0.004 | 0.766±0.004 | 0.895±0.005 | 0.765±0.005 | 0.894±0.004 | 0.764±0.003 | 0.892±0.006 | 0.760±0.011 |
| **ACI-Momentum-0.0005** | 0.894±0.004 | 0.765±0.004 | 0.896±0.004 | 0.765±0.005 | 0.895±0.004 | 0.764±0.004 | 0.893±0.006 | 0.760±0.012 |
| **ACI-Simple-0.001** | 0.901±0.003 | 0.755±0.007 | 0.902±0.004 | 0.751±0.006 | 0.902±0.003 | 0.750±0.006 | 0.903±0.005 | 0.746±0.015 |
| **ACI-Momentum-0.001** | 0.902±0.002 | 0.753±0.010 | 0.903±0.003 | 0.750±0.009 | 0.903±0.002 | 0.750±0.007 | 0.903±0.005 | 0.745±0.017 |
| **ACI-Simple-0.005** | 0.940±0.006 | 0.603±0.081 | 0.940±0.007 | 0.612±0.079 | 0.941±0.008 | 0.596±0.097 | 0.945±0.015 | 0.533±0.134 |
| **ACI-Momentum-0.005** | 0.944±0.007 | 0.586±0.080 | 0.945±0.006 | 0.578±0.089 | 0.944±0.007 | 0.577±0.101 | 0.948±0.016 | 0.508±0.148 |
| **ACI-Simple-0.01** | 0.952±0.011 | 0.504±0.117 | 0.952±0.011 | 0.509±0.123 | 0.951±0.013 | 0.514±0.127 | 0.961±0.011 | 0.440±0.124 |
| **ACI-Momentum-0.01** | 0.966±0.012 | 0.387±0.140 | 0.968±0.005 | 0.339±0.084 | 0.968±0.004 | 0.335±0.078 | 0.971±0.016 | 0.297±0.116 |
| **CoFact** | 0.896±0.010 | 0.748±0.006 | 0.895±0.009 | 0.748±0.006 | 0.895±0.008 | 0.748±0.006 | 0.897±0.008 | 0.749±0.006 |

Table 8: Comparison with method handling robust online conformal prediction on MedLFQA dataset.

| | Lin | | Squ | | Sin | | Ber | |
|---|---|---|---|---|---|---|---|---|
| | Factuality | Claims Retained | Factuality | Claims Retained | Factuality | Claims Retained | Factuality | Claims Retained |
| **rACI-0.05** | 0.893±0.002 | 0.343±0.081 | 0.894±0.002 | 0.322±0.078 | 0.894±0.002 | 0.321±0.057 | 0.894±0.001 | 0.358±0.058 |
| **rACI-0.10** | 0.891±0.004 | 0.285±0.075 | 0.891±0.003 | 0.294±0.059 | 0.891±0.003 | 0.298±0.061 | 0.892±0.003 | 0.273±0.064 |
| **rACI-0.15** | 0.888±0.006 | 0.271±0.071 | 0.887±0.007 | 0.263±0.079 | 0.887±0.006 | 0.271±0.065 | 0.889±0.005 | 0.248±0.070 |
| **rACI-0.20** | 0.884±0.009 | 0.249±0.080 | 0.886±0.005 | 0.233±0.050 | 0.886±0.006 | 0.236±0.057 | 0.886±0.006 | 0.226±0.062 |
| **CoFact** | 0.895±0.026 | 0.715±0.031 | 0.897±0.022 | 0.718±0.030 | 0.894±0.018 | 0.715±0.031 | 0.900±0.019 | 0.714±0.036 |

Table 9: Comparison with method handling robust online conformal prediction on WikiData dataset.

| | Lin | | Squ | | Sin | | Ber | |
|---|---|---|---|---|---|---|---|---|
| | Factuality | Claims Retained | Factuality | Claims Retained | Factuality | Claims Retained | Factuality | Claims Retained |
| **rACI-0.05** | 0.895±0.001 | 0.731±0.021 | 0.895±0.001 | 0.737±0.018 | 0.895±0.001 | 0.732±0.019 | 0.895±0.000 | 0.738±0.015 |
| **rACI-0.10** | 0.891±0.001 | 0.741±0.016 | 0.890±0.001 | 0.742±0.016 | 0.891±0.001 | 0.740±0.016 | 0.890±0.001 | 0.728±0.021 |
| **rACI-0.15** | 0.886±0.001 | 0.717±0.027 | 0.886±0.001 | 0.717±0.028 | 0.886±0.001 | 0.719±0.030 | 0.886±0.002 | 0.713±0.029 |
| **rACI-0.20** | 0.880±0.001 | 0.719±0.025 | 0.881±0.001 | 0.712±0.027 | 0.881±0.001 | 0.715±0.022 | 0.879±0.002 | 0.734±0.015 |
| **CoFact** | 0.896±0.010 | 0.748±0.006 | 0.895±0.009 | 0.748±0.006 | 0.895±0.008 | 0.748±0.006 | 0.897±0.008 | 0.749±0.006 |

to ACI with well-tuned $\gamma$ values, further underscoring its effectiveness in our online setting. It is important to note that while ACI can achieve controlled factuality and, in some cases, retain slightly more claims than CoFact, this does not imply that ACI is superior. ACI relies on access to true labels after making predictions at each time step, a requirement that is impractical in our online setting.

### D.3 COMPARISON WITH ROBUST ONLINE CONFORMAL PREDICTION

In this section, we relax the assumption of inaccessible true labels and instead assume access to noisy labels in the online conformal prediction setting. Under this relaxed setting, we compare our method with rACI, an extension of ACI proposed by Xi et al. (2025), which is designed to handle label noise.

To simulate a weak supervision scenario, we introduced uniform random noise to the true factuality labels in our experiments and compared the performance of CoFact with rACI. The results are presented in Table 8 and Table 9 for the MedLFQA and WikiData datasets, respectively. The noise ratio was varied from 5% to 20% to evaluate the robustness of both methods under different noise conditions.

From these results, we observe that while rACI exhibits some robustness to label noise, its performance deteriorates significantly as the noise level increases. Notably, on the MedQA dataset, the Claims Retained metric for rACI is extremely low, making it impractical for real-world applications. Furthermore, rACI relies on strong assumptions about the nature of noise, specifically that label-flipping probabilities are consistent across different sub-claims and samples, and that the true noise rates are known. These assumptions are rarely realistic in practical scenarios. Given these limitations, CoFact demonstrates clear advantages in terms of practicality and robustness for factuality control in online settings.

Table 10: Experiments with varying calibration fractions on MedLFQA dataset.

| | Lin | | Squ | | Sin | | Ber | |
|---|---|---|---|---|---|---|---|---|
| | Factuality | Claims Retained | Factuality | Claims Retained | Factuality | Claims Retained | Factuality | Claims Retained |
| calibration_frac=0.05 | 0.920±0.028 | 0.613±0.094 | 0.921±0.024 | 0.612±0.097 | 0.923±0.023 | 0.622±0.099 | 0.913±0.049 | 0.621±0.189 |
| calibration_frac=0.1 | 0.902±0.015 | 0.669±0.050 | 0.900±0.019 | 0.674±0.053 | 0.904±0.018 | 0.683±0.050 | 0.911±0.029 | 0.673±0.126 |
| calibration_frac=0.3 | 0.890±0.014 | 0.722±0.057 | 0.889±0.016 | 0.721±0.056 | 0.891±0.018 | 0.733±0.057 | 0.911±0.015 | 0.666±0.047 |

Table 11: Experiments with varying calibration fractions on WikiData dataset.

| | Lin | | Squ | | Sin | | Ber | |
|---|---|---|---|---|---|---|---|---|
| | Factuality | Claims Retained | Factuality | Claims Retained | Factuality | Claims Retained | Factuality | Claims Retained |
| calibration_frac=0.05 | 0.899±0.012 | 0.744±0.019 | 0.898±0.012 | 0.744±0.019 | 0.898±0.011 | 0.743±0.020 | 0.891±0.013 | 0.760±0.016 |
| calibration_frac=0.1 | 0.905±0.013 | 0.741±0.013 | 0.905±0.013 | 0.741±0.013 | 0.905±0.013 | 0.741±0.013 | 0.897±0.015 | 0.749±0.017 |
| calibration_frac=0.3 | 0.892±0.012 | 0.751±0.011 | 0.892±0.013 | 0.750±0.011 | 0.893±0.013 | 0.750±0.011 | 0.894±0.010 | 0.754±0.009 |

Table 12: Experiments with varying expert number base on MedLFQA dataset.

| | Lin | | Squ | | Sin | | Ber | |
|---|---|---|---|---|---|---|---|---|
| | Factuality | Claims Retained | Factuality | Claims Retained | Factuality | Claims Retained | Factuality | Claims Retained |
| expert_num_base=3 | 0.883±0.016 | 0.716±0.022 | 0.888±0.019 | 0.712±0.020 | 0.887±0.022 | 0.714±0.022 | 0.896±0.012 | 0.713±0.032 |
| expert_num_base=5 | 0.888±0.022 | 0.720±0.026 | 0.887±0.022 | 0.719±0.025 | 0.891±0.021 | 0.723±0.028 | 0.899±0.020 | 0.706±0.032 |
| expert_num_base=10 | 0.892±0.020 | 0.715±0.025 | 0.896±0.020 | 0.714±0.024 | 0.900±0.021 | 0.714±0.022 | 0.894±0.014 | 0.713±0.029 |

Table 13: Experiments with varying expert number base on WikiData dataset.

| | Lin | | Squ | | Sin | | Ber | |
|---|---|---|---|---|---|---|---|---|
| | Factuality | Claims Retained | Factuality | Claims Retained | Factuality | Claims Retained | Factuality | Claims Retained |
| expert_num_base=3 | 0.900±0.011 | 0.745±0.009 | 0.899±0.011 | 0.744±0.008 | 0.899±0.011 | 0.745±0.009 | 0.893±0.015 | 0.750±0.009 |
| expert_num_base=5 | 0.895±0.007 | 0.746±0.010 | 0.894±0.007 | 0.748±0.010 | 0.894±0.007 | 0.747±0.010 | 0.889±0.013 | 0.746±0.008 |
| expert_num_base=10 | 0.901±0.016 | 0.745±0.008 | 0.903±0.015 | 0.746±0.007 | 0.903±0.015 | 0.746±0.008 | 0.901±0.012 | 0.745±0.005 |

## D.4 ABLATION STUDY ON CALIBRATION SAMPLE SIZE

We conducted experiments using different calibration set ratios, specifically $\{0.05, 0.1, 0.2\}$, on the MedLFQA dataset (corresponding to approximately 240, 480, and 1440 samples) and the WikiData dataset (approximately 425, 850, and 2550 samples) under four shift types. These experiments were designed to assess the impact of calibration set size on the performance of our method. The results for the two datasets are summarized in Table 10 and Table 11, respectively.

The results indicate that our method exhibits strong robustness to variations in calibration set size. Even with a relatively small calibration set (approximately 240 samples for MedQA and 425 samples for Wiki), our method achieves satisfactory performance in terms of both Factuality and Claims Retained. This underscores the practicality of our approach, particularly in scenarios where only limited calibration data is available.

## D.5 ABLATION STUDY ON VARYING EXPERT NUMBER

To investigate the impact of the number of experts, we adjusted the expert lifetime schedule by modifying the lifetime scaling factor from $2^i$ to $3^i$, $5^i$, and $10^i$. This change alters the number of active experts at each time step to $\log_3 t$, $\log_5 t$, and $\log_{10} t$, respectively. The experimental results on the MedLFQA and WikiData datasets, evaluated under four shift types, are summarized in Table 12 and Table 13.

The results show that our method consistently achieves reliable performance in terms of both Factuality and Claims Retained across different expert base settings. This indicates that the method is robust to variations in the number of active experts, further highlighting its adaptability to different configurations.

## D.6 ABLATION STUDY ON VARYING STEPS PER UPDATE

To further evaluate the applicability of our method in an online batch learning setup, we performed experiments using different batch sizes ($\{1, 5, 10, 20\}$) on the MedLFQA and WikiData datasets across four shift types. The results are summarized in Table 14 and Table 15.

The findings indicate that our method consistently maintains strong performance in terms of both Factuality and Claims Retained when processing incoming samples in batches. This demonstrates the suitability of our approach for online batch learning scenarios.

Table 14: Experiments with varying steps per update on MedLFQA dataset.

| | Lin | | Squ | | Sin | | Ber | |
|---|---|---|---|---|---|---|---|---|
| | Factuality | Claims Retained | Factuality | Claims Retained | Factuality | Claims Retained | Factuality | Claims Retained |
| per_step_num=5 | 0.893±0.024 | 0.723±0.020 | 0.890±0.025 | 0.722±0.027 | 0.887±0.029 | 0.726±0.023 | 0.896±0.014 | 0.712±0.030 |
| per_step_num=10 | 0.897±0.024 | 0.703±0.028 | 0.895±0.025 | 0.701±0.028 | 0.897±0.025 | 0.709±0.027 | 0.902±0.012 | 0.696±0.027 |
| per_step_num=20 | 0.899±0.024 | 0.696±0.031 | 0.900±0.025 | 0.693±0.032 | 0.899±0.025 | 0.698±0.030 | 0.904±0.011 | 0.684±0.028 |

Table 15: Experiments with varying steps per update on WikiData dataset.

| | Lin | | Squ | | Sin | | Ber | |
|---|---|---|---|---|---|---|---|---|
| | Factuality | Claims Retained | Factuality | Claims Retained | Factuality | Claims Retained | Factuality | Claims Retained |
| per_step_num=5 | 0.900±0.011 | 0.750±0.008 | 0.899±0.010 | 0.750±0.010 | 0.901±0.011 | 0.747±0.008 | 0.893±0.016 | 0.749±0.011 |
| per_step_num=10 | 0.903±0.009 | 0.748±0.006 | 0.905±0.009 | 0.749±0.006 | 0.901±0.008 | 0.748±0.007 | 0.897±0.008 | 0.747±0.011 |
| per_step_num=20 | 0.899±0.007 | 0.748±0.006 | 0.901±0.006 | 0.747±0.006 | 0.900±0.007 | 0.748±0.007 | 0.900±0.013 | 0.751±0.011 |

Table 16: Experiments with varying noise levels on MedLFQA Dataset.

| | Lin | | Squ | | Sin | | Ber | |
|---|---|---|---|---|---|---|---|---|
| | Factuality | Claims Retained | Factuality | Claims Retained | Factuality | Claims Retained | Factuality | Claims Retained |
| noise_level=0.05 | 0.898±0.021 | 0.722±0.030 | 0.897±0.018 | 0.720±0.028 | 0.899±0.021 | 0.725±0.027 | 0.909±0.016 | 0.713±0.027 |
| noise_level=0.10 | 0.892±0.024 | 0.743±0.041 | 0.891±0.021 | 0.742±0.040 | 0.893±0.025 | 0.746±0.039 | 0.902±0.021 | 0.734±0.035 |
| noise_level=0.15 | 0.884±0.028 | 0.776±0.048 | 0.879±0.024 | 0.776±0.047 | 0.882±0.028 | 0.781±0.046 | 0.893±0.026 | 0.768±0.041 |

Table 17: Experiments with varying noise levels on the WikiData Dataset.

| | Lin | | Squ | | Sin | | Ber | |
|---|---|---|---|---|---|---|---|---|
| | Factuality | Claims Retained | Factuality | Claims Retained | Factuality | Claims Retained | Factuality | Claims Retained |
| noise_level=0.05 | 0.898±0.006 | 0.755±0.008 | 0.897±0.005 | 0.755±0.008 | 0.898±0.005 | 0.755±0.007 | 0.898±0.003 | 0.752±0.006 |
| noise_level=0.10 | 0.894±0.008 | 0.761±0.010 | 0.893±0.008 | 0.761±0.010 | 0.894±0.008 | 0.761±0.009 | 0.895±0.009 | 0.758±0.009 |
| noise_level=0.15 | 0.892±0.009 | 0.765±0.010 | 0.891±0.009 | 0.765±0.010 | 0.892±0.008 | 0.765±0.010 | 0.894±0.010 | 0.761±0.011 |

### D.7 EXPERIMENTS ON VARYING NOISE LEVELS IN FEATURE REPRESENTATION

We conducted experiments to evaluate the sensitivity of our method to feature representations by introducing Gaussian noise with varying levels of variance to the feature representations on the MedLFQA and WikiData datasets. The results are summarized in Table 16 and Table 17.

To ensure a meaningful analysis, the feature representations were normalized prior to adding noise, resulting in a median feature value of approximately 0.15. As such, introducing a noise level of 0.15 constitutes a substantial perturbation to the features.

The results show that even under relatively high noise levels, our method consistently maintains robust performance in terms of both Factuality and Claims Retained. This highlights the resilience of our approach to variations and noise in feature representations.

## E MORE CASE STUDIES

**Prompt:** *What is MVC in software engineering?*

**Filtered Response:** MVC stands for Model-View-Controller, which is a design pattern commonly used in software engineering.

- Model: Represents the data and behavior of the application. ~~It acts as a bridge between the View and the Controller~~, handling the business logic and data manipulation.

- View: Represents the user interface of the application. It displays the data to the user and sends user inputs to the Controller.

- Controller: Acts as an intermediary between the Model and the View. It processes user input from the View, interacts with the Model to retrieve or update data, ~~and updates the View accordingly~~.

MVC helps to separate concerns and facilitate maintainability, reusability, and scalability of the codebase. ~~It is commonly used in web development frameworks like Ruby on Rails, ASP.NET MVC, and Spring MVC.~~

In this case, CoFact effectively removes the hallucinated claims "It acts as a bridge between the View and the Controller" while retaining most of the correct claims.

## F  THEOREM AND PROOF

### F.1  USEFUL LEMMA

**Lemma 1** (Hoeffding's Inequality). *Let $X_1, \ldots, X_n$ be independent random variables with $X_i \in [l_{\text{low}}, l_{\text{low}} + L]$ almost surely. Define the sample mean $\bar{X}_n = \frac{1}{n} \sum_{i=1}^{n} X_i$ and $\mu = \mathbb{E}[\bar{X}_n]$. Then for any $\delta > 0$, with probability at least $1 - \delta$,*

$$|\bar{X}_n - \mu| \leq L \cdot \sqrt{\frac{\log(2/\delta)}{2n}}. \tag{21}$$

**Lemma 2** (Azuma–Hoeffding Inequality). *Let $\{M_t\}_{t=1}^{n}$ be a martingale difference sequence with respect to the filtration $\{\mathcal{F}_t\}_{t=0}^{n}$, i.e.,*

$$\mathbb{E}[M_t \mid \mathcal{F}_{t-1}] = 0, \quad \forall\, t = 1, \ldots, n.$$

*Suppose the differences are bounded almost surely by*

$$|M_t| \leq c, \quad \forall\, t = 1, \ldots, n.$$

*Define the partial average $B_n = \frac{1}{n} \sum_{t=1}^{n} M_t$. Then for any $\delta > 0$ with probability at least $1 - \delta$, we have*

$$|B_n| \leq c\sqrt{\frac{2}{n} \log(2/\delta)}. \tag{22}$$

**Lemma 3** (Theorem 1 in Zhang et al. (2023)). *Suppose Assumptions 2 and 3 hold. Denote $[z]_+ = \max\{z, 0\}$. Let $d$ be the dimension of the parameter space $\Theta$. Then, for any density ratio estimator $\hat{r}_t(Z) = h(Z; \theta) \in \mathcal{H}_r$, the empirical estimation error is bounded by*

$$\frac{1}{T} \sum_{t=1}^{T} \mathbb{E}_{x \sim D_0(x)} \left[ |r_t^*(x) - \hat{r}_t(x)| \right] \leq \sqrt{\frac{4}{\mu T} \left[ \sum_{t=1}^{T} \tilde{L}_t^{\psi}(\hat{r}_t) - \sum_{t=1}^{T} \tilde{L}_t^{\psi}(r_t^*) \right]_+} + \mathcal{O}\left( \frac{\sqrt{d \log(T/\delta)}}{\mu \sqrt{n}} \right), \tag{23}$$

*provided that $h(Z, \theta)$ is bounded for any $Z \in \mathcal{Z}$ and $\theta \in \Theta$ and Lipschitz continuous.*

**Lemma 4** (Theorem 2 in Zhang et al. (2023)). *Suppose Assumptions 2 and 3 hold. Then, with probability at least $1 - \delta$, the dynamic regret of the density ratio estimator sequence $\{\hat{r}_t\}_{t=1}^{T}$ learned from Algorithm 1 is bounded by*

$$\sum_{t=1}^{T} \tilde{L}_t^{\psi}(\hat{r}_t) - \sum_{t=1}^{T} \tilde{L}_t^{\psi}(r_t^*) \leq \tilde{\mathcal{O}}\left( \max\left\{ T^{\frac{1}{3}} V_T^{\frac{2}{3}}, 1 \right\} + \frac{T}{n} \right), \tag{24}$$

*when the parameters are set as $\gamma = 3(1 + \beta)$ and $\lambda = 1$. In the above, $V_T = \sum_{t=2}^{T} \|\mathcal{D}_t(\mathbf{x}) - \mathcal{D}_{t-1}(\mathbf{x})\|_1$ measures the variation of input densities. $\beta = \exp(SR)$ represents the maximum value of the estimated density ratio $\hat{r}_t$.*

**Corollary 2.** *Suppose Assumption 2 and 3 hold. Then, with probability at least $1 - \delta$, the dynamic regret of the density ratio estimator $\hat{r}_t(\mathbf{x}) = \exp(-\phi(\mathbf{z})^\top \hat{\theta}_t)$ is bounded by*

$$\frac{1}{T} \sum_{t=1}^{T} \mathbb{E}_{\mathbf{x} \sim D_0} \left[ |r_t^*(\mathbf{x}) - \hat{r}_t(\mathbf{x})| \right] \leq \tilde{\mathcal{O}}\left( n^{-\frac{1}{2}} + \max\left\{ T^{-\frac{1}{3}} V_T^{\frac{1}{3}}, T^{-\frac{1}{2}} \right\} \right). \tag{25}$$

*when the parameters are set as $\gamma = 3(1 + \beta)$ and $\lambda = 1$. In the above, $V_T = \sum_{t=2}^{T} \|\mathcal{D}_t(\mathbf{x}) - \mathcal{D}_{t-1}(\mathbf{x})\|_1$ measures the variation of input densities.*

### F.2  NOTATIONS AND PRELIMINARIES FOR MAIN THEOREM PROOF

For clarity, here we define some notations used in the proof.

Let $\mathcal{E}_t$ denote the event that there exists any hallucinated claim in the prediction set obtained by true density ratio using at time step $t$, i.e.,

$$\mathcal{E}_t = \left\{ \exists\, C_{n+t,i} \in F_t(\mathbf{C}_{n+t}) \text{ such that } W_{n+t,i} = 0 \right\}. \tag{26}$$

Similarly, we denote the event that there exists any hallucinated claim in the prediction set obtained by estimated density ratio at time step $t$ as $\hat{\mathcal{E}}_t$, i.e.,

$$\hat{\mathcal{E}}_t = \left\{ \exists\, C_{n+t,i} \in \hat{F}_t(\mathbf{C}_{n+t}) \text{ such that } W_{n+t,i} = 0 \right\}. \tag{27}$$

As a result, we have $\mathrm{err}_t = \mathbb{1}[\mathcal{E}_t]$ and $\widehat{\mathrm{err}}_t = \mathbb{1}[\hat{\mathcal{E}}_t]$.

Recall that the initial calibration dataset is

$$D_0 = \{(P_i, R_i, \mathbf{C}_i, \mathbf{W}_i)\}_{i=1}^n.$$

At time step $t$, the algorithm observes the partially labeled data point

$$D_t = Z_{n+t} = (P_{n+t}, R_{n+t}, \mathbf{C}_{n+t}),$$

while the label $\mathbf{W}_{n+t}$ is not observed at decision time. For analytical convenience, we define the corresponding label-augmented data point

$$\tilde{D}_t = (Z_{n+t}, \mathbf{W}_{n+t}) = (P_{n+t}, R_{n+t}, \mathbf{C}_{n+t}, \mathbf{W}_{n+t}),$$

which contains the unobserved label $\mathbf{W}_{n+t}$.

We define a single filtration that captures all information available prior to making the decision at time $t$:

$$\mathcal{F}_{t-1} := \sigma\big(D_0, \{\tilde{D}_s\}_{s=1}^{t-1}, D_t\big), \qquad t = 1, \ldots, T.$$

By construction, the threshold $\hat{\tau}_t$ is $\mathcal{F}_{t-1}$-measurable, since it is determined by the initial calibration dataset $D_0$ and the previously observed data points $\{D_s\}_{s=1}^t$. The event $\hat{\mathcal{E}}_t$ is $\mathcal{F}_t$-measurable, as it depends on $\hat{\tau}_t$ and the label $\mathbf{W}_{n+t}$.

Moreover, conditional on the observed covariates $\mathbf{C}_{n+t}$ (equivalently, on $D_t$), the label $\mathbf{W}_{n+t}$ is independent of the past history encoded in $\mathcal{F}_{t-1}$. Although $\mathbf{W}_{n+t}$ is not observed by the algorithm at time $t$, it is well-defined on the underlying probability space. This distinction between observability and measurability is sufficient for the martingale-based analysis used in subsequent proofs.

### F.3 Proof of Theorem 1

**Lemma 5.** *For any $\delta > 0$, with probability at least $1 - \delta$,*

$$\left| \frac{1}{T} \sum_{t=1}^T \widehat{\mathrm{err}}_t - \frac{1}{T} \sum_{t=1}^T \mathbb{P}\left(\hat{\mathcal{E}}_t \mid \mathcal{F}_{t-1}\right) \right| \le \sqrt{\frac{2}{T} \log\left(\frac{2}{\delta}\right)}. \tag{28}$$

*Proof.* Define

$$Y_t := \widehat{\mathrm{err}}_t - \mathbb{P}(\hat{\mathcal{E}}_t \mid \mathcal{F}_{t-1}) = \mathbb{1}[\hat{\mathcal{E}}_t] - \mathbb{E}[\mathbb{1}[\hat{\mathcal{E}}_t] \mid \mathcal{F}_{t-1}], \quad t = 1, \ldots, T.$$

Since $0 \le \mathbb{1}[\hat{\mathcal{E}}_t] \le 1$, we have $|Y_t| \le 1$ almost surely. Moreover, by construction,

$$\mathbb{E}[Y_t \mid \mathcal{F}_{t-1}] = \mathbb{E}[\mathbb{1}[\hat{\mathcal{E}}_t] \mid \mathcal{F}_{t-1}] - \mathbb{E}[\mathbb{1}[\hat{\mathcal{E}}_t] \mid \mathcal{F}_{t-1}] = 0.$$

Thus $\{Y_t\}_{t=1}^T$ is a martingale difference sequence with respect to $\{\mathcal{F}_t\}_{t=0}^T$.

Applying the Azuma–Hoeffding inequality yields that, with probability at least $1 - \delta$,

$$\left| \sum_{t=1}^T Y_t \right| \le \sqrt{2T \log\left(\frac{2}{\delta}\right)}.$$

Dividing both sides by $T$ completes the proof. $\qquad\square$

**Lemma 6.** *Suppose Assumption 1 holds. Let $\hat{r}_t$ and $r_t^*$ denote the estimated and true density ratios at time step $t$, respectively, and let $Z_{n+t}$ be a test sample drawn independently at time step $t$. Then, the following inequality holds with probability at least $1 - \delta$:*

$$\frac{1}{T} \sum_{t=1}^T |\hat{r}_t(Z_{n+t}) - r_t^*(Z_{n+t})| \le \frac{B}{T} \sum_{t=1}^T \mathbb{E}_{Z \sim \mathcal{D}_0} \left[|r_t^*(Z) - \hat{r}_t(Z)|\right] + \sqrt{\frac{2}{T} \log\left(\frac{2}{\delta}\right)} \cdot \beta', \tag{29}$$

*where $\beta'$ is a bound on the differences $|\hat{r}_t(z) - r_t^*(z)|$ for all $z$.*

*Proof.* Here, we define a filtration to facilitate the analysis. Let

$$\mathcal{F}'_{t-1} = \sigma\left(D_0, \{D_s\}_{s=1}^{t-1}\right), \quad t = 1, \dots, T.$$

For the purpose of the martingale argument, we consider the filtration containing only information available strictly before the test sample at time $t$. This filtration may differ from those used elsewhere; however, all probability statements are taken over the same underlying probability space and can therefore be combined via union bounds

For each time step $t$,

$$U_t = |\hat{r}_t(Z_{n+t}) - r_t^*(Z_{n+t})| - \mathbb{E}_{Z \sim \mathcal{D}_t}\left[|\hat{r}_t(Z) - r_t^*(Z)|\right]. \tag{30}$$

By construction, $\hat{r}_t$ is measurable with respect to $\mathcal{F}'_{t-1}$, and $Z_{n+t}$ is drawn independently from $\mathcal{D}_t$. Thus, we have $\mathbb{E}[U_t \mid \mathcal{F}'_{t-1}] = 0$. Therefore, $\{U_t\}_{t=1}^T$ forms a martingale difference sequence with respect to the filtration $\{\mathcal{F}'_t\}_{t=0}^T$. Additionally, since $|\hat{r}_t(z) - r_t^*(z)|$ is bounded by $\beta'$ for all $z$, we have $|U_t| \leq \beta'$ almost surely.

Applying the Azuma–Hoeffding inequality (Lemma 2), we obtain that with probability at least $1 - \delta$,

$$\left| \frac{1}{T} \sum_{t=1}^T |\hat{r}_t(Z_{n+t}) - r_t^*(Z_{n+t})| - \frac{1}{T} \sum_{t=1}^T \mathbb{E}_{Z \sim \mathcal{D}_t}\left[|\hat{r}_t(Z) - r_t^*(Z)|\right] \right|$$
$$\leq \sqrt{\frac{2}{T} \log\left(\frac{2}{\delta}\right)} \cdot \beta', \tag{31}$$

Using the change-of-measure technique, we have

$$\mathbb{E}_{Z \sim \mathcal{D}_t}\left[|\hat{r}_t(Z) - r_t^*(Z)|\right] = \mathbb{E}_{Z \sim \mathcal{D}_0}\left[r_t^*(Z)|\hat{r}_t(Z) - r_t^*(Z)|\right]$$
$$\leq B\mathbb{E}_{Z \sim \mathcal{D}_0}\left[|r_t^*(Z) - \hat{r}_t(Z)|\right], \tag{32}$$

where $B$ is the bound on $r_t^*(Z)$. Averaging over $t = 1, \dots, T$ gives

$$\frac{1}{T} \sum_{t=1}^T \mathbb{E}_{Z \sim \mathcal{D}_t}\left[|\hat{r}_t(Z) - r_t^*(Z)|\right] \leq \frac{B}{T} \sum_{t=1}^T \mathbb{E}_{Z \sim \mathcal{D}_0}\left[|r_t^*(Z) - \hat{r}_t(Z)|\right]. \tag{33}$$

Combining Equation 31 and Equation 33, we prove the lemma. $\qquad \square$

**Lemma 7.** *Given the hypothesis space $\mathcal{H}_r$ satisfying Assumption 2, let $\beta' = \max_{r \in \mathcal{H}_r, z \in \mathcal{Z}} |r(z) - r_t^*(z)|$ and $G_h = \max_{Z \in \mathcal{Z}, \theta \in \Theta} \|\nabla h(Z, \theta)\|_2$. For any sequence of density ratio estimators $\{\hat{r}_t\}_{t=1}^T$ and corresponding true density ratio $\{r_t^*\}_{t=1}^T$ under distribution $\mathcal{D}_0$, the following bound holds with probability at least $1 - \delta$:*

$$\left| \frac{1}{T} \sum_{t=1}^T \mathbb{E}_{Z \sim \mathcal{D}_0}[|\hat{r}_t(Z) - r_t^*(Z)|] - \frac{1}{T} \sum_{t=1}^T \mathbb{E}_{Z \sim D_0}[|\hat{r}_t(Z) - r_t^*(Z)|] \right| \leq 2\beta' \sqrt{\frac{d \log\left(\frac{6SG_h T}{\delta}\right)}{2n}} + \frac{2}{T} \tag{34}$$

*Proof.* We aim to bound the discrepancy between the population and empirical absolute errors for a sequence of density ratio estimators. The main difficulty is that the estimators $\hat{r}_t$ are data-dependent, which prevents a direct application of concentration inequalities such as Hoeffding's inequality. To address this issue, we employ a standard covering number argument: we first control the deviation uniformly over an $\epsilon$-net of the hypothesis space $\mathcal{H}_r$, and then extend the bound to the entire space by considering approximation errors.

Fix an arbitrary hypothesis $r \in \mathcal{H}_r$. Let $r'$ be an element from an $\epsilon$-net of $\mathcal{H}_r$ such that $\|r - r'\|_\infty \leq \epsilon$. Define the function

$$g_r(Z) = |r(Z) - r_t^*(Z)| \tag{35}$$

We decompose the population-empirical discrepancy as follows:

$$
\left| \mathbb{E}_{Z \sim \mathcal{D}_0}[g_r(Z)] - \frac{1}{n} \sum_{i=1}^{n} g_r(Z_i) \right|
$$

$$
\leq \underbrace{|\mathbb{E}_{Z \sim \mathcal{D}_0}[g_r(Z)] - \mathbb{E}_{Z \sim \mathcal{D}_0}[g_{r'}(Z)]|}_{\text{Term}(a)} + \underbrace{\left| \mathbb{E}_{Z \sim \mathcal{D}_0}[g_{r'}(Z)] - \frac{1}{n} \sum_{i=1}^{n} g_{r'}(Z_i) \right|}_{\text{Term}(b)}
\tag{36}
$$

$$
+ \underbrace{\left| \frac{1}{n} \sum_{i=1}^{n} g_{r'}(Z_i) - \frac{1}{n} \sum_{i=1}^{n} g_r(Z_i) \right|}_{\text{Term}(c)}.
$$

We analyze each term separately:

- **Term (a)** This term captures the approximation error due to using $r'$ instead of $r$. By the properties of the $\epsilon$-net, we have:

$$
|\mathbb{E}_{Z \sim \mathcal{D}_0}[g_r(Z)] - \mathbb{E}_{Z \sim \mathcal{D}_0}[g_{r'}(Z)]| = |\mathbb{E}_{Z \sim \mathcal{D}_0}[|r(Z) - r_t^*(Z)| - |r'(Z) - r_t^*(Z)|]|
$$
$$
\leq \mathbb{E}_{Z \sim \mathcal{D}_0}[|r(Z) - r'(Z)|] \leq \epsilon.
\tag{37}
$$

- **Term (b)** This term represents the estimation error for the approximating model $r'$.

  For a fixed $r'$, the random variables $g_{r'}(Z_i)$ are i.i.d. and bounded by $[0, \beta']$. By applying Hoeffding's inequality, we have with probability at least $1 - \delta$:

$$
\left| \mathbb{E}_{Z \sim \mathcal{D}_0}[g_{r'}(Z)] - \frac{1}{n} \sum_{i=1}^{n} g_{r'}(Z_i) \right| \leq \beta' \sqrt{\frac{\log(2/\delta)}{2n}}
\tag{38}
$$

  Applying the union bound over all elements in the $\epsilon$-net $\mathcal{N}(\mathcal{H}_r, \epsilon, \|\cdot\|_\infty)$, we obtain:

$$
\sup_{r' \in \mathcal{N}(\mathcal{H}_r, \epsilon, \|\cdot\|_\infty)} \left| \mathbb{E}_{Z \sim \mathcal{D}_0}[g_{r'}(Z)] - \frac{1}{n} \sum_{i=1}^{n} g_{r'}(Z_i) \right| \leq \beta' \sqrt{\frac{\log\left(\frac{2\mathcal{N}(\mathcal{H}_r, \epsilon, \|\cdot\|_\infty)}{\delta}\right)}{2n}}
\tag{39}
$$

  with probability at least $1 - \delta$.

- **Term (c)** This term is also bounded by $\epsilon$, similar to term (a), due to the $\epsilon$-closeness of $r$ and $r'$.

Combining the bounds for Terms (a), (b), and (c), we obtain that, with probability at least $1 - \delta$,

$$
\sup_{r \in \mathcal{H}_r} \left| \mathbb{E}_{Z \sim \mathcal{D}_0}[g_r(Z)] - \frac{1}{n} \sum_{i=1}^{n} g_r(Z_i) \right| \leq 2\epsilon + \beta' \sqrt{\frac{\log\left(\frac{2\mathcal{N}(\mathcal{H}_r, \epsilon, \|\cdot\|_\infty)}{\delta}\right)}{2n}}.
\tag{40}
$$

By summing over all $t \in [T]$ and setting $\epsilon = \frac{1}{T}$ and , the final aggregated bound for the entire sequence of estimators under consideration becomes:

$$
\left| \frac{1}{T} \sum_{t=1}^{T} \mathbb{E}_{Z \sim \mathcal{D}_0}[g_{\hat{r}_t}(Z)] - \frac{1}{T} \sum_{t=1}^{T} \frac{1}{n} \sum_{i=1}^{n} g_{\hat{r}_t}(Z_i) \right| \leq \frac{2}{T} + \beta' \sqrt{\frac{\log\left(\frac{2\mathcal{N}(\mathcal{H}_r, 1/T, \|\cdot\|_\infty)}{\delta}\right)}{2n}}.
\tag{41}
$$

Next, we focus on bounding the covering number of the hypothesis space $\mathcal{H}_r$. Since we parameterize the density ratio functions in $\mathcal{H}_r$ using a parametric model $h(\mathbf{x}, \theta)$ with parameters $\theta$ in a bounded set $\Theta$, we can relate the covering number of $\mathcal{H}_r$ to that of $\Theta$.

Let $\theta, \theta' \in \Theta$ be the parameters corresponding to the two density ratio functions $r, r' \in \mathcal{H}_\theta$. We can show that for any $\|\theta - \theta'\|_2 \leq \epsilon$, the following inequality holds:

$$\|r - r'\|_\infty = \max_{Z \in \mathcal{Z}} |r(Z, \theta) - r(Z, \theta')| \leq G_h \|\theta - \theta'\|_2,$$

where $G_h = \max_{Z \in \mathcal{Z}, \theta \in \Theta} \|\nabla h(Z, \theta)\|_2$ is the Lipschitz continuity constant of $h$.

As a result, we can bound the covering number of $\mathcal{H}_r$ in terms of $\|\cdot\|_\infty$ by the covering number of $\Theta$ in terms of $\|\cdot\|_2$. Specifically, we have:

$$\mathcal{N}(\mathcal{H}_r, 1/T, \|\cdot\|_\infty) \leq \mathcal{N}(\Theta, 1/(G_h T), \|\cdot\|_2).$$

Given that the parameter space $\Theta$ is essentially a $L_2$-ball with radius $S$, its covering number is bounded by $(3S/\epsilon)^d$. Therefore, choosing $\epsilon = 1/(G_h T)$, we obtain:

$$\mathcal{N}(\Theta, 1/(G_h T), \|\cdot\|_2) \leq (3SG_h T)^d.$$

Combining these results, we conclude:

$$\mathcal{N}(\mathcal{H}_\theta, 1/T, \|\cdot\|_\infty) \leq (3SG_h T)^d.$$

Substituting this bound into our earlier expression, we complete the proof.

$\square$

**Lemma 8.** *Suppose Assumption 1, 2 and 3 hold, then for any sequence of density ratio estimators $\{\hat{r}_t\}_{t=1}^T$ and corresponding true density ratios $\{r_t^*\}_{t=1}^T$, the following bound holds with probability at least $1 - \delta$:*

$$\frac{1}{T} \sum_{t=1}^T |\hat{r}_t(Z_{n+t}) - r_t^*(Z_{n+t})| \leq \tilde{\mathcal{O}}\left(n^{-\frac{1}{2}} + \max\{T^{-\frac{1}{3}} V_T^{\frac{1}{3}}, T^{-\frac{1}{2}}\}\right) \tag{42}$$

*Proof.* The proof is straightforward by combining the results from Lemma 6, Lemma 7, and Corollary 2.

$\square$

**Lemma 9.** *Let $V : \mathcal{Z} \to \mathbb{R}$ be any measurable score and define the population CDF under $\mathcal{D}_t$ by*

$$\Psi_t(v) := \mathbb{P}_{Z \sim \mathcal{D}_t}(V(Z) \leq v).$$

*Define the self-normalized weighted empirical CDF that* includes *the test point:*

$$\Psi_t'(v) := \sum_{i \in [n] \cup \{n+t\}} w_{t,i}^* \mathbb{1}\{V(Z_i) \leq v\}, \qquad w_{t,i}^* := \frac{r_t^*(Z_i)}{\sum_{j \in [n] \cup \{n+t\}} r_t^*(Z_j)}.$$

*Then for any $\delta \in (0, 1)$, with probability at least $1 - \delta$, simultaneously for all $t \in [T]$,*

$$\sup_{v \in \mathbb{R}} |\Psi_t'(v) - \Psi_t(v)| \leq 8B\sqrt{\frac{\log\left(\frac{4T}{\delta}\right)}{n}} + \frac{2B}{n}, \tag{43}$$

*provided $n \geq 2B^2 \log\left(\frac{2T}{\delta}\right)$.*

*Proof.* Fix $t \in [T]$. Define

$$g_v(z) := \mathbb{1}\{V(z) \leq v\}.$$

Let

$$S_n := \sum_{i=1}^n r_t^*(Z_i), \qquad R_t := r_t^*(Z_{n+t}) \in [0, B].$$

Define the standard self-normalized importance sampling CDF based only on $\mathcal{D}_0$ samples:

$$\widetilde{\Psi}_{t,n}(v) := \frac{\sum_{i=1}^n r_t^*(Z_i) g_v(Z_i)}{S_n}.$$

We begin by decomposing the error:

$$\sup_{v \in \mathbb{R}} |\Psi_t'(v) - \Psi_t(v)| \le \underbrace{\sup_{v \in \mathbb{R}} \left| \Psi_t'(v) - \widetilde{\Psi}_{t,n}(v) \right|}_{\text{Term (a)}} + \underbrace{\sup_{v \in \mathbb{R}} \left| \widetilde{\Psi}_{t,n}(v) - \Psi_t(v) \right|}_{\text{Term (b)}}.$$

We bound the two terms separately.

**Bounding Term (a).** This term captures the perturbation effect of including the test point $Z_{n+t}$ in the self-normalized IS CDF. We first express $\Psi_t'(v)$ in terms of $\widetilde{\Psi}_{t,n}(v)$ and $g_v(Z_{n+t})$. Specifically, using $S_n + R_t = \sum_{i \in [n] \cup \{n+t\}} r_t^*(Z_i)$, we can write

$$\Psi_t'(v) = \frac{S_n}{S_n + R_t} \widetilde{\Psi}_{t,n}(v) + \frac{R_t}{S_n + R_t} g_v(Z_{n+t}).$$

Hence

$$|\Psi_t'(v) - \widetilde{\Psi}_{t,n}(v)| = \frac{R_t}{S_n + R_t} |g_v(Z_{n+t}) - \widetilde{\Psi}_{t,n}(v)| \le \frac{R_t}{S_n + R_t} \le \frac{B}{S_n},$$

and therefore

$$\sup_v |\Psi_t'(v) - \widetilde{\Psi}_{t,n}(v)| \le \frac{B}{S_n}. \tag{44}$$

Then, we lower bound $S_n$ using concentration. Note that $0 \le r_t^*(Z_i) \le B$ and $\mathbb{E}_{\mathcal{D}_0}[r_t^*(Z_i)] = 1$, Hoeffding's inequality gives

$$\mathbb{P}\left( \frac{S_n}{n} \le \frac{1}{2} \right) = \mathbb{P}\left( \frac{1}{n} \sum_{i=1}^n r_t^*(Z_i) - 1 \le -\frac{1}{2} \right) \le \exp\left( -\frac{n}{2B^2} \right).$$

Thus, if $n \ge 2B^2 \log\left( \frac{2T}{\delta} \right)$, then by a union bound over $t \in [T]$, with probability at least $1 - \delta/2$ we have $S_n \ge n/2$ simultaneously for all $t$. On this event,

$$\sup_v |\Psi_t'(v) - \widetilde{\Psi}_{t,n}(v)| \le \frac{B}{S_n} \le \frac{2B}{n}. \tag{45}$$

**Bounding Term (b)** This term captures the standard self-normalized importance sampling CDF approximation error based on $n$ samples from $\mathcal{D}_0$. Recall that

$$\Psi_t(v) = \mathbb{E}_{\mathcal{D}_t}[g_v(Z)] = \mathbb{E}_{\mathcal{D}_0}[r_t^*(Z) g_v(Z)] \quad \text{and} \quad \mathbb{E}_{\mathcal{D}_0}[r_t^*(Z)] = 1.$$

Let

$$A_n(v) := \frac{1}{n} \sum_{i=1}^n r_t^*(Z_i) g_v(Z_i), \qquad B_n := \frac{1}{n} \sum_{i=1}^n r_t^*(Z_i) = \frac{S_n}{n},$$

so that $\widetilde{\Psi}_{t,n}(v) = A_n(v)/B_n$. For any $v$,

$$\left| \frac{A_n(v)}{B_n} - \Psi_t(v) \right| \le \frac{|A_n(v) - \Psi_t(v)|}{B_n} + \frac{|\Psi_t(v)| |B_n - 1|}{B_n} \le \frac{|A_n(v) - \Psi_t(v)|}{B_n} + \frac{|B_n - 1|}{B_n},$$

since $\Psi_t(v) \in [0, 1]$. Taking the supremum over $v$ gives

$$\sup_v |\widetilde{\Psi}_{t,n}(v) - \Psi_t(v)| \le \frac{\sup_v |A_n(v) - \Psi_t(v)|}{B_n} + \frac{|B_n - 1|}{B_n}. \tag{46}$$

Because $0 \le r_t^*(Z_i) g_v(Z_i) \le B$ and the class $\{g_v : v \in \mathbb{R}\}$ has VC dimension 1, standard VC maximal inequalities (e.g., uniform Hoeffding bounds for VC classes) imply that there exists a universal constant $c_0 > 0$ such that, with probability at least $1 - \delta'$,

$$\sup_v |A_n(v) - \Psi_t(v)| \le c_0 B \sqrt{\frac{\log(1/\delta')}{n}}.$$

In addition, Hoeffding gives with probability at least $1 - \delta'$,

$$|B_n - 1| \le B \sqrt{\frac{\log(2/\delta')}{2n}}.$$

Choose $\delta' = \delta/(2T)$ and union bound over $t \in [T]$ to make these hold simultaneously for all $t$ with probability at least $1 - \delta/2$.

On the event $B_n \geq 1/2$, combining with equation 46 gives

$$\sup_v |\widetilde{\Psi}_{t,n}(v) - \Psi_t(v)| \leq 2 \sup_v |A_n(v) - \Psi_t(v)| + 2|B_n - 1| \leq 8B\sqrt{\frac{\log\left(\frac{4T}{\delta}\right)}{n}}$$

after absorbing constants.

**Combining the bounds**  Combining the bounds for Term (a) and Term (b), and using the high-probability events from both steps, we proves equation 43. □

**Lemma 10.** *Suppose Assumptions 1, 2, 3, and 4 hold. Then with probability at least $1 - \delta$,*

$$\left| \frac{1}{T} \sum_{t=1}^T \mathbb{P}(\mathcal{E}_t) - \frac{1}{T} \sum_{t=1}^T \mathbb{P}(\hat{\mathcal{E}}_t \mid \mathcal{F}_{t-1}) \right| \leq \tilde{O}\left( n^{-1/2} + \max\left\{ T^{-1/3}V_T^{1/3}, T^{-1/2} \right\} \right). \tag{47}$$

*Proof.* Let $\Psi_t(v)$ denote the true cumulative distribution function (CDF) of $V(\mathbf{C}_{n+t}, \mathbf{W}_{n+t})$ and the estimated weighted CDF using the oracle density ratio estimator $\hat{r}_t$ as

$$\Psi'_t(v) = \sum_{i \in [n] \cup \{n+t\}} w_t^*(Z_i) \mathbb{1}[v_i \leq v],$$

where $w_t^*(Z_i) = \frac{r_t^*(Z_i)}{\sum_{j \in [n] \cup \{n+t\}} r_t^*(Z_j)}$ are the true weights derived from the true density ratio $r_t^*$, and $v_i$ denotes the value of $V(\mathbf{C}_i, \mathbf{W}_i)$.

Similarly, the estimated weighted CDF at time step $t$ is:

$$\hat{\Psi}_t(v) = \sum_{i \in [n] \cup \{n+t\}} \hat{w}_t(Z_i) \mathbb{1}[v_i \leq v],$$

where $\hat{w}_t(Z_i) = \frac{\hat{r}_t(Z_i)}{\sum_{j \in [n] \cup \{n+t\}} \hat{r}_t(Z_j)}$ are the estimated weights based on the estimated density ratio $\hat{r}_t$.

Due to the nested property, i.e., the size of $\hat{F}_t$ non-decreasing w.r.t $\hat{\tau}_t$, the following equation holds:

$$\mathcal{E}_t = \{V(\mathbf{C}_{n+t}, \mathbf{W}_{n+t}) > \tau_t\}, \quad \hat{\mathcal{E}}_t = \{V(\mathbf{C}_{n+t}, \mathbf{W}_{n+t}) > \hat{\tau}_t\},$$

where $\tau_t$ and $\hat{\tau}_t$ are $(1-\alpha)$-quantiles of $\Psi_t$ and $\hat{\Psi}_t$, respectively. Consequently, we have:

$$\mathbb{P}[\mathcal{E}_t] = 1 - \Psi_t(\tau_t), \quad \mathbb{P}[\hat{\mathcal{E}}_t \mid \mathcal{F}_{t-1}] = 1 - \Psi_t(\hat{\tau}_t).$$

Using these definitions, we can express the difference in average probabilities as:

$$\left| \frac{1}{T} \sum_{t=1}^T \mathbb{P}[\mathcal{E}_t] - \frac{1}{T} \sum_{t=1}^T \mathbb{P}[\hat{\mathcal{E}}_t \mid \mathcal{F}_{t-1}] \right| = \left| \frac{1}{T} \sum_{t=1}^T [(1 - \Psi_t(\tau_t)) - (1 - \Psi_t(\hat{\tau}_t))] \right|$$

$$= \left| \frac{1}{T} \sum_{t=1}^T [\Psi_t(\hat{\tau}_t) - \Psi_t(\tau_t)] \right|$$

$$= \left| \frac{1}{T} \sum_{t=1}^T \left[ \Psi_t(\hat{\tau}_t) - \Psi_t(\tau_t) + \hat{\Psi}_t(\hat{\tau}_t) - \hat{\Psi}_t(\hat{\tau}_t) \right] \right|$$

$$\leq \left| \frac{1}{T} \sum_{t=1}^T \left[ \Psi_t(\hat{\tau}_t) - \hat{\Psi}_t(\hat{\tau}_t) \right] \right| + \mathcal{O}(1/n)$$

$$= \left| \frac{1}{T} \sum_{t=1}^T \left[ \Psi_t(\hat{\tau}_t) - \Psi'_t(\hat{\tau}_t) + \Psi'_t(\hat{\tau}_t) - \hat{\Psi}_t(\hat{\tau}_t) \right] \right| + \mathcal{O}(1/n)$$

$$\leq \left| \frac{1}{T} \sum_{t=1}^T \left[ \Psi_t(\hat{\tau}_t) - \Psi'_t(\hat{\tau}_t) \right] \right| + \left| \frac{1}{T} \sum_{t=1}^T \left[ \Psi'_t(\hat{\tau}_t) - \hat{\Psi}_t(\hat{\tau}_t) \right] \right| + \mathcal{O}(1/n)$$

$$\leq \underbrace{\frac{1}{T} \sum_{t=1}^T \sup_v \left| \Psi_t(v) - \Psi'_t(v) \right|}_{\text{term (a)}} + \underbrace{\left| \frac{1}{T} \sum_{t=1}^T \left[ \Psi'_t(\hat{\tau}_t) - \hat{\Psi}_t(\hat{\tau}_t) \right] \right|}_{\text{term (b)}} + \mathcal{O}(1/n).$$

The additional $\mathcal{O}(1/n)$ term arises from the discretization error of the empirical weighted CDF $\hat{\Psi}_t$. Although the conformity scores are perturbed by independent continuous random noise to eliminate ties and ensure the uniqueness of the quantiles, the empirical weighted CDF remains a step function. Consequently, the quantile threshold $\hat{\tau}_t$ may not satisfy the equality

$$\hat{\Psi}_t(\hat{\tau}_t) = 1 - \alpha$$

exactly. Instead, the discrepancy is bounded by the maximum jump size of $\hat{\Psi}_t$, which is of order $\mathcal{O}(1/n)$ under the bounded density ratio assumption. Therefore,

$$\left| \hat{\Psi}_t(\hat{\tau}_t) - (1-\alpha) \right| = \left| \hat{\Psi}_t(\hat{\tau}_t) - \Psi_t(\tau_t) \right| \leq \mathcal{O}(1/n).$$

**Bounding term (a)** By Lemma 9, we have with probability at least $1 - \delta$:

$$\frac{1}{T} \sum_{t=1}^{T} \sup_v |\Psi_t(v) - \Psi_t'(v)| \leq 8\beta \sqrt{\frac{\log\left(\frac{4T}{\delta}\right)}{n}} + \frac{2\beta}{n} = \tilde{\mathcal{O}}\left(n^{-\frac{1}{2}}\right).$$

**Bounding term (b)** Expanding $\Psi_t'(\hat{\tau}_t)$ and $\hat{\Psi}_t(\hat{\tau}_t)$, we have:

$$\left| \frac{1}{T} \sum_{t=1}^{T} \Psi_t'(\hat{\tau}_t) - \hat{\Psi}_t(\hat{\tau}_t) \right| = \left| \frac{1}{T} \sum_{t=1}^{T} \sum_{i \in [n] \cup \{n+t\}} \mathbb{1}[v_i \leq \hat{\tau}_t] \left( w_t^*(Z_i) - \hat{w}_t(Z_i) \right) \right|.$$

Substituting $w_t^*(Z_i)$ and $\hat{w}_t(Z_i)$, we obtain:

$$\left| \frac{1}{T} \sum_{t=1}^{T} \sum_{i \in [n] \cup \{n+t\}} \mathbb{1}[v_i \leq \hat{\tau}_t] \left( \frac{r_t^*(Z_i)}{\sum_{j \in [n] \cup \{n+t\}} r_t^*(Z_j)} - \frac{\hat{r}_t(Z_i)}{\sum_{j \in [n] \cup \{n+t\}} \hat{r}_t(Z_j)} \right) \right|$$

$$\leq \left| \frac{1}{T} \sum_{t=1}^{T} \sum_{i \in [n] \cup \{n+t\}} \mathbb{1}[v_i \leq \hat{\tau}_t] \left( \frac{r_t^*(Z_i) \cdot \sum_{j \in [n] \cup \{n+t\}} \hat{r}_t(Z_j) - \hat{r}_t(Z_i) \sum_{j \in [n] \cup \{n+t\}} r_t^*(Z_j)}{\sum_{j \in [n] \cup \{n+t\}} r_t^*(Z_j) \cdot \sum_{j \in [n] \cup \{n+t\}} \hat{r}_t(Z_j)} \right) \right|.$$

Simplifying the numerator of the fraction, let:

$$\text{Term}_i = r_t^*(Z_i) \cdot \sum_{j \in [n] \cup \{n+t\}} \hat{r}_t(Z_j) - \hat{r}_t(Z_i) \cdot \sum_{j \in [n] \cup \{n+t\}} r_t^*(Z_j).$$

Expanding $\text{Term}_i$, we have:

$$\text{Term}_i = r_t^*(Z_i) \cdot \sum_{j \in [n] \cup \{n+t\}} (\hat{r}_t(Z_j) - r_t^*(Z_j)) + (r_t^*(Z_i) - \hat{r}_t(Z_i)) \cdot \sum_{j \in [n] \cup \{n+t\}} r_t^*(Z_j).$$

Summing over $i \in [n] \cup \{n+t\}$, we bound $|\text{Term}_i|$ as:

$$\sum_{i \in [n] \cup \{n+t\}} |\text{Term}_i| \leq \sum_{i \in [n] \cup \{n+t\}} r_t^*(Z_i) \cdot \sum_{j \in [n] \cup \{n+t\}} |\hat{r}_t(Z_j) - r_t^*(Z_j)|$$

$$+ \sum_{i \in [n] \cup \{n+t\}} |r_t^*(Z_i) - \hat{r}_t(Z_i)| \cdot \sum_{j \in [n] \cup \{n+t\}} r_t^*(Z_j)$$

$$= 2 \cdot \sum_{i \in [n] \cup \{n+t\}} r_t^*(Z_i) \cdot \sum_{j \in [n] \cup \{n+t\}} |\hat{r}_t(Z_j) - r_t^*(Z_j)|.$$

Combining terms and simplifying, we find:

$$
\begin{aligned}
&\left| \frac{1}{T} \sum_{t=1}^{T} \Psi'_t(\hat{\tau}_t) - \frac{1}{T} \sum_{t=1}^{T} \hat{\Psi}_t(\hat{\tau}_t) \right| \\
&= \left| \frac{1}{T} \sum_{t=1}^{T} \sum_{i \in [n] \cup \{n+t\}} \mathbb{1}[v_i \le \hat{\tau}_t] \left( \frac{\text{Term}_i}{\sum_{j \in [n] \cup \{n+t\}} r_t^*(Z_j) \cdot \sum_{j \in [n] \cup \{n+t\}} \hat{r}_t(Z_j)} \right) \right| \\
&\le \frac{1}{T} \sum_{t=1}^{T} \sum_{i \in [n] \cup \{n+t\}} \frac{|\text{Term}_i|}{\sum_{j \in [n] \cup \{n+t\}} r_t^*(Z_j) \cdot \sum_{j \in [n] \cup \{n+t\}} \hat{r}_t(Z_j)} \\
&\le \frac{1}{T} \sum_{t=1}^{T} \frac{2}{\sum_{j \in [n] \cup \{n+t\}} \hat{r}_t(Z_j)} \cdot \sum_{j \in [n] \cup \{n+t\}} |\hat{r}_t(Z_j) - r_t^*(Z_j)|
\end{aligned}
$$

Invoking Assumption 4 and taking $t_0 = 1$, we obtain

$$
\text{term (b)} \le \frac{2}{nl} \left[ \frac{1}{T} \sum_{t=1}^{T} \frac{1}{n} \sum_{j \in [n]} |\hat{r}_t(Z_j) - r_t^*(Z_j)| + \frac{1}{n} \cdot \frac{1}{T} \sum_{t=1}^{T} |\hat{r}_t(Z_{n+t}) - r_t^*(Z_{n+t})| \right].
$$

If instead $t_0 > 1$, splitting the time average into $t \le t_0$ and $t > t_0$ yields an additional additive term at most $t_0/T$ (using $|\Psi'_t(v) - \hat{\Psi}_t(v)| \le 1$), which is negligible and does not change the final rate.

Finally, bounding the terms using Corollary 2 and Lemma 8 and combining it with the bound on term (a), we conclude the proof. $\qquad\square$

**Theorem 1.** *Under the assumptions 1, 2, 3 and 4, with probability at least $1 - 2\delta$, the gap between the averaged hallucination rate over $T$ time steps and the target level $\alpha$ is bounded as*

$$
\left| \frac{1}{T} \sum_{t=1}^{T} \widehat{\text{err}}_t - \alpha \right| \le \tilde{\mathcal{O}} \left( \max \left\{ T^{-\frac{2}{3}} V_T^{\frac{2}{3}}, T^{-\frac{1}{2}} \right\} + n^{-\frac{1}{2}} \right) \tag{19}
$$

*when the parameter of the online ensemble is properly set. Here, $V_T = \sum_{t=2}^{T} \|\mathcal{D}_t(\mathbf{z}) - \mathcal{D}_{t-1}(\mathbf{z})\|_1$ measures the variation of input densities and the notation $\tilde{\mathcal{O}}$ hides logarithmic factors of $T$ and $1/\delta$.*

*Proof.* We begin with the following decomposition:

$$
\begin{aligned}
\left| \frac{1}{T} \sum_{t=1}^{T} \widehat{\text{err}}_t - \alpha \right| \le & \underbrace{\left| \frac{1}{T} \sum_{t=1}^{T} \widehat{\text{err}}_t - \frac{1}{T} \sum_{t=1}^{T} \mathbb{P}(\hat{\mathcal{E}}_t \mid \mathcal{F}_{t-1}) \right|}_{\text{(a) concentration error}} \\
&+ \underbrace{\left| \frac{1}{T} \sum_{t=1}^{T} \mathbb{P}(\hat{\mathcal{E}}_t \mid \mathcal{F}_{t-1}) - \frac{1}{T} \sum_{t=1}^{T} \mathbb{P}(\mathcal{E}_t) \right|}_{\text{(b) estimation bias}} \\
&+ \underbrace{\left| \frac{1}{T} \sum_{t=1}^{T} \mathbb{P}(\mathcal{E}_t) - \alpha \right|}_{\text{(c) oracle error}}.
\end{aligned} \tag{48}
$$

**Bounding term (a).** By Lemma 5, which establishes concentration around the conditional mean, we have

$$
\text{term (a)} \le \tilde{\mathcal{O}}(T^{-1/2})
$$

with probability at least $1 - \delta$.

**Bounding term (b).** By Lemma 10, which controls the discrepancy between the conditional error probability and its oracle counterpart, we obtain

$$\text{term (b)} \leq \tilde{\mathcal{O}}\Big(n^{-1/2} + \max\big\{T^{-1/3}V_T^{1/3}, T^{-1/2}\big\}\Big)$$

with probability at least $1 - \delta$.

**Bounding term (c).** By construction of the oracle filtering rule, $\mathbb{P}(\mathcal{E}_t) = \alpha$ for all $t$, and hence

$$\text{term (c)} = 0.$$

**Conclusion.** Combining the bounds on terms (a)–(c) and applying a union bound completes the proof. $\square$

## G   LIMITATIONS AND FUTURE WORK

CoFact's theoretical guarantees apply to the entire time horizon and may not hold for smaller time intervals. Developing methods that provide finer-grained factuality guarantees is an important direction for future work. Additionally, while CoFact focuses on ensuring the factuality of filtered claims, other response qualities, such as informativeness and diversity, could also be required in certain scenarios. Extending CoFact to incorporate these aspects presents another promising avenue for future research.

