# OpenReview forum: "CoFact: Conformal Factuality Guarantees for Language Models under Covariate Shift"
_ICLR.cc/2026/Conference — ICLR 2026 Poster_

### Official Review · Reviewer_C6P4 · 2025-10-29

**Soundness:** 3
**Presentation:** 2
**Contribution:** 2
**Rating:** 4
**Confidence:** 3

**Summary:**

The paper introduces CoFact, a novel conformal prediction framework designed to ensure factuality guarantees for large language models (LLMs) under dynamic, real-world distribution shifts. Specifically, CoFact incorporates online density ratio estimation (DRE), enabling adaptive reweighting of calibration data to align with the changing test distribution. CoFact is validated through a theoretical analysis and empirical experiments.

**Strengths:**

1. The paper addresses a critical limitation of existing conformal prediction methods by introducing online density ratio estimation, which allows for adaptation to dynamic and non-stationary distributions.

2. The authors provide a solid mathematical foundation for CoFact by establishing a theoretical upper bound on the hallucination rate under shifting distributions.

3. A new dataset, WildChat+, is proposed to evaluate the approach, featuring real-world user-generated prompts that effectively capture distribution shifts.

4. CoFact is rigorously evaluated across multiple experimental settings, demonstrating its robustness and effectiveness.

**Weaknesses:**

1. The writing in the paper is difficult to follow, as many key terms are introduced without proper explanation. For instance, the concept of calibration is presented without clarifying its meaning or role within the framework, making it harder for readers to grasp its significance.

2. The framework heavily depends on accurate online density ratio estimation, which can be computationally intensive and challenging to implement efficiently, particularly for high-dimensional or complex data distributions.

3. While the inclusion of WildChat+ is a notable contribution, the paper could benefit from exploring additional domain-specific real-world applications, such as legal, financial, or healthcare contexts, to further demonstrate its practical impact in high-stake tasks.

4. CoFact's emphasis on factuality overlooks other crucial response qualities, such as informativeness. For example, the framework might encourage the model to produce overly short or simplistic responses that sacrifice depth or detail in order to meet factuality requirements. This potential trade-off warrants further investigation.

5. The paper lacks an extensive discussion on distribution shifts, such as distinguishing between in-distribution and out-of-distribution user prompts. Providing examples or a detailed analysis of these distinctions would help clarify how the framework handles varying prompt distributions.

**Questions:**

Including case studies and error analysis can significantly enhance the clarity and impact of the paper.

1. Case studies can illustrate how the proposed method performs under distribution shifts, offering concrete examples of its effectiveness.

2. Meanwhile, an in-depth error analysis can help readers understand why and how CoFact outperforms previous approaches, shedding light on its strengths and limitations in handling challenging real-world scenarios.

---

> ### Author Response · Authors · 2025-11-20
>
> > W1: The writing in the paper is difficult to follow, as many key terms are introduced without proper explanation. For instance, the concept of calibration is presented without clarifying its meaning or role within the framework, making it harder for readers to grasp its significance.
>
> We appreciate the reviewer’s feedback regarding the lack of clarity surrounding the term "calibration" in the paper. Below, we provide a more precise explanation of calibration and its critical role within the CoFact framework.
>
> - Definition and Role of Calibration
> In the context of conformal prediction, **calibration** refers to the process of using a held-out dataset (commonly referred to as the calibration set) to estimate the distribution of training samples's conformity scores. These conformity scores are then leveraged to compute thresholds that ensure finite-sample coverage guarantees.
>
> - The Role of Calibration in CoFact
> Calibration is a fundamental component of the CoFact framework as it enables the method to adaptively estimate conformity thresholds while accounting for distributional shifts. Specifically, the calibration set serves two key purposes within CoFact:
>     1. **Estimating the density ratio between the training and test distributions**: The calibration set, along with the sequentially arriving test samples, is used to train the online density ratio estimator to provide the necessary weights for adjusting conformity scores under distributional shifts.
>     2. **Providing information about the distribution of conformity scores**: The calibration set is used to compute the empirical distribution of conformity scores under the training data distribution. These scores are then adjusted using the estimated density ratios to compute adaptive thresholds for test samples, ensuring that factuality is controlled under distributional shifts.
>
> We hope this clarification addresses your concern and makes the concept of calibration more accessible to readers. Please let us know if further elaboration or refinement is needed.
>
> > W2: The framework heavily depends on accurate online density ratio estimation, which can be computationally intensive and challenging to implement efficiently, particularly for high-dimensional or complex data distributions.
>
> Thank you for highlighting this important concern regarding the computational demands of CoFact. We acknowledge that ensuring computational efficiency is crucial for practical deployment, especially in scenarios involving high-dimensional or complex data distributions. To address this, we have carefully incorporated several design strategies aimed at mitigating these challenges in CoFact:
>
> - Small Number of Activated Experts and Parallel Updates:
>    - At any time step $t$, only $\log t$ experts are activated and need to be updated, according to the scheduling rule.
>    - The updates for each expert are independent, allowing for parallel computation.
>
> - Simplicity of Each Expert:
>    - Each expert is modeled using a linear model, which is computationally lightweight both for forward inference and backward updates. This simplicity ensures that the cost per expert remains low.
>
> - Efficient Update via Online Newton Step (ONS):
>    - Instead of directly computing the inverse of the Hessian matrix (which is computationally expensive), we use the ONS method. In ONS, the matrix obtained by the outer product of gradients is used as a surrogate for the Hessian matrix. This approximation greatly reduces the computational cost while maintaining the robustness of the updates.
>
> With these optimizations in place, the computational cost of the CoFact framework remains manageable. While maintaining multiple experts and performing updates introduces some overhead, the incorporation of logarithmic scaling, parallelization, and efficient matrix operations ensures that the framework can operate effectively even in challenging scenarios, such as high-dimensional or complex data distributions.

---

> > ### Author Response · Authors · 2025-11-20
> >
> > > W3: While the inclusion of WildChat+ is a notable contribution, the paper could benefit from exploring additional domain-specific real-world applications, such as legal, financial, or healthcare contexts, to further demonstrate its practical impact in high-stake tasks.
> >
> > Thank you for the constructive suggestion.
> >
> > We would like to clarify that while WildChat+ does not specifically target legal or financial domains, it already encompasses a wide variety of naturally occurring topics, including several high-stakes areas such as healthcare and politics. As such, the evaluation conducted on WildChat+ already demonstrates the practical impact of our method in real-world, high-stakes scenarios.
> >
> > Additionally, our experiments include evaluations on the MedQA dataset, which specifically represents a medically oriented, high-stakes task within the healthcare domain. This provides further evidence that our framework is effective in safety-critical contexts, reinforcing its applicability to real-world challenges.
> >
> > That said, we agree that extending our framework to broader application domains, such as legal or financial contexts, and conducting comprehensive evaluations in these areas would significantly enhance its practical relevance and versatility. We sincerely appreciate your thoughtful suggestion and will incorporate it into our future research endeavors.
> >
> > > W4: CoFact's emphasis on factuality overlooks other crucial response qualities, such as informativeness. For example, the framework might encourage the model to produce overly short or simplistic responses that sacrifice depth or detail in order to meet factuality requirements. This potential trade-off warrants further investigation
> >
> > In our paper, we already include the "claims retained" metric, which quantifies the proportion of sub-claims preserved after applying CoFact’s filtering. This metric provides a direct measure of how much content is retained in the responses after factuality-based adjustments. Across all settings and datasets, the results show that a substantial and reasonable fraction of claims is retained, indicating that CoFact does not indiscriminately shorten responses or sacrifice depth.
> >
> > Furthermore, we emphasize this point in the case study, where the filtered response example remains detailed and informative despite the removal of hallucinated claims. This demonstrates that CoFact can maintain informativeness while prioritizing factuality.
> >
> > While our method demonstrates the ability to balance factuality and informativeness to some extent, we acknowledge that explicitly integrating mechanisms to better balance or control these response qualities, alongside factuality, represents a promising avenue for future research.

---

> > > ### Author Response · Authors · 2025-11-20
> > >
> > > > W5: The paper lacks an extensive discussion on distribution shifts, such as distinguishing between in-distribution and out-of-distribution user prompts. Providing examples or a detailed analysis of these distinctions would help clarify how the framework handles varying prompt distributions.
> > >
> > > Thank you for the valuable suggestion. We will incorporate a more detailed discussion on distribution shifts in the revised paper. Here, we provide an expanded explanation below:
> > >
> > > ### Discussion on Distribution Shifts
> > > We address the presence of distribution shifts by analyzing the evolution of user prompt topics over time. As shown in Figure 2 of the paper, we plot the distribution of topics in user prompts, illustrating how the proportions of these topics change dynamically. In Figure 2, each line represents a distinct topic (denoted by a unique color), while the y-axis indicates the proportion of prompts belonging to each topic at different time steps. From the figure, it is evident that the distribution of user prompts shifts significantly over time, demonstrating the dynamic nature of real-world scenarios.
> > >
> > > To provide an intuitive understanding of these shifts, we include representative examples of prompts for several topics depicted in Figure 2:
> > >
> > > - Topic 1 (Mathematics):
> > >   - *"Shear force formula using shear area and resistance factor for plastic material"*
> > >   - *"What is the formula for the correlation coefficient?"*
> > >
> > > - Topic 2 (Health):
> > >   - *"What are the benefits of being left-footed in football?"*
> > >   - *"Write a research paper on racial and ethnic differences in health behaviors and preventive health services among prostate cancer survivors in the United States."*
> > >
> > > - Topic 3 (Music):
> > >   - *"EDM is the more popular term to refer to house music genres."*
> > >
> > > Roughly speaking, these topics correspond to mathematics, health, and music, respectively. Based on this, we can provide a clearer explanation of distribution shifts. For instance, at time step 0, most user prompts might be related to mathematics, whereas at time step 500, the majority of prompts may focus on health. This shift from mathematics to health over time exemplifies how the distribution of user prompts evolves in real-world scenarios.
> > >
> > > ### Distinction Between In-Distribution and Out-of-Distribution Prompts
> > > We define **in-distribution prompts** as those sampled from the same distribution as the training data. In contrast, **out-of-distribution prompts** originate from a test distribution that may diverge significantly from the training distribution. Such distribution divergence poses challenges for traditional conformal prediction methods, which typically assume that the test distribution matches the training distribution. This mismatch can lead to failures in maintaining factuality guarantees.

---

> ### Author Response · Authors · 2025-11-20
>
> > Q1: Case studies can illustrate how the proposed method performs under distribution shifts, offering concrete examples of its effectiveness.**
>
> We present several case studies on the WildChat+ dataset below to demonstrate how CoFact performs under distribution shifts. Each row in the table corresponds to a prompt-response pair, along with the weight assigned by CoFact. These weights represent the estimated density ratios under distribution shifts.
>
> It is important to note that the weights are normalized, with the standard weight corresponding to no reweighting being 0.0008.
>
>
> |                             Prompt                             |                                     Response                                      | Weights |     |
> | :------------------------------------------------------------: | :-------------------------------------------------------------------------------: | :-----: | --- |
> |          What type of brain damage could be probable           |         There are various forms of brain damage that could potentially...         | 0.0002  |     |
> |                            food web                            |            The food web of the lotus flower ecosystem is a complex ...            | 0.0003  |     |
> | past perfect differ from present perfect Include in a timeline |           The past perfect tense is used to describe an action that ...           | 0.0026  |     |
> |              use HOME in ENV directive dockerfile              | To use the `HOME` environment variable in a Dockerfile's `ENV` directive, you ... | 0.0316  |     |
>
>
>
> From the table, we observe that CoFact assigns significantly different weights to different prompt-response pairs. For the first two pairs, the weights are lower than the standard weight of 0.0008, indicating that CoFact considers these pairs to be "more likely" sampled from the training distribution. Conversely, for the last two pairs, the weights are substantially higher than the standard weight, suggesting that CoFact identifies these pairs as "more likely" originating from the test distribution. By leveraging these adaptive weights, CoFact effectively addresses distribution shifts in real-world scenarios, ensuring the factuality of its responses.
>
> Since the ground truth density ratios are not available for the WildChat+ dataset, we further provide quantitative analysis on the MedQA and Wiki datasets, where distribution shifts are simulated, and the ground truth density ratios are known. Specifically, we compute the cosine similarity between CoFact's estimated density ratios and the ground truth density ratios at each time step. The results for the MedQA and Wiki datasets are presented below:
>
>
>
> |                | Lin | Squ | Sin| Ber|
> | :------------: | :---------: | :-------: | :---------: | :------------: |
> | time_step=500  |    0.750    |   0.812   |    0.646    |     0.807      |
> | time_step=1000 |    0.689    |   0.752   |    0.627    |     0.657      |
> | time_step=1500 |    0.697    |   0.624   |    0.678    |     0.663      |
> | time_step=2000 |    0.663    |   0.869   |    0.704    |     0.871      |
>
> |                | Lin | Squ | Sin| Ber|
> | :------------: | :---------: | :-------: | :---------: | :------------: |
> | time_step=500  |    0.985    |   0.988   |    0.973    |     0.988      |
> | time_step=1000 |    0.979    |   0.988   |    0.987    |     0.988      |
> | time_step=1500 |    0.976    |   0.972   |    0.975    |     0.972      |
> | time_step=2000 |    0.974    |   0.988   |    0.981    |     0.974      |
>
> From these results, we observe that CoFact achieves high cosine similarity with the ground truth density ratios under various distribution shifts. This demonstrates CoFact's ability to accurately estimate density ratios, further validating its effectiveness in handling distribution shifts across diverse real-world scenarios.

---

> ### Author Response · Authors · 2025-11-20
>
> > Q2: Meanwhile, an in-depth error analysis can help readers understand why and how CoFact outperforms previous approaches, shedding light on its strengths and limitations in handling challenging real-world scenarios.
>
> To demonstrate how CoFact effectively addresses the challenges of distribution shifts in real-world scenarios and outperforms previous methods, we present several cases where CoFact successfully identifies and filters out hallucinated claims from responses, while the standard conformal prediction method fails. The table structure is consistent with that in the previous answer, and the standard weight remains 0.0008.
>
> |                                         Prompt                                         |                           Response                           | Weights |     |
> | :------------------------------------------------------------------------------------: | :----------------------------------------------------------: | :-----: | --- |
> |                                   Types of Asbestos                                    |  There are different types of asbestos, each with its own distinct characteristics...   |0.0320  |     |
> | Explore the value of Chinese culture to the world in Cultural diversity and traditions | Chinese culture holds tremendous value in promoting cultural... | 0.0269  |     |
> |                                 what was canadas role                                  |  Canada played a significant role in the Battle of Hill 70...  | 0.0297  |     |
>
> From the table, we observe that CoFact's key advantage over the standard conformal prediction method lies in its ability to adaptively assign appropriate weights to different prompt-response pairs. Specifically, CoFact assigns significantly higher weights compared to the fixed standard weight of 0.0008, indicating that, at the time steps when these pairs arrive, they are more likely to be sampled from the test distribution rather than the training distribution.
>
> This adaptive weighting mechanism plays a critical role. Higher weights correspond to larger thresholds, which enforce stricter filtering to ensure factuality. Consequently, if the standard conformal prediction method with its fixed weight of 0.0008 were used, it would fail to effectively filter out hallucinated claims in these pairs. In contrast, CoFact dynamically adjusts to distribution shifts by leveraging adaptive weights, thereby ensuring both the factuality of responses and the preservation of reasonable claims.

---

### Official Review · Reviewer_H5QG · 2025-10-30

**Soundness:** 3
**Presentation:** 3
**Contribution:** 3
**Rating:** 8
**Confidence:** 3

**Summary:**

This paper addresses the challenge of maintaining the factuality of large language model (LLM) outputs in real-world scenarios where the distribution of user prompts evolves. The authors correctly point out that existing conformal prediction (CP) methods for providing factuality guarantees fall short in this setting. This is because they depend on the exchangeability assumption, which holds when the calibration and test data come from the same distribution. However, this assumption no longer holds when the prompt distribution changes.

To address this limitation, the paper proposes CoFact. This is a new conformal prediction framework designed to maintain factual reliability under distribution shift. The key idea behind CoFact is to employ online density ratio estimation to adaptively reweight the static calibration data. This way, it aligns with the evolving test distribution at each time step. This enables the computation of an adaptive conformal threshold that effectively filters out hallucinated claims. Moreover, the system no longer relies on the exchangeability assumption or the unrealistic requirement of having ground-truth labels for test instances.

**Strengths:**

This paper addresses the problem of LLM factuality guarantees under the realistic condition of distribution shift. Moreover, the experimental validation is comprehensive. A key strength is the inclusion of the new challenging real-world benchmark (WildChat+). The results show that CoFact outperforms baselines and achieves its stated goal.

**Weaknesses:**

The CoFact methodology relies on an online ensemble framework where each expert is updated using an Online Newton Step (ONS). I wonder if this method is computationally more expensive. If it is more costly, this is a potential limitation for real-world, low-latency deployment.

**Questions:**

The CoFact methodology relies on an online ensemble framework where each expert is updated using an Online Newton Step (ONS). I wonder if this method is computationally more expensive. If it is more costly, this is a potential limitation for real-world, low-latency deployment.

---

> ### Author Response · Authors · 2025-11-20
>
> > W1 and Q1: The CoFact methodology relies on an online ensemble framework ... (The computational efficiency of CoFact)
>
> Thank you for raising this important question regarding the computational cost and its implications for real-world deployment. We recognize that computational efficiency is a critical consideration for practical applications, and we have adopted several design choices in CoFact to address this concern:
>
> - Small Number of Activated Experts and Parallel Updates:
>    - At any time step $ t $, only $ \log t $ experts are activated and need to be updated, according to the scheduling rule.
>    - The updates for each expert are independent, allowing for parallel computation.
>
> - Simplicity of Each Expert:
>    - Each expert is modeled using a linear model, which is computationally lightweight both for forward inference and backward updates. This simplicity ensures that the cost per expert remains low.
>
> - Efficient Update via Online Newton Step (ONS):
>    - Instead of directly computing the inverse of the Hessian matrix (which is computationally expensive), we use the ONS method. In ONS, the matrix obtained by the outer product of gradients is used as a surrogate for the Hessian matrix. This approximation greatly reduces the computational cost while maintaining the robustness of the updates.
>
> With these optimizations, the computational cost of the CoFact methodology is kept manageable, even in online settings with real-time requirements. While there is some overhead associated with maintaining multiple experts and performing updates, the combination of logarithmic scaling, parallelization, and efficient matrix operations ensures that the method remains practical for low-latency applications.

---

### Official Review · Reviewer_XA7a · 2025-10-31

**Soundness:** 3
**Presentation:** 3
**Contribution:** 2
**Rating:** 4
**Confidence:** 3

**Summary:**

To theoretically control the rate of hallucination of large language models (LLMs) for their trustworthy use on safety-critical applications under dynamic environments where the test distribution may be different from the calibration data for training, authors propose the online confomal prediction method using an online density estimation technique.

Specifically, they assume that the test data distribution continuously changes over time, not in a too extreme manner (Assumption 1).

Besides, for the definition of hallucination of the generated answer on a given prompt, following Mohri et al. (2024), authors define the term as whether a filtered answer contain any hallucinated facts.

In terms of the technical contribution when compared to existing works on online conformal prediction under the distribution shift scenario, authors assume more challenging problem set-ups.

(1) Specifically, the authors assume continuous distribution shift scenario on the test data distribution, unlike a single distribution shift scenario (Tibshirani et al., 2019).
(2) Furthermore, they assume the correctness labels on the test data (i.e., whether a hallucinated sub-claim is contained is the filtered sub-claims) remain unrevealed even after the online evaluation.

This is a more challenging scenario compared to existing online conformal prediction methods (Gibbs and Candes, 2021; Gibbs and Candes, 2024; Areces et al., 2025) in which correct labels are revealed, enabling them to be used for training in subsequent time steps.

In addition to the theoretical guarantee on the upper bound of the gap between the target coverage level and the average hallucination rate, empirical results on existing and newly proposed benchmarks show that CoFact controls the hallucination rate to the desired degree.

**Strengths:**

- It is a well-written paper which is easy-to-follow.

- They tackle the online conformal prediction problem for the theoretical control of hallucination of LLMs for its trustworthy use in safety-critical applications, which is one of primary interests from LLM users these days.
Specifically, they propose the method which is valid under the more challenging set-up compared to existing works, which most resembles the dynamic real world problems.

**Weaknesses:**

[Weakness 1] Generalizability to online batch setting: I have understood that you are assuming a problem set-up, where single sample is provided for each time step for simplicity (Line 185-186). Then, how the Eq. (8) would look like if you consider a general setting, where a batch of samples may be provided for each time step? If not, the proposed threshold estimator may not be used in an online batch learning set-up.

[Weakness 2] While existing online conformal prediction methods require additional information in terms of training which is not possible in the current problem set-up, it would be more informative to provide results of these methods equipped with the necessary information for training, since baselines in the Experiment section all assume i.i.d. data generating process under the batch learning set-up. As an concrete example, you may consider a problem set-up with (1) a single distribution shift scenario as Tibshirani et al. (2019) and (2) an accessibility to the ground truth labels in an online manner. While existing methods are expected to show more superior performance comapred to CoFact since they utilize additional information in terms of online threshold selection, I think it would be much more informative than just comparing with baselines assuming i.i.d. data generating process under the batch learning set-up.

**Questions:**

[Question 1] Shouldn't the "T" in Eq. (14) be substituted to "t"? Additionally, the term \theta_t^\ast is not formally defined.

[Question 2] How the Eq. (8) would look like if you consider a general setting, where a batch of samples may be provided for each time step? If you can propose one, does it also enjoy the theoretical guarantee that Eq. (8) has? Please refer to [Weakness 1] in Weaknesses Section for detail.

[Question 3] Could you compare CoFact with existing online conformal prediction methods under the scenario where existing methods assume? Please refer to [Weakness 2] in Weaknesses Section for detail.

[Queston 4]  The followings are typos to be addressed.

(Line 35) Despite of => Despite

(Line 77) CoFact bypass => CoFact bypasses

(Line 106-107) The following expression seems awkward to me: "... that transforms the outputs of a black-box predictor into prediction sets..."

(Line 115-118) \alpha => \beta or \beta => \alpha

(Line 314-315) \leq => \geq

(Line 269, Line 328) The same divergence function \psi is defined differently.

---

> ### Author Response · Authors · 2025-11-20
>
> > W1 and Q2: Generalizability to online batch setting ...
>
> We apologize for the confusion caused by the unclear description in the original paper.
>
> Let us explicitly address the question by considering the scenario at time step $t$, where $n_t$ test samples $\{Z_{t,1}, Z_{t,2}, \ldots, Z_{t,n_t}\}$ arrive. Equation (8) then becomes:
>
> $$
>     \tau_{t,k} = \mathrm{Quantile}(1 - \alpha; \sum_{i=1}^n w^\ast_{t, k}(Z_i)\delta_{V_i} + w^\ast_{t, k}(Z_{t,k})\delta_{\infty}), \quad \text{for all} \quad k \in [n_t],
> $$
>
> where
>
> $$
>     w_{t, k}^\ast(Z) = \frac{r_t^\ast(Z)}{\sum_{i=1}^n r_t^\ast(Z_i) + r_t^\ast(Z_{t, k})}.
> $$
>
> In the original paper, we assumed $n_t = 1$ for simplicity of notation, dropping the subscript $k$. However, this assumption was made solely for notational convenience and does not affect the theoretical guarantees of our method.
>
> In fact, CoFact calculate the threshold and determine whether to retain sub-claims for each arriving sample independently.
>
> Furthermore, a batch of samples can be interpreted as a specific case of sequentially arriving individual samples, where the samples within the batch are treated as if they arrive one by one from the same distribution at a single time step $t$. Therefore, the theoretical guarantees established in our paper remain valid in the online batch learning setting.
>
>
> ### Experiments on Online Batch Learning
> To further demonstrate the applicability of our method to the online batch learning setup, we conducted experiments with different batch sizes (\{1, 5, 10, 20\}) on the MedQA and Wiki datasets under four shift types. The results are summarized below:
>
>
> |         |     Lin     |                 |     Squ     |                 |     Sin     |                 |     Ber     |                 |
> | :-------------: | :---------: | :-------------: | :---------: | :-------------: | :---------: | :-------------: | :------------: | :-------------: |
> |                 | Factuality  | Claims Retained | Factuality  | Claims Retained | Factuality  | Claims Retained |   Factuality   | Claims Retained |
> | per_step_num=5  | 0.893±0.024 |   0.723±0.020   | 0.890±0.025 |   0.722±0.027   | 0.887±0.029 |   0.726±0.023   |  0.896±0.014   |   0.712±0.030   |
> | per_step_num=10 | 0.897±0.024 |   0.703±0.028   | 0.895±0.025 |   0.701±0.028   | 0.897±0.025 |   0.709±0.027   |  0.902±0.012   |   0.696±0.027   |
> | per_step_num=20 | 0.899±0.024 |   0.696±0.031   | 0.900±0.025 |   0.693±0.032   | 0.899±0.025 |   0.698±0.030   |  0.904±0.011   |   0.684±0.028   |
>
> |         |     Lin     |                 |     Squ     |                 |     Sin     |                 |     Ber     |                 |
> | :-------------: | :---------: | :-------------: | :---------: | :-------------: | :---------: | :-------------: | :------------: | :-------------: |
> |                 | Factuality  | Claims Retained | Factuality  | Claims Retained | Factuality  | Claims Retained |   Factuality   | Claims Retained |
> | per_step_num=5  | 0.900±0.011 |   0.750±0.008   | 0.899±0.010 |   0.750±0.010   | 0.901±0.011 |   0.747±0.008   |  0.893±0.016   |   0.749±0.011   |
> | per_step_num=10 | 0.903±0.009 |   0.748±0.006   | 0.905±0.009 |   0.749±0.006   | 0.901±0.008 |   0.748±0.007   |  0.897±0.008   |   0.747±0.011   |
> | per_step_num=20 | 0.899±0.007 |   0.748±0.006   | 0.901±0.006 |   0.747±0.006   | 0.900±0.007 |   0.748±0.007   |  0.900±0.013   |   0.751±0.011   |
>
>
>
> From the results, we observe that our method consistently achieves reasonable performance in terms of Factuality and Claims Retained when handling arriving samples in batches. This further validates that our method is well-suited for the online batch learning setup.

---

> ### Author Response · Authors · 2025-11-20
>
> > W2 and Q3: Comparison with methods for conformal prediction under covariate shift and online conformal prediction.
>
> Thank you for your valuable insights regarding the need for a more comprehensive evaluation of our method. In response to your suggestions, we conducted additional experiments incorporating two new baselines: one tailored for conformal prediction under covariate shift [1], and another designed for online conformal prediction [2]. Below, we summarize experimental setup and results for both baselines.
>
> ### Comparison with Method in [1] (Covariate Shift Handling)
>
> The method in [1] requires access to all test samples to train a fixed density ratio estimator between the training and test distributions. For comprehensive evaluation, we implemented two variants:
> - **Total:** We use all test samples to estimate the density ratio and filter potentially hallucinated sub-claims
> - **Partial:** We use only the samples observed during the first 500 time steps to estimate the density ratio and filter potentially hallucinated sub-claims for all time steps.
>
> It is important to note that both variants are impractical in our online setting, as they require access to either all test samples or a large portion of them in advance, which violates the constraints of online learning.
>
> We compare these two variants with CoFact on the MedQA and Wiki datasets across five types of distributional shifts: four defined in our paper (LinearShift, BernoulliShift, SquareShift, and SineShift) and a fixed shift (Fix), where the test distribution remains constant but differs from the training distribution, similar to the setting adopted in [1]. The results are shown below:
>
>
> |         |     Lin     |                 |     Squ     |                 |     Sin     |                 |     Ber     |                 |  Fix         |                 |
> | ------- | ----------- | --------------- | ----------- | --------------- | ----------- | --------------- | ----------- | --------------- | ----------- | --------------- |
> |         | Factuality  | Claims Retained | Factuality  | Claims Retained | Factuality  | Claims Retained | Factuality  | Claims Retained | Factuality  | Claims Retained |
> | Total   | 0.849±0.025 | 0.856±0.015     | 0.849±0.021 | 0.861±0.012     | 0.848±0.022 | 0.858±0.013     | 0.855±0.015 | 0.853±0.011     | 0.859±0.011 | 0.862±0.012     |
> | Partial | 0.819±0.023 | 0.900±0.010     | 0.825±0.026 | 0.898±0.012     | 0.822±0.025 | 0.897±0.012     | 0.836±0.016 | 0.895±0.008     | 0.842±0.012 | 0.883±0.011     |
> | CoFact  | 0.895±0.026 | 0.715±0.031     | 0.897±0.022 | 0.718±0.030     | 0.894±0.018 | 0.715±0.031     | 0.900±0.019 | 0.713±0.027     | 0.904±0.014 | 0.713±0.027     |
>
>
> |         |     Lin     |                 |     Squ     |                 |     Sin     |                 |     Ber     |           | Fix |  |
> |---|---|---|---|---|---|---|---|---|---|---|
> |  | Factuality | Claims Retained | Factuality | Claims Retained | Factuality | Claims Retained | Factuality | Claims Retained | Factuality | Claims Retained |
> | Total | 0.895±0.010 | 0.768±0.004 | 0.895±0.009 | 0.768±0.005 | 0.895±0.008 | 0.768±0.004 | 0.887±0.012 | 0.773±0.004 | 0.895±0.009 | 0.771±0.004 |
> | Partial | 0.884±0.013 | 0.780±0.007 | 0.885±0.012 | 0.780±0.008 | 0.884±0.011 | 0.780±0.007 | 0.875±0.014 | 0.783±0.007 | 0.886±0.012 | 0.783±0.007 |
> | CoFact | 0.896±0.010 | 0.748±0.006 | 0.895±0.009 | 0.748±0.006 | 0.895±0.008 | 0.748±0.006 | 0.897±0.008 | 0.749±0.006 | 0.896±0.010 | 0.753±0.007 |
>
>
> From the results, we observe the following:
> 1. The **Total** variant performs better than the **Partial** variant, indicating that access to all test samples improves density ratio estimation and enables the construction of more accurate factuality control.
> 2. CoFact consistently outperforms both **Total** and **Partial** variants across nearly all shift types and datasets in terms of factuality control. Even under fixed shifts, CoFact achieves superior or comparable performance to the baselines, demonstrating its robustness in handling both stable and dynamic distributional shifts.

---

> ### Author Response · Authors · 2025-11-20
>
> ### Comparison with Method in [2] (Online Conformal Prediction)
>
> The method ACI proposed in [2] is designed for Online Conformal Prediction settings where the test distribution may shift over time. However, unlike our setting, this method requires access to true labels after making predictions at each time step. While this assumption is not feasible in our scenario, we implemented the method as a baseline for comparison.
>
> We tested two update strategies from [2]:
> - **ACI-Simple:** Defined by Equation (2) in the original paper.
> - **ACI-Momentum:** Defined by Equation (3) in the original paper.
>
> Additionally, we varied the step size parameter $\gamma$ among $\{0.0005, 0.001, 0.005, 0.01\}$ to observe its impact on performance.
>
> The results on the MedQA and Wiki datasets are shown below:
>
>
> |         |     Lin     |                 |     Squ     |                 |     Sin     |                 |     Ber     |                 |
> | :-----------------: | :---------: | :-------------: | :---------: | :-------------: | :---------: | :-------------: | :---------: | :-------------: |
> |                     | Factuality  | Claims Retained | Factuality  | Claims Retained | Factuality  | Claims Retained | Factuality  | Claims Retained |
> |  ACI-Simple-0.0005  | 0.882±0.006 |   0.747±0.035   | 0.884±0.003 |   0.723±0.055   | 0.885±0.002 |   0.736±0.056   | 0.884±0.001 |   0.751±0.038   |
> | ACI-Momentum-0.0005 | 0.882±0.007 |   0.747±0.035   | 0.885±0.003 |   0.723±0.055   | 0.885±0.003 |   0.733±0.056   | 0.885±0.001 |   0.747±0.037   |
> |  ACI-Simple-0.001   | 0.909±0.003 |   0.635±0.051   | 0.912±0.003 |   0.617±0.073   | 0.912±0.002 |   0.608±0.077   | 0.911±0.003 |   0.653±0.070   |
> | ACI-Momentum-0.001  | 0.910±0.004 |   0.633±0.051   | 0.913±0.004 |   0.612±0.073   | 0.912±0.002 |   0.606±0.078   | 0.913±0.003 |   0.639±0.069   |
> |  ACI-Simple-0.005   | 0.975±0.003 |   0.207±0.050   | 0.977±0.003 |   0.194±0.035   | 0.978±0.003 |   0.185±0.041   | 0.972±0.006 |   0.269±0.088   |
> | ACI-Momentum-0.005  | 0.976±0.005 |   0.206±0.063   | 0.980±0.004 |   0.153±0.043   | 0.980±0.002 |   0.167±0.052   | 0.977±0.006 |   0.227±0.099   |
> |   ACI-Simple-0.01   | 0.986±0.003 |   0.131±0.029   | 0.987±0.003 |   0.114±0.041   | 0.988±0.002 |   0.115±0.034   | 0.985±0.005 |   0.161±0.080   |
> |  ACI-Momentum-0.01  | 0.987±0.004 |   0.121±0.065   | 0.990±0.002 |   0.088±0.030   | 0.989±0.003 |   0.102±0.044   | 0.993±0.003 |   0.069±0.026   |
> |       CoFact        | 0.895±0.026 |   0.715±0.031   | 0.897±0.022 |   0.718±0.030   | 0.894±0.018 |   0.715±0.031   | 0.900±0.019 |   0.714±0.036   |
>
>
>
> |         |     Lin     |                 |     Squ     |                 |     Sin     |                 |     Ber     |                 |
> | :-----------------: | :---------: | :-------------: | :---------: | :-------------: | :---------: | :-------------: | :---------: | :-------------: |
> |                     | Factuality  | Claims Retained | Factuality  | Claims Retained | Factuality  | Claims Retained | Factuality  | Claims Retained |
> |  ACI-Simple-0.0005  | 0.894±0.004 |   0.766±0.004   | 0.895±0.005 |   0.765±0.005   | 0.894±0.004 |   0.764±0.003   | 0.892±0.006 |   0.760±0.011   |
> | ACI-Momentum-0.0005 | 0.894±0.004 |   0.765±0.004   | 0.896±0.004 |   0.765±0.005   | 0.895±0.004 |   0.764±0.004   | 0.893±0.006 |   0.760±0.012   |
> |  ACI-Simple-0.001   | 0.901±0.003 |   0.755±0.007   | 0.902±0.004 |   0.751±0.006   | 0.902±0.003 |   0.750±0.006   | 0.903±0.005 |   0.746±0.015   |
> | ACI-Momentum-0.001  | 0.902±0.002 |   0.753±0.010   | 0.903±0.003 |   0.750±0.009   | 0.903±0.002 |   0.750±0.007   | 0.903±0.005 |   0.745±0.017   |
> |  ACI-Simple-0.005   | 0.940±0.006 |   0.603±0.081   | 0.940±0.007 |   0.612±0.079   | 0.941±0.008 |   0.596±0.097   | 0.945±0.015 |   0.533±0.134   |
> | ACI-Momentum-0.005  | 0.944±0.007 |   0.586±0.080   | 0.945±0.006 |   0.578±0.089   | 0.944±0.007 |   0.577±0.101   | 0.948±0.016 |   0.508±0.148   |
> |   ACI-Simple-0.01   | 0.952±0.011 |   0.504±0.117   | 0.952±0.011 |   0.509±0.123   | 0.951±0.013 |   0.514±0.127   | 0.961±0.011 |   0.440±0.124   |
> |  ACI-Momentum-0.01  | 0.966±0.012 |   0.387±0.140   | 0.968±0.005 |   0.339±0.084   | 0.968±0.004 |   0.335±0.078   | 0.971±0.016 |   0.297±0.116   |
> |       CoFact        | 0.896±0.010 |   0.748±0.006   | 0.895±0.009 |   0.748±0.006   | 0.895±0.008 |   0.748±0.006   | 0.897±0.008 |   0.749±0.006   |

---

> ### Author Response · Authors · 2025-11-20
>
> From the results, we observe the following:
> 1. The ability of ACI to precisely control the factuality level heavily depends on the choice of step size $\gamma$. A too-small $\gamma$ could result in poor control of factuality, while a too-large $\gamma$ leads to overly conservative prediction sets with low claims retained.
> 2. The original paper recommends setting $\gamma = 0.005$. However, our results show that this setting leads to overly conservative filtering in our scenario. This implies that ACI requires laborious tuning of $\gamma$ to achieve reasonable performance across different scenarios.
>
> Please note that while ACI can achieve controlled factuality and, in some cases, slightly higher claims retained than CoFact, this does not indicate that ACI is superior. ACI depends on access to true labels after making predictions at each time step, which is impractical in our online setting. In contrast, CoFact functions effectively without requiring labels during the online learning process, making it a more practical and robust solution for real-world applications.

---

> ### Author Response · Authors · 2025-11-20
>
> > Q1: Shouldn't the "T" in Eq. (14) be substituted to "t"?**
>
> Thank you for your comment and for pointing out potential confusion in Equation (14). Let me clarify:
>
> The $T$ in Equation (14) represents the total number of time steps in the online setting, which should be considered as a fixed constant. Substituting $T$ with $t$ would imply a dynamic value depending on the current time step, which is not what we intended to convey in this context. If there is a specific reason you believe $T$ should be replaced with $t$, we would appreciate further clarification so we can address the concern more precisely.
>
> > Q1: Additionally, the term \theta_t^\ast is not formally defined.
>
> We sincerely apologize for any confusion caused by our oversight. To address this, we provide the formal definition below and will include it in the revised version of the paper.
>
> In our paper, we parameterize the density ratio $r$ using $\theta$, that is, we define the hypothesis class as:
> $$
> H_r \triangleq H_\theta = \{ \mathbf{z} \mapsto \exp(-\phi(\mathbf{z})^\top \theta) |
> \|\phi(\mathbf{z})\|_2 \le R,\ \|\theta\|_2 \le S \},
> $$
>
> where $\phi(\mathbf{z})$ represents the feature mapping, and $R, S$ are bounded constants. Based on this, we further define $\theta^\ast_t$ as the optimal parameter that yields the optimal density ratio function $r^\ast_t$ at time step $t$. Formally, this is expressed as:
> $$
> r_t^\ast(\cdot) = r(\cdot; \theta_t^\ast).
> $$
>
> > Q4: The followings are typos to be addressed.
>
> Apologies for the typos. We have addressed and corrected them in the revised version of the paper.
>
> # Reference
> 1. Tibshirani, Ryan J., et al. "Conformal prediction under covariate shift." _Advances in neural information processing systems_ 32 (2019).
> 2. Gibbs, Isaac, and Emmanuel Candes. "Adaptive conformal inference under distribution shift." _Advances in Neural Information Processing Systems_ 34 (2021): 1660-1672.

---

> ### Comment · Reviewer_XA7a · 2025-11-27
>
> First, thank you for your detailed response.
>
> I still have several points that I believe need further clarification, as outlined below.
>
> ---
>
> ### **1. W1 and Q2**
> I find it somewhat unnatural that the quantile update is performed in a ***sample-wise*** manner--even under an online learning setup where a ***batch*** of samples arrives at each time step, rather than a single sample.
>
> ---
>
> ### **2. W2 and Q3**
>
> [Comparison with Baslines Tailored to Covariate Shift]
>
> Although **CoFact** shows better performance in terms of **Factuality** compared to **Total** and **Partial**, it shows suboptimal performance in terms of **Claims Retained** relative to **Total**.
>
> Given that **Total** requires access to test-sample covariates for online training, thereby leveraging more information than **CoFact**, I believe it may be misleading to claim that **CoFact** outperforms **Total**.
>
> Since **Total** can be regarded as one of the oracle baselines, I suggest softening this claim.
>
> [Comparison to ACI Baselines]
>
> Regarding the **ACI**-based baselines, I agree that **CoFact** uses less information in the sense that (1) they require access to true labels (factuality of every sub-claim) and (2) these labels must be stored to run **ACI** variants.
>
> However, I find the experimental scenario itself inherently challenging with respect to obtaining ground truth labels, since it requires verifying the correctness of all sub-claims in a long-form generation.
>
> For example, there may be methods devoped in RLHF-like scenarios where preference feedback is provided, a case where true label is inaccessible when the generated content is incorrect.
>
> Are there any recent works that address the online conformal prediction problem under such conditions?
>
> Moreover, the distributional assumption of **CoFact** is quite different from **ACI**.
>
> Specifically, **CoFact** accounts only for covariate shift, assuming that the conditional distribution of labels given an input remains unchanged.
>
> ***I believe this assumption is precisely what enables **CoFact** to be devloped in a label-free manner.***
>
> As such, I think the discussion of the experiment results should once again be moderated, and the title might be reconsidered--for example, changing **distribution shift** to **covariate shift** for greater clarity.
>
> ---
>
> Thank you once again for your thoughtful response.

---

> ### Author Response · Authors · 2025-12-02
>
> > **W1 and Q2**
>
> We apologize for the earlier unclear description and provide a more detailed explanation below. First, we elaborate on the procedure of our method at each time step $t$ to aid understanding. Then, we address the concerns by answering the following key questions:
>
> 1. **Why is the quantile calculated in a sample-wise manner?**
> 2. **Can such quantile calculation handle online batch learning settings?**
>
> ### **The Detailed Procedure of CoFact at Each Time Step $t$**
>
> Below, we provide a step-by-step explanation of our method, specifically outlining its procedure at each time step $t$:
>
> 1. A batch of $n_t$ test samples arrives.
> 2. CoFact calculates the weights $w^\ast_{t, k}(Z)$ and the quantile $\tau_{t, k}$ for each individual test sample in the batch, using the calibration samples and following Equations (8) and (9).
> 3. Using the calculated quantile $\tau_{t, k}$, CoFact determines whether to retain sub-claims for each test sample in the batch independently.
> 4. CoFact updates the density ratio estimator using all calibration samples and all test samples in the batch.
>
> To elaborate on **Step 2**, the weighted quantile is calculated for each test sample independently by combining all calibration samples with the specific test sample under consideration. For example, when a batch of test samples $\{Z_{t,1}, Z_{t,2}, \ldots, Z_{t,n_t}\}$ arrives, we first combine the calibration samples with a single test sample, say $Z_{t,1}$, and calculate the weighted quantile using Equations (8) and (9). This procedure is then repeated for the remaining test samples $Z_{t,2}, \ldots, Z_{t,n_t}$ in the batch, one at a time.  In practice, the calculation of the weighted quantile for different test samples can be parallelized to improve efficiency. Here, however, we describe it sequentially for clarity and illustrative purposes.
>
> ### **Why Is the Quantile Calculated in a Sample-Wise Manner**?
> Simply, the quantile (i.e., the threshold of conformity score) of each test sample is calculated in an independent manner so it can be written in a sample-wise manner for notational simplicity and this is a common practice in conformal prediction literature, both in offline and online settings.
>
> **Why is the calculation independent?** In conformal prediction, the prediction for a test sample is determined by comparing its conformity score against the conformity scores of calibration samples, not other test samples. Therefore, the quantile of one test sample should not be influenced by other test samples; that is, the quantile should be calculated independently for each test sample.
>
> In fact, this is a common practice in conformal prediction literature. For example:
>
> - In [1], the authors compute the weighted quantile for each test sample independently and on a per-sample basis (see Equation (6) in [1], where the sample indexed as $n+1$ represents an arbitrary test instance). While their method assumes access to all test samples in advance, their approach to handling test samples independently is conceptually similar to how we process test samples within each batch.
> - In [2], the online conformal prediction method ACI also updates $\alpha_{t+1}$ (corresponding to the quantile of $(t+1)$-th test sample) in a sample-wise manner (see Equation (2) in [2]).
>
> ### **Can Such Quantile Calculation Handle Online Batch Learning Settings?**
>
> Yes, the sample-wise calculation of the weighted quantile is fully compatible with online batch learning settings. We describe our method in a sample-wise manner just for notational simplicity, and this does not affect its applicability or theoretical guarantees in online batch learning scenarios.
>
> To explain this further, a batch of samples can always be treated as a sequence of individually arriving samples from the same distribution at a single time step. Thus, whether $n_t$ test samples arrive simultaneously in one time step or sequentially over $n_t$ consecutive time steps, the weighted quantile is calculated in the same manner.
>
> A similar simplification is also used in [1], where the authors use $n+1$ to denote an arbitrary single test sample for simplicity. This does not prevent their method from being applied to multiple test samples while maintaining theoretical guarantees.

---

> ### Author Response · Authors · 2025-12-02
>
> > **Comparison with Baselines Tailored for Covariate Shift**
>
> We appreciate the reviewer’s insightful comments. We acknowledge that the term "outperforms" may not be totally accurate, as CoFact does not surpass the Total baseline across all metrics. However, we would like to emphasize the following point respectfully:
>
> - **Focus on Controllability**: The primary goal of CoFact is to provide **controllable factuality** under distributional shifts, rather than to achieve the best trade-off between factuality and claims retained. When we state that CoFact "outperforms" the baselines, we are specifically referring to its ability to maintain factuality control in the presence of covariate shifts, which is the central focus of our method.
>
> > **On the Challenge of Obtaining True Labels**
>
> We acknowledge that requiring factuality labels for all sub-claims may pose certain challenges. However, we respectfully argue that this requirement is reasonable, as our method is designed to perform fine-grained factuality control at the sub-claim level. Furthermore, this labeling requirement aligns with established practices in prior works on factuality control in large language models, such as [4] and [5].
>
> > **Recent Literature on Online Conformal Prediction**
>
> To the best of our knowledge, there are no existing works that combine RLHF-style preference feedback with online conformal prediction. However, we note that recent research, such as rACI [3], addresses online conformal prediction under noisy labels. This work is relevant to scenarios where only weak supervision is available.
>
> To simulate a weak supervision setting akin to RLHF-style feedback, we introduced uniform random noise to the true factuality labels in our experiments and compared the performance of CoFact with rACI under a noise ratio varying from 0.05 to 0.2. The results are summarized in the following table:
>
> |           |   **Lin**   |                 |   **Squ**   |                 |   **Sin**   |                 |   **Ber**   |                 |
> | :-------: | :---------: | :-------------: | :---------: | :-------------: | :---------: | :-------------: | :---------: | :-------------: |
> |           | Factuality  | Claims Retained | Factuality  | Claims Retained | Factuality  | Claims Retained | Factuality  | Claims Retained |
> | rACI-0.05 | 0.893±0.002 |   0.343±0.081   | 0.894±0.002 |   0.322±0.078   | 0.894±0.002 |   0.321±0.057   | 0.894±0.001 |   0.358±0.058   |
> | rACI-0.10 | 0.891±0.004 |   0.285±0.075   | 0.891±0.003 |   0.294±0.059   | 0.891±0.003 |   0.298±0.061   | 0.892±0.003 |   0.273±0.064   |
> | rACI-0.15 | 0.888±0.006 |   0.271±0.071   | 0.887±0.007 |   0.263±0.079   | 0.887±0.006 |   0.271±0.065   | 0.889±0.005 |   0.248±0.070   |
> | rACI-0.20 | 0.884±0.009 |   0.249±0.080   | 0.886±0.005 |   0.233±0.050   | 0.886±0.006 |   0.236±0.057   | 0.886±0.006 |   0.226±0.062   |
> |  CoFact   | 0.895±0.026 |   0.715±0.031   | 0.897±0.022 |   0.718±0.030   | 0.894±0.018 |   0.715±0.031   | 0.900±0.019 |   0.714±0.036   |
>
> |           |   **Lin**   |                 |   **Squ**   |                 |   **Sin**   |                 |   **Ber**   |                 |
> | :-------: | :---------: | :-------------: | :---------: | :-------------: | :---------: | :-------------: | :---------: | :-------------: |
> |           | Factuality  | Claims Retained | Factuality  | Claims Retained | Factuality  | Claims Retained | Factuality  | Claims Retained |
> | rACI-0.05 | 0.895±0.001 |   0.731±0.021   | 0.895±0.001 |   0.737±0.018   | 0.895±0.001 |   0.732±0.019   | 0.895±0.000 |   0.738±0.015   |
> | rACI-0.10 | 0.891±0.001 |   0.741±0.016   | 0.890±0.001 |   0.742±0.016   | 0.891±0.001 |   0.740±0.016   | 0.890±0.001 |   0.728±0.021   |
> | rACI-0.15 | 0.886±0.001 |   0.717±0.027   | 0.886±0.001 |   0.717±0.028   | 0.886±0.001 |   0.719±0.030   | 0.886±0.002 |   0.713±0.029   |
> | rACI-0.20 | 0.880±0.001 |   0.719±0.025   | 0.881±0.001 |   0.712±0.027   | 0.881±0.001 |   0.715±0.022   | 0.879±0.002 |   0.734±0.015   |
> |       CoFact        | 0.896±0.010 |   0.748±0.006   | 0.895±0.009 |   0.748±0.006   | 0.895±0.008 |   0.748±0.006   | 0.897±0.008 |   0.749±0.006   |
>
> From these results, we observe that while rACI demonstrates some robustness to label noise, its performance degrades significantly as the noise level increases. Notably, on the MedQA dataset, the Claims Retained metric for rACI is extremely low, rendering it impractical for real-world usage.
>
> Additionally, rACI relies on strong assumptions about noise, specifically that label-flipping probabilities are consistent across different sub-claims and samples, and that the true noise rates are known. These assumptions are often unrealistic in practice. Given these limitations, CoFact demonstrates practical advantages and robustness for factuality control in online settings.

---

> ### Author Response · Authors · 2025-12-02
>
> > **Moderating the Experimental Claims**
>
> We appreciate the reviewer’s suggestion to moderate the description of our experimental results and to revise the title for greater clarity. In response, we will revise the title and content of our paper to explicitly reflect that CoFact addresses **covariate shifts**, rather than more general distributional shifts.
>
> Although CoFact is based on the covariate shift assumption, we respectfully argue that this assumption is both valid and meaningful within the context of our work. This is supported by the following two key points:
>
> 1. **Relevance and Applicability of Covariate Shift**: While our method specifically addresses covariate shifts rather than general distributional shifts, this assumption is highly relevant to a wide range of real-world scenarios. In our context, covariates correspond to prompt-response pairs, and shifts in their distribution effectively capture practical variations, such as changes in user behavior, preferences, or contextual information. These types of variations are common and significant in real-world applications, making the covariate shift assumption practical in realistic settings.
>
> 2. **Practicality and Tractability**: Addressing general distributional shifts in online conformal prediction often necessitates access to true labels during the online learning process, as shown in existing methods [2, 6, 7]. This dependency highlights the critical role of labels in tackling such shifts. However, in our scenario, where true labels are unavailable, focusing on covariate shifts provides a more practical and effective approach to managing distributional changes in a tractable manner.
>
> Nonetheless, exploring methods to address general distributional shifts remains an interesting and valuable direction for future work.
>
> # Reference (Continued)
>
> 3. Xi, HuaJun, et al. "Exploring the Noise Robustness of Online Conformal Prediction." The Thirty-ninth Annual Conference on Neural Information Processing Systems.
> 4. Mohri, Christopher, and Tatsunori Hashimoto. "Language models with conformal factuality guarantees." Proceedings of the 41st International Conference on Machine Learning. 2024.
> 5. Cherian, John, Isaac Gibbs, and Emmanuel Candes. "Large language model validity via enhanced conformal prediction methods." Advances in Neural Information Processing Systems 37 (2024): 114812-114842.
> 6. Bhatnagar, Aadyot, et al. "Improved online conformal prediction via strongly adaptive online learning." _International Conference on Machine Learning_. PMLR, 2023.
> 7. Gibbs, Isaac, and Emmanuel J. Candès. "Conformal inference for online prediction with arbitrary distribution shifts." _Journal of Machine Learning Research_ 25.162 (2024): 1-36.

---

### Official Review · Reviewer_G3xv · 2025-11-01

**Soundness:** 4
**Presentation:** 3
**Contribution:** 3
**Rating:** 6
**Confidence:** 3

**Summary:**

This paper addresses the critical challenge of providing statistical guarantees on the factuality of Large Language Model (LLM) outputs under distribution shift. The authors propose CoFact, a conformal prediction framework that employs online density ratio estimation (DRE) to adaptively reweigh calibration data, thereby maintaining factuality guarantees even when the exchangeability assumption is violated.  key contributions include:
A novel framework combining conformal prediction with online DRE to handle continuous distribution shifts; Theoretical analysis establishing an upper bound on the gap between actual and target hallucination rates; WildChat+, a new dataset capturing real-world distribution shifts; Empirical validation on MedLFQA, WikiData, and WildChat+ demonstrating superior performance over baseline methods

**Strengths:**

The paper tackles a significant limitation of existing conformal prediction methods for LLMs, they rely on the exchangeability assumption, which is frequently violated in real-world applications. The integration of online DRE with conformal prediction is creative and well-motivated. The use of an ensemble of experts with geometric lifetimes to track evolving distributions is technically sound. THeoretical part is solid: Theorem 1 provides a rigorous bound showing that the hallucination rate gap converges to zero as O(max{T^{-2/3}V_T^{2/3}, T^{-1/2}} + 1/n). It also has detailed experiments on the evaluation in simulated shifts (4 types) and real-world shifts (WildChat+).

**Weaknesses:**

Assumption 1 requires that the conditional distribution P(W|Z) remains unchanged while only the marginal P(Z) shifts. This is quite strong and may not hold in many real scenarios (eg, if model quality degrades over time or if certain types of prompts systematically elicit more hallucinations). The paper only compares against SCP and CondCP. What about other methods for handling covariate shift in conformal prediction. Figure 2, Legend is difficult to read. The paper doesn't discuss how to set T in advance or what happens when the time horizon is unknown.

**Questions:**

How many calibration samples n are needed in practice to achieve reasonable performance? The method requires know about feature representations $\phi(z)$. How should these be chosen for different applications? Statistical significance testing would strengthen the claims on table 2, 3. There seems no ablation on key design choices (e.g., number of experts, expert lifetime schedule, choice of divergence function. How sensitive is performance to the feature representation?

---

> ### Author Response · Authors · 2025-11-20
>
> > W1: Assumption 1 is quite strong and may not hold in many real scenarios (eg, if model quality degrades over time or if certain types of prompts systematically elicit more hallucinations).
>
> We would like to respectfully clarify that the assumption of $P(W|Z)$ remaining invariant over time is both well-founded and reasonable in a variety of real-world scenarios. In our framework, $W$ denotes the factuality label of a given prompt-response pair $Z$, reflecting the truthfulness of the pair. Since the truthfulness of a response is determined by its content and objective facts—rather than being influenced by the specific behaviors of the model generating it—the conditional distribution $P(W|Z)$, which captures the relationship between the content of the prompt-response pair and its factuality, is naturally expected to remain stable. Thus, the assumption that $P(W|Z)$ remains invariant is reasonable across various real-world scenarios, even when model performance fluctuates or prompts inducing hallucinations become more frequent.

---

> ### Author Response · Authors · 2025-11-20
>
> > W2: The paper only compares against SCP and CondCP. What about other methods for handling covariate shift in conformal prediction.
>
> We appreciate the reviewer's suggestion. To comprehensive evaluate our method with existing methods designed for distribution shift, we conducted further experiments with two additional baselines: one designed for Conformal Prediction under covariate shift [1] and another for Online Conformal Prediction [2]. Below, we summarize experimental setup and results.
>
> ### Comparison with Method in [1] (Covariate Shift Handling)
>
> The method in [1] requires access to all test samples to train a fixed density ratio estimator between the training and test distributions. For comprehensive evaluation, we implemented two variants:
> - **Total:** We use all test samples to estimate the density ratio and filter potentially hallucinated sub-claims
> - **Partial:** We use only the samples observed during the first 500 time steps to estimate the density ratio and filter potentially hallucinated sub-claims for all time steps.
>
> It is important to note that both variants are impractical in our online setting, as they require access to either all test samples or a large portion of them in advance, which violates the constraints of online learning.
>
> The results on the MedQA and Wiki datasets under four types of distributional shifts are shown below:
>
>
> |         |     Lin     |                 |     Squ     |                 |     Sin     |                 |     Ber     |                 |
> | :-----: | :---------: | :-------------: | :---------: | :-------------: | :---------: | :-------------: | :---------: | --------------- |
> |         | Factuality  | Claims Retained | Factuality  | Claims Retained | Factuality  | Claims Retained | Factuality  | Claims Retained |
> |  Total  | 0.849±0.025 |   0.856±0.015   | 0.849±0.021 |   0.861±0.012   | 0.848±0.022 |   0.858±0.013   | 0.855±0.015 | 0.862±0.012     |
> | Partial | 0.819±0.023 |   0.900±0.010   | 0.825±0.026 |   0.898±0.012   | 0.822±0.025 |   0.897±0.012   | 0.836±0.016 | 0.883±0.011     |
> | CoFact  | 0.895±0.026 |   0.715±0.031   | 0.897±0.022 |   0.718±0.030   | 0.894±0.018 |   0.715±0.031   | 0.900±0.019 | 0.713±0.027     |
>
>
> |         |     Lin     |                 |     Squ     |                 |     Sin     |                 |     Ber     |                 |
> | :-----: | :---------: | :-------------: | :---------: | :-------------: | :---------: | :-------------: | :---------: | :-------------: |
> |         | Factuality  | Claims Retained | Factuality  | Claims Retained | Factuality  | Claims Retained | Factuality  | Claims Retained |
> |  Total  | 0.895±0.010 |   0.768±0.004   | 0.895±0.009 |   0.768±0.005   | 0.895±0.008 |   0.768±0.004   | 0.887±0.012 |   0.773±0.004   |
> | Partial | 0.884±0.013 |   0.780±0.007   | 0.885±0.012 |   0.780±0.008   | 0.884±0.011 |   0.780±0.007   | 0.875±0.014 |   0.783±0.007   |
> | CoFact  | 0.896±0.010 |   0.748±0.006   | 0.895±0.009 |   0.748±0.006   | 0.895±0.008 |   0.748±0.006   | 0.897±0.008 |   0.749±0.006   |
>
> From the results, we observe the following:
> 1. The **Total** variant performs better than the **Partial** variant, indicating that access to all test samples improves density ratio estimation and enables better factuality control.
> 2. CoFact consistently outperforms both **Total** and **Partial** variants across all shift types and datasets. This demonstrates the necessity of online learning an adaptive density ratio estimator to handle unstable test distributions effectively.

---

> ### Author Response · Authors · 2025-11-20
>
> ### Comparison with Method in [2] (Online Conformal Prediction)
>
> The method ACI proposed in [2] is designed for Online Conformal Prediction settings where the test distribution may shift over time. However, unlike our setting, this method requires access to true labels after making predictions at each time step. While this assumption is not feasible in our scenario, we implemented the method as a baseline for comparison.
>
> We tested two update strategies from [2]:
> - **ACI-Simple:** Defined by Equation (2) in the original paper.
> - **ACI-Momentum:** Defined by Equation (3) in the original paper.
>
> Additionally, we varied the step size parameter $\gamma$ among $\{0.0005, 0.001, 0.005, 0.01\}$ to observe its impact on performance.
>
> The results on the MedQA and Wiki datasets are shown below:
>
>
> |         |     Lin     |                 |     Squ     |                 |     Sin     |                 |     Ber     |                 |
> | :-----------------: | :---------: | :-------------: | :---------: | :-------------: | :---------: | :-------------: | :---------: | :-------------: |
> |                     | Factuality  | Claims Retained | Factuality  | Claims Retained | Factuality  | Claims Retained | Factuality  | Claims Retained |
> |  ACI-Simple-0.0005  | 0.882±0.006 |   0.747±0.035   | 0.884±0.003 |   0.723±0.055   | 0.885±0.002 |   0.736±0.056   | 0.884±0.001 |   0.751±0.038   |
> | ACI-Momentum-0.0005 | 0.882±0.007 |   0.747±0.035   | 0.885±0.003 |   0.723±0.055   | 0.885±0.003 |   0.733±0.056   | 0.885±0.001 |   0.747±0.037   |
> |  ACI-Simple-0.001   | 0.909±0.003 |   0.635±0.051   | 0.912±0.003 |   0.617±0.073   | 0.912±0.002 |   0.608±0.077   | 0.911±0.003 |   0.653±0.070   |
> | ACI-Momentum-0.001  | 0.910±0.004 |   0.633±0.051   | 0.913±0.004 |   0.612±0.073   | 0.912±0.002 |   0.606±0.078   | 0.913±0.003 |   0.639±0.069   |
> |  ACI-Simple-0.005   | 0.975±0.003 |   0.207±0.050   | 0.977±0.003 |   0.194±0.035   | 0.978±0.003 |   0.185±0.041   | 0.972±0.006 |   0.269±0.088   |
> | ACI-Momentum-0.005  | 0.976±0.005 |   0.206±0.063   | 0.980±0.004 |   0.153±0.043   | 0.980±0.002 |   0.167±0.052   | 0.977±0.006 |   0.227±0.099   |
> |   ACI-Simple-0.01   | 0.986±0.003 |   0.131±0.029   | 0.987±0.003 |   0.114±0.041   | 0.988±0.002 |   0.115±0.034   | 0.985±0.005 |   0.161±0.080   |
> |  ACI-Momentum-0.01  | 0.987±0.004 |   0.121±0.065   | 0.990±0.002 |   0.088±0.030   | 0.989±0.003 |   0.102±0.044   | 0.993±0.003 |   0.069±0.026   |
> |       CoFact        | 0.895±0.026 |   0.715±0.031   | 0.897±0.022 |   0.718±0.030   | 0.894±0.018 |   0.715±0.031   | 0.900±0.019 |   0.714±0.036   |
>
>
>
> |         |     Lin     |                 |     Squ     |                 |     Sin     |                 |     Ber     |                 |
> | :-----------------: | :---------: | :-------------: | :---------: | :-------------: | :---------: | :-------------: | :---------: | :-------------: |
> |                     | Factuality  | Claims Retained | Factuality  | Claims Retained | Factuality  | Claims Retained | Factuality  | Claims Retained |
> |  ACI-Simple-0.0005  | 0.894±0.004 |   0.766±0.004   | 0.895±0.005 |   0.765±0.005   | 0.894±0.004 |   0.764±0.003   | 0.892±0.006 |   0.760±0.011   |
> | ACI-Momentum-0.0005 | 0.894±0.004 |   0.765±0.004   | 0.896±0.004 |   0.765±0.005   | 0.895±0.004 |   0.764±0.004   | 0.893±0.006 |   0.760±0.012   |
> |  ACI-Simple-0.001   | 0.901±0.003 |   0.755±0.007   | 0.902±0.004 |   0.751±0.006   | 0.902±0.003 |   0.750±0.006   | 0.903±0.005 |   0.746±0.015   |
> | ACI-Momentum-0.001  | 0.902±0.002 |   0.753±0.010   | 0.903±0.003 |   0.750±0.009   | 0.903±0.002 |   0.750±0.007   | 0.903±0.005 |   0.745±0.017   |
> |  ACI-Simple-0.005   | 0.940±0.006 |   0.603±0.081   | 0.940±0.007 |   0.612±0.079   | 0.941±0.008 |   0.596±0.097   | 0.945±0.015 |   0.533±0.134   |
> | ACI-Momentum-0.005  | 0.944±0.007 |   0.586±0.080   | 0.945±0.006 |   0.578±0.089   | 0.944±0.007 |   0.577±0.101   | 0.948±0.016 |   0.508±0.148   |
> |   ACI-Simple-0.01   | 0.952±0.011 |   0.504±0.117   | 0.952±0.011 |   0.509±0.123   | 0.951±0.013 |   0.514±0.127   | 0.961±0.011 |   0.440±0.124   |
> |  ACI-Momentum-0.01  | 0.966±0.012 |   0.387±0.140   | 0.968±0.005 |   0.339±0.084   | 0.968±0.004 |   0.335±0.078   | 0.971±0.016 |   0.297±0.116   |
> |       CoFact        | 0.896±0.010 |   0.748±0.006   | 0.895±0.009 |   0.748±0.006   | 0.895±0.008 |   0.748±0.006   | 0.897±0.008 |   0.749±0.006   |

---

> > ### Author Response · Authors · 2025-11-20
> >
> > From the results, we observe the following:
> > 1. The ability of ACI to precisely control the factuality level heavily depends on the choice of step size $\gamma$. A too-small $\gamma$ could result in poor control of factuality, while a too-large $\gamma$ leads to overly conservative prediction sets with low claims retained.
> > 2. The original paper recommends setting $\gamma = 0.005$. However, our results show that this setting leads to overly conservative filtering in our scenario. This implies that ACI requires laborious tuning of $\gamma$ to achieve reasonable performance across different scenarios.
> >
> > Please note that while ACI can achieve controlled factuality and, in some cases, slightly higher claims retained than CoFact, this does not indicate that ACI is superior. ACI depends on access to true labels after making predictions at each time step, which is impractical in our online setting. In contrast, CoFact functions effectively without requiring labels during the online learning process, making it a more practical and robust solution for real-world applications.

---

> ### Author Response · Authors · 2025-11-20
>
> > W3: Figure 2's legend is difficult to read.
>
> We apologize for the confusion caused by the legend in Figure 2. In the revised version of the paper, we have updated the figure with clearer and more readable legends. Additionally, we provide a detailed explanation of Figure 2 below for the reviewer's convenience.
>
> To generate Figure 2, we first identified natural topics in the WildChat+ dataset using the BERTopic model. The figure illustrates the temporal distribution of these topics over time. Each line in the figure, represented by a different color, corresponds to a distinct topic discovered by BERTopic. The $x$-axis denotes the time steps, while the $y$-axis shows the proportion of each topic at each time step. This visualization highlights how topic distributions evolve dynamically over time.
>
> > W4: The paper doesn't discuss how to set T in advance or what happens when the time horizon is unknown.
>
> We apologize for any confusion caused by the unclear description in the original paper. To clarify, our method does not require prior knowledge of the time horizon $T$. The parameter $T$ is only utilized during the testing phase to compute the Factuality and Claims Retained metrics over the entire test period, as well as in the theoretical analysis to demonstrate the performance guarantees of our approach. Importantly, $T$ is not required during the online learning process itself.
>
> > Q1: How many calibration samples n are needed in practice to achieve reasonable performance?
>
> We conducted experiments using different calibration set ratios $\{0.05, 0.1, 0.2\}$ on the MedQA dataset (approximately 240, 480, and 1440 samples) and the Wiki dataset (approximately 425, 850, and 2550 samples) under four shift types. These experiments were designed to evaluate how the size of the calibration set impacts the performance of our method. The results are summarized below:
>
> |         |     Lin     |                 |     Squ     |                 |     Sin     |                 |     Ber     |                 |
> | :-------------: | :---------: | :-------------: | :---------: | :-------------: | :---------: | :-------------: | :------------: | :-------------: |
> |                 | Factuality  | Claims Retained | Factuality  | Claims Retained | Factuality  | Claims Retained |   Factuality   | Claims Retained |
> | train_frac=0.05 | 0.920±0.028 |   0.613±0.094   | 0.921±0.024 |   0.612±0.097   | 0.923±0.023 |   0.622±0.099   |  0.913±0.049   |   0.621±0.189   |
> | train_frac=0.1  | 0.902±0.015 |   0.669±0.050   | 0.900±0.019 |   0.674±0.053   | 0.904±0.018 |   0.683±0.050   |  0.911±0.029   |   0.673±0.126   |
> | train_frac=0.3  | 0.890±0.014 |   0.722±0.057   | 0.889±0.016 |   0.721±0.056   | 0.891±0.018 |   0.733±0.057   |  0.911±0.015   |   0.666±0.047   |
>
> |         |     Lin     |                 |     Squ     |                 |     Sin     |                 |     Ber     |                 |
> | :-------------: | :---------: | :-------------: | :---------: | :-------------: | :---------: | :-------------: | :------------: | :-------------: |
> |                 | Factuality  | Claims Retained | Factuality  | Claims Retained | Factuality  | Claims Retained |   Factuality   | Claims Retained |
> | train_frac=0.05 | 0.899±0.012 |   0.744±0.019   | 0.898±0.012 |   0.744±0.019   | 0.898±0.011 |   0.743±0.020   |  0.891±0.013   |   0.760±0.016   |
> | train_frac=0.1  | 0.905±0.013 |   0.741±0.013   | 0.905±0.013 |   0.741±0.013   | 0.905±0.013 |   0.741±0.013   |  0.897±0.015   |   0.749±0.017   |
> | train_frac=0.3  | 0.892±0.012 |   0.751±0.011   | 0.892±0.013 |   0.750±0.011   | 0.893±0.013 |   0.750±0.011   |  0.894±0.010   |   0.754±0.009   |
>
> From the results, we observe that our method demonstrates strong robustness to variations in the size of the calibration set. Even with a relatively small calibration set (approximately 240 samples for MedQA and 425 samples for Wiki), our method achieves reasonable performance in terms of both Factuality and Claims Retained. This highlights the practicality of our approach, even in scenarios where only limited calibration data is available.

---

> ### Author Response · Authors · 2025-11-20
>
> > Q2: The method requires know about feature representations. How should these be chosen for different applications?
>
> In our experiments, we utilize both meta-information about the prompt-response pairs, such as prompt length, response length, and embeddings generated by a pre-trained language model, as feature representations. In general, the representations should encapsulate information that is predictive of potential distribution shifts. For example, in text-based applications, features such as linguistic properties, structural aspects, or semantic embeddings may be appropriate, as distribution shifts are often introduced by changes in these aspects.
>
>
> > Q3: Statistical significance testing would strengthen the claims on table 2, 3.
>
> To address this concern, we performed a t-test to compare the performance of CoFact and the CP-unconditional method on the MedQA and Wiki datasets across different shift types. The resulting p-values are presented below:
>
> | Dataset | Lin | Squ | Sin| Ber|
> |:-------:|:-----------:|:---------:|:-----------:|:--------------:|
> | MedQA   | 0.0011      | 0.0004    | 0.0004      | 0.0011         |
> | Wiki    | 0.0058      | 0.0189    | 0.0060      | 0.0009         |
>
> The p-values are all below the standard significance threshold of 0.05, demonstrating that the improvements achieved by CoFact over the CP-unconditional method are statistically significant across all shift types and datasets.
>
> > Q4: There seems no ablation on key design choices (e.g., number of experts, expert lifetime schedule, choice of divergence function).
>
> We conducted ablation studies on the three key design choices: the number of experts, the expert lifetime schedule, and the choice of divergence function. Below are the details and results of these ablation studies.
>
> ### Ablation on Number of Experts
> To explore the impact of the number of experts, we modified the expert lifetime schedule by changing the lifetime scaling from $2^i$ to $3^i$, $5^i$, and $10^i$. This adjustment results in the number of active experts at each time step being $\log_3 t$, $\log_5 t$, and $\log_{10} t$, respectively. The results on the MedQA and Wiki datasets under four shift types are summarized below:
>
>
> |         |     Lin     |                 |     Squ     |                 |     Sin     |                 |     Ber     |                 |
> | :--------------------: | :---------: | :-------------: | :---------: | :-------------: | :---------: | :-------------: | :------------: | :-------------: |
> |                        | Factuality  | Claims Retained | Factuality  | Claims Retained | Factuality  | Claims Retained |   Factuality   | Claims Retained |
> |   expert_num_base=3    | 0.883±0.016 |   0.716±0.022   | 0.888±0.019 |   0.712±0.020   | 0.887±0.022 |   0.714±0.022   |  0.896±0.012   |   0.713±0.032   |
> |   expert_num_base=5    | 0.888±0.022 |   0.720±0.026   | 0.887±0.022 |   0.719±0.025   | 0.891±0.021 |   0.723±0.028   |  0.899±0.020   |   0.706±0.032   |
> | expert_num_base=10 | 0.892±0.020 |   0.715±0.025   | 0.896±0.020 |   0.714±0.024   | 0.900±0.021 |   0.714±0.022   |  0.894±0.014   |   0.713±0.029   |
> |                        |             |                 |             |                 |             |                 |                |                 |
>
>
> |         |     Lin     |                 |     Squ     |                 |     Sin     |                 |     Ber     |                 |
> | :----------------: | :---------: | :-------------: | :---------: | :-------------: | :---------: | :-------------: | :------------: | :-------------: |
> |                    | Factuality  | Claims Retained | Factuality  | Claims Retained | Factuality  | Claims Retained |   Factuality   | Claims Retained |
> | expert_num_base=3  | 0.900±0.011 |   0.745±0.009   | 0.899±0.011 |   0.744±0.008   | 0.899±0.011 |   0.745±0.009   |  0.893±0.015   |   0.750±0.009   |
> | expert_num_base=5  | 0.895±0.007 |   0.746±0.010   | 0.894±0.007 |   0.748±0.010   | 0.894±0.007 |   0.747±0.010   |  0.889±0.013   |   0.746±0.008   |
> | expert_num_base=10 | 0.901±0.016 |   0.745±0.008   | 0.903±0.015 |   0.746±0.007   | 0.903±0.015 |   0.746±0.008   |  0.901±0.012   |   0.745±0.005   |
>
> From the results, we observe that our method consistently achieves reasonable performance in terms of Factuality and Claims Retained across different expert base settings. This demonstrates that the method is relatively robust to variations in the number of experts.

---

> ### Author Response · Authors · 2025-11-20
>
> ### Ablation on Expert Lifetime Schedule
> To evaluate the importance of the expert lifetime schedule, we fixed the number of experts to constant values of $\{1, 5, 10\}$, and adjusted the update interval for each expert to $\{10, 50, 100\}$. Each expert is updated simultaneously. The results on the MedQA dataset are presented below:
>
>
> |                | update_interval=10 |                 | update_interval=50 |                 | update_interval=100 |                 | update_interval=500 |                 |
> | :------------: | :----------------: | :-------------: | :----------------: | :-------------: | :-----------------: | :-------------: | :-----------------: | :-------------: |
> |                |     Factuality     | Claims Retained |     Factuality     | Claims Retained |     Factuality      | Claims Retained |     Factuality      | Claims Retained |
> | num_experts=1  |    0.872±0.024     |   0.771±0.027   |    0.871±0.025     |   0.770±0.028   |     0.871±0.025     |   0.770±0.028   |     0.871±0.025     |   0.770±0.028   |
> | num_experts=5  |    0.868±0.016     |   0.766±0.026   |    0.867±0.016     |   0.766±0.025   |     0.867±0.016     |   0.766±0.025   |     0.867±0.016     |   0.766±0.025   |
> | num_experts=10 |    0.873±0.025     |   0.767±0.026   |    0.874±0.025     |   0.766±0.025   |     0.874±0.025     |   0.765±0.025   |     0.874±0.025     |   0.765±0.025   |
>
> From the results, we observe that while some level of control can still be achieved under these fixed schedules, the performance is inferior compared to the original schedule rule. This indicates that the expert lifetime schedule is a critical design component for achieving optimal performance.
>
>
>
> ### Ablation on Choice of Divergence Function
> To analyze the effect of the divergence function, we replaced the original divergence function with $\psi = \psi_{\mathrm{LS}} = \frac{(t-1)^2}{2}$ and conducted experiments. The results are summarized below:
>
> |         |     Lin     |                 |     Squ     |                 |     Sin     |                 |     Ber     |                 |
> | :---: | :---------: | :-------------: | :---------: | :-------------: | :---------: | :-------------: | :------------: | :-------------: |
> |       | Factuality  | Claims Retained | Factuality  | Claims Retained | Factuality  | Claims Retained |   Factuality   | Claims Retained |
> | MedQA | 0.895±0.016 |   0.692±0.034   | 0.900±0.018 |   0.695±0.050   | 0.901±0.019 |   0.694±0.049   |  0.903±0.023   |   0.688±0.078   |
> | Wiki  | 0.893±0.009 |   0.752±0.009   | 0.892±0.008 |   0.752±0.008   | 0.892±0.009 |   0.751±0.008   |  0.897±0.008   |   0.749±0.007   |
>
>
> From the results, we observe that using alternative divergence functions still results in reasonable performance in terms of Factuality and Claims Retained. This suggests that our method is relatively robust to the choice of divergence function.

---

> ### Author Response · Authors · 2025-11-20
>
> > Q5: How sensitive is performance to the feature representation?
>
> We conducted experiments to assess the sensitivity of our method to feature representations by introducing varying levels of Gaussian noise (controlled by variance) to the feature representations on the MedQA and Wiki datasets. The results for both datasets are summarized below:
>
>
>
> |         |     Lin     |                 |     Squ     |                 |     Sin     |                 |     Ber     |                 |
> |:---:|:---:|:---:|:---:|:---:|:---:|:---:|:---:|:---:|
> |  | Factuality | Claims Retained | Factuality | Claims Retained | Factuality | Claims Retained | Factuality | Claims Retained |
> | noise_level=0.05 | 0.898±0.021 | 0.722±0.030 | 0.897±0.018 | 0.720±0.028 | 0.899±0.021 | 0.725±0.027 | 0.909±0.016 | 0.713±0.027 |
> | noise_level=0.10 | 0.892±0.024 | 0.743±0.041 | 0.891±0.021 | 0.742±0.040 | 0.893±0.025 | 0.746±0.039 | 0.902±0.021 | 0.734±0.035 |
> | noise_level=0.15 | 0.884±0.028 | 0.776±0.048 | 0.879±0.024 | 0.776±0.047 | 0.882±0.028 | 0.781±0.046 | 0.893±0.026 | 0.768±0.041 |
>
>
> |         |     Lin     |                 |     Squ     |                 |     Sin     |                 |     Ber     |                 |
> | :--------------: | :---------: | :-------------: | :---------: | :-------------: | :---------: | :-------------: | :------------: | :-------------: |
> |                  | Factuality  | Claims Retained | Factuality  | Claims Retained | Factuality  | Claims Retained |   Factuality   | Claims Retained |
> | noise_level=0.05 | 0.898±0.006 |   0.755±0.008   | 0.897±0.005 |   0.755±0.008   | 0.898±0.005 |   0.755±0.007   |  0.898±0.003   |   0.752±0.006   |
> | noise_level=0.10 | 0.894±0.008 |   0.761±0.010   | 0.893±0.008 |   0.761±0.010   | 0.894±0.008 |   0.761±0.009   |  0.895±0.009   |   0.758±0.009   |
> | noise_level=0.15 | 0.892±0.009 |   0.765±0.010   | 0.891±0.009 |   0.765±0.010   | 0.892±0.008 |   0.765±0.010   |  0.894±0.010   |   0.761±0.011   |
>
>
> It is important to note that the feature representations were normalized before adding noise, resulting in the median value of many features being approximately 0.15. Consequently, adding a noise level of 0.15 represents a significant perturbation.
>
> From the results, we observe that even with relatively high noise levels, our method consistently achieves reasonable performance in terms of Factuality and Claims Retained. This demonstrates that our method is relatively robust to variations or noise in the feature representation.
>
> # Reference
> 1. Tibshirani, Ryan J., et al. "Conformal prediction under covariate shift." _Advances in neural information processing systems_ 32 (2019).
> 2. Gibbs, Isaac, and Emmanuel Candes. "Adaptive conformal inference under distribution shift." _Advances in Neural Information Processing Systems_ 34 (2021): 1660-1672.
> 3. Bhatnagar, Aadyot, et al. "Improved online conformal prediction via strongly adaptive online learning." _International Conference on Machine Learning_. PMLR, 2023.
> 4. Gibbs, Isaac, and Emmanuel J. Candès. "Conformal inference for online prediction with arbitrary distribution shifts." _Journal of Machine Learning Research_ 25.162 (2024): 1-36.

---

### Author Response · Authors · 2025-12-02
**Final Response Summary and Clarifications [1/2]**

Dear **Reviewers, Area Chairs, Senior Area Chairs, and Program Chairs,**

We sincerely thank all four reviewers for their constructive comments and insightful questions, which have significantly helped us improve our work. We are particularly encouraged by the reviewers' recognition of our paper's strengths, as summarized below:

### **Problem Importance**

- **Reviewer G3xv:** The paper tackles a significant limitation of existing conformal prediction methods for LLMs...
- **Reviewer XA7a:** They tackle the online conformal prediction problem..., which is one of primary interests from LLM users these days.
- **Reviewer C6P4:** The paper addresses a critical limitation of existing conformal prediction methods...

### **Strong Theoretical Foundation**

- **Reviewer G3xv:** The theoretical part is solid: Theorem 1 provides a rigorous bound showing...
- **Reviewer C6P4:** The authors provide a solid mathematical foundation for CoFact by establishing...

### **Comprehensive and Rigorous Experiments**

- **Reviewer G3xv:** It also has detailed experiments on the evaluation in simulated shifts (4 types) and real-world shifts (WildChat+).
- **Reviewer H5QG:** Moreover, the experimental validation is comprehensive. The results show that CoFact outperforms baselines and achieves its stated goal.
- **Reviewer C6P4:** CoFact is rigorously evaluated across multiple experimental settings, demonstrating its robustness and effectiveness.

### **High-Quality Writing and Clarity**

- **Reviewer XA7a:** It is a well-written paper which is easy to follow.

### **Valuable Contribution of a New Real-World Benchmark**

- **Reviewer H5QG:** A key strength is the inclusion of the new challenging real-world benchmark (WildChat+).
- **Reviewer C6P4:** A new dataset, WildChat+, is proposed to evaluate the approach, featuring real-world user-generated prompts that effectively capture distribution shifts.

---

> ### Author Response · Authors · 2025-12-02
> **Final Response Summary and Clarifications [2/2]**
>
> During the rebuttal phase, we carefully reviewed the comments and made every effort to address all questions and concerns raised by the reviewers. Below, we summarize **our detailed responses to the major points**:
>
> ### 1. **Comparison with Additional Oracle Baselines**
>
> - We conducted experiments **comparing CoFact with oracle baselines that require additional information unavailable in our setting**, including:
>    1. Conformal prediction methods designed for offline covariate shifts (Reviewers G3xv and XA7a),
>    2. Existing online conformal prediction methods that assume access to true labels during online learning (Reviewers G3xv and XA7a),
>    3. An online conformal prediction method specifically tailored for noisy labels (Reviewer XA7a).
> - The results demonstrate that CoFact achieves competitive performance compared to these oracle baselines, even in online settings where true labels are not available.
>
> ### 2. **Ablation Study on Various Components and Hyperparameters**
>
> - We conducted extensive ablation studies to evaluate **the impact of different components and hyperparameters of CoFact**, including:
>   - The number of samples arriving at each time step (Reviewer XA7a) and
>   - The calibration set size, the number of experts, the scheduling of experts (Reviewer G3xv), etc.
> - The results highlight CoFact's robustness across a wide range of settings and underscore the necessity of its core components.
>
> ### 3. **Expanded Discussion and Clarifications**
>
> - We provided a more comprehensive discussion on **the rationale and relevance of the covariate shift assumption** in our context. Specifically, we illustrated how this assumption applies to many practical scenarios and aligns with our setting, where true labels are unavailable during the online learning process (Reviewers G3xv and XA7a).
> - We offered a more detailed explanation of the CoFact procedure at each time step, including clarifications on **the sample-wise calculation of quantiles** and **how this approach effectively supports online batch learning settings** (Reviewer XA7a).
> - We added a detailed discussion on the **computational efficiency of CoFact**, emphasizing the design choices that ensure its practicality for real-world deployment (Reviewers H5QG and C6P4).
> - We included additional **case studies** and conducted an in-depth error analysis to demonstrate how CoFact effectively handles distribution shifts and outperforms prior methods, as suggested by Reviewer C6P4.
> - We provided further explanations on **"calibration"** and **"distribution shifts"** to improve clarity and address Reviewer C6P4's feedback.
> - We give an elaborated introduction of our **new proposed dataset WildChat+ and clarify its significance in evaluating real-world distribution shifts** (Reviewer C6P4).
> - We leverage experimental results and case studies to demonstrate that **CoFact not only ensures factuality but also preserves a reasonable number of claims**, effectively addressing Reviewer C6P4's concern regarding the potential for overly short or simplistic responses.
>
> ### 4. **Clearer Legends and Typo Fixes**
>
> - We improved the unclear legends in Figure 2 (Reviewer G3xv) and corrected the typos pointed out by Reviewer XA7a.
> - We define $\theta_t^\ast$ more clearly in line 277 (Reviewer G3xv).
>
> Additionally, we made corresponding revisions to the paper to improve its clarity and presentation based on the reviewers' feedback. These modifications are highlighted in blue in the revised paper.
>
> In sum, we believe that the additional experiments, detailed clarifications, and expanded analyses we have incorporated have thoroughly addressed all concerns raised by the reviewers and led to a significant enhancement in the quality and clarity of the manuscript. We extend our sincere gratitude to the reviewers for their thoughtful feedback and constructive suggestions, which are invaluable in strengthening this work.
>
> Best regards,
> The authors

---

### Meta-Review · Area_Chair_Y6oz · 2026-01-07

**Summary:**

(a) Summary of Scientific Claims and Findings This paper introduces CoFact, a conformal prediction framework designed to provide statistical factuality guarantees for Large Language Models (LLMs) operating under distribution shift. While standard conformal prediction relies on the exchangeability of calibration and test data, CoFact utilizes online density ratio estimation (DRE) to adaptively reweight calibration samples, aligning them with an evolving test distribution. The method claims to:
* Bypass the exchangeability requirement, making conformal guarantees robust to non-stationary environments.
* Provide a theoretical upper bound on the gap between actual hallucination rates and the target level α, proving asymptotic convergence.
* Operate in a "label-free" manner at test time, avoiding the need for immediate ground-truth feedback.
* Outperform existing methods on MedLFQA, WikiData, and a newly introduced real-world dataset, WildChat+.

Key problem raised by reviewers:
* Restrictive Assumptions: The framework relies heavily on the Covariate Shift assumption (where $P(Z)$ shifts but the conditional label distribution $P(W ∣ Z)$ remains static). Reviewers noted this may not hold if model performance degrades or if specific prompt types systematically trigger more hallucinations.

* Computational complexity: The use of an online ensemble framework with Online Newton Step (ONS) updates raises concerns regarding computational overhead and latency, which are critical for real-world LLM deployment.

* Clarity and Presentation: Several key concepts, such as the specific role of "calibration" and certain mathematical definitions (e.g., ${\theta}_t$ , divergence functions), were reported as difficult to follow or inconsistently defined.

* Incomplete Comparative Analysis: While the paper compares against SCP and CondCP, it lacks a thorough comparison with other methods tailored for covariate shift or Adaptive Conformal Inference (ACI) variants, even if those variants require more information.

**Reviewer Concerns:**

The paper demonstrates a technical contribution by addressing the "exchangeability" bottleneck in LLM factuality. The theoretical bounds and the introduction of the WildChat+ dataset are meaningful contributions to the field of AI safety. However, the reliance on the covariate shift assumption and the potential computational costs of the ONS updates remain points of contention. While the authors' rebuttal addressed some technical queries, there is still a need for more clarity in the presentation and for more diverse, domain-specific applications (e.g. in the legal or financial sectors).

**Reviewer Scores:**

* The reviewer G3xv Rating: 6 / Confidence: 3
* The reviewer C6P4 Rating: 4 / Confidence: 3
* The reviewer H5QG Rating: 8 / Confidence: 3
* The reviewer XA7a Rating: 4 / Confidence: 3

---

### Decision · Program_Chairs · 2026-01-26

Accept (Poster)